# EFFICIENT DISTRIBUTED OPTIMIZATION UNDER HEAVY-TAILED NOISE

**Su Hyeong Lee**[1]*, **Manzil Zaheer**[2], **Tian Li**[1]
[1]University of Chicago, [2]Meta
*sulee@uchicago.edu

## ABSTRACT

Distributed optimization has become the default training paradigm in modern machine learning due to the growing scale of models and datasets. To mitigate communication overhead, local updates are often applied before global aggregation, resulting in a nested optimization approach with inner and outer steps. However, heavy-tailed stochastic gradient noise remains a significant challenge, particularly in attention-based models, hindering effective training. In this work, we propose TailOPT, an efficient framework designed to address heavy-tailed noise by leveraging adaptive optimization or clipping techniques. We establish convergence guarantees for the TailOPT framework under heavy-tailed noise with potentially unbounded gradient variance and local updates. Among its variants, we highlight a memory and communication efficient instantiation which we call $Bi^2Clip$, which performs coordinate-wise clipping at both the inner and outer optimizers, achieving adaptive-like performance (e.g., Adam) without the cost of maintaining or transmitting additional gradient statistics. Empirically, TailOPT, including $Bi^2Clip$, demonstrates superior performance on several language tasks and models, outperforming state-of-the-art methods.

## 1 INTRODUCTION

The training of deep learning models including large language models (LLMs) has become increasingly resource-intensive, driven by expansive datasets and models with billions of parameters (Rosa et al., 2022; Liu et al., 2024b; Sriram et al., 2022; Dehghani et al., 2023). As the computational demands escalate, distributed learning has emerged as the default approach, enabling the parallel activation of training processes across multiple compute nodes such as GPUs or datacenters. However, this paradigm introduces a new bottleneck of communication overhead, especially as the progress in compute power has outpaced that of network infrastructure (Wu et al., 2023; DeepSeek-AI, 2024).

To mitigate these communication challenges, one promising strategy is the utilization of local updates. By allowing each compute node to perform multiple gradient updates locally before aggregation, the frequency and volume of intra-node communication can be significantly reduced (Smith et al., 2018; Stich, 2018; McMahan et al., 2017; Lee et al., 2024; Liu et al., 2024a; Jaghouar et al., 2024). For instance, the state-of-the-art DiLoCo algorithm for training LLMs in datacenter environments can apply around 500 local gradient updates prior to aggregation to relieve communication costs (Douillard et al., 2024). This approach naturally formulates a nested optimization problem, where *inner* optimization occurs within each compute node, and *outer* optimization is orchestrated by the coordinating node(s).

However, training attention-based models such as LLMs introduce an additional challenge due to the properties of their stochastic gradient distributions. Empirical and theoretical investigations have consistently demonstrated that the gradient noise in these models follows a heavy-tailed distribution (Ahn et al., 2024; Nguyen et al., 2019; Simsekli et al., 2019; 2020; Kunstner et al., 2024; Gorbunov et al., 2020). This heavy-tailed behavior, characterized by high or infinite variance and potentially very large deviations, poses significant challenges to the stability and convergence of existing optimization algorithms (Zhang et al., 2020b; Lee et al., 2024). Addressing these challenges necessitates the development of novel optimization strategies and a more principled understanding of their theoretical underpinnings.

In this work, we propose TailOPT, an efficient and theoretically principled nested training framework, designed to address the challenges posed by heavy-tailed gradient noise in distributed training with local updates. TailOPT introduces several key strategies, including clipping mechanisms (such as coordinate-wise or $L_2$-clipping) and adaptivity, applied at both inner and outer optimizers, to mitigate the adverse effects of heavy-tailed noise. We analyze the convergence of TailOPT while incorporating said adaptive methods, while allowing for heavy-tailed noise with unbounded variance. Our empirical and theoretical results demonstrate that TailOPT is strongly effective in mollifying heavy-tailed noise, enhancing the stability and convergence of the training dynamics across several language benchmarks as well as synthetic data. We include an extensive and detailed literature review in Appendix A.

Our contributions may be summarized as follows.

- We introduce TailOPT, a general distributed training framework for large-scale models under communication-efficient local updates and heavy-tailed gradient distributions. Among its instantiations, we highlight $Bi^2Clip$, which deploys adaptive approximations or mimicry to enhance performance while avoiding the utilization of gradient preconditioners.

- We provide convergence guarantees for a class of TailOPT algorithms that leverage adaptive optimizers and various clipping strategies, effectively addressing heavy-tailed noise with potentially infinite variance. This is achieved using a nested optimization framework, where the inner optimizer employs clipping operations to mitigate heavy-tailed gradient noise, while the outer optimizer utilizes either fully adaptive or efficient approximations of adaptive updates to guide the optimization process.

- We validate the practicality and effectiveness of TailOPT through extensive experiments on synthetic and real-world datasets in large-scale settings. Our experiments demonstrate that TailOPT produces several algorithmic instantiations that consistently outperform state-of-the-art baselines despite being more efficient.

## 2 PROBLEM FORMULATION

In distributed optimization, the global objective is constructed by taking a weighted average over the local node objectives $F_i(x)$ for model parameters $x \in \mathbb{R}^d$ and node $i$. In scenarios where data sizes at each node are unbalanced or sampling probabilities vary, the objective becomes:

$$F(x) = \sum_{i=0}^{N-1} p_i F_i(x), \tag{1}$$

where $p_i$ is proportional to the local data size of node $i$. Here, $F_i(x)$ is defined as $\mathbb{E}_{\xi \sim \mathcal{D}_i}[F_i(x, \xi)]$, where $F_i(x, \xi) = F_i(x) + \langle \xi, x \rangle$ represents the stochastic local objective, and $\mathcal{D}_i$ is the noise distribution of node $i$. This term comes from integrating the gradient noise model $\nabla F_i(x_i^t, \xi_i^t) = \nabla F(x_i^t) + \xi_i^t$, where $x_i^t, \xi_i^t$ are the parameter weights and gradient noise of node $i$ at timestep $t$. In our formulation and theoretical analysis (Section 3), we allow for both independent and identically distributed (IID) data across $N$ nodes, as commonly observed in datacenter environments, as well as more challenging non-IID data distributions. We now present the assumptions used in the analysis.

**Assumption 1** (L-smoothness). *For all $x, y \in \mathcal{X}$ and $i \in [N]$, the local objectives $F_i(x)$ satisfy $F_i(x) \leq F_i(y) + \langle x - y, \nabla F_i(y) \rangle + L_i \|x - y\|^2/2$.*

**Assumption 2** (Bounded $\alpha$-moment). *For all nodes $i \in [N]$ with noise distribution $\mathcal{D}_i$, there exists $\alpha_i \in (1, 2)$, $B_i > 0$ such that $\mathbb{E}[\|\xi_i\|^{\alpha_i}] < B_i^{\alpha_i}$.*

Assumption 2 expresses that the noise distribution can be heavy-tailed. In particular, we note that the variance of the noise can be infinite ($\alpha_i = 2$), a setting in which distributed SGD was shown to fail to converge, both empirically and theoretically (Yang et al., 2022; Lee et al., 2024) This condition on the $\alpha_i$ is 'optimally weakest', in that sending $\alpha_i \to 1^+$ recovers the integrability condition of the noise, the minimal assumption necessary to form expectations. Furthermore, we note that $\mathbb{E}\|\xi\|^\alpha < \infty \implies \mathbb{E}\|\xi\|^\beta < \infty$ for $\forall \beta < \alpha, \alpha \in \mathbb{R}$. Therefore, we let $\alpha := \min_{i \in [N]} \alpha_i \in (1, 2)$ in the proceeding analysis for notational convenience.

We also note that some works in the literature also define heavy-tailed distributions with *bounded* variance when establishing algorithm convergence bounds (e.g., Gorbunov et al. (2020); Parletta et al. (2024); Li & Liu (2022); Das et al. (2024)), which differs from our definition. We carry out our convergence proofs which subsumes the more general infinite variance setting, which naturally implies convergence under bounded stochastic gradients or variance.

## 3 TailOPT: An Efficient Heavy-Tailed Optimization Framework

In this section, we motivate the Heavy-Tailed Optimization Framework (TailOPT), a scalable training setup for heavy-tailed learning. SGD is a strong candidate given its simplicity and efficiency, but has been shown to diverge under heavy-tailed noise in both centralized (Zhang et al., 2020b) and distributed settings (Lee et al., 2024). Gradient clipping is a widely adopted technique to modulate model updates by mitigating the impact of large gradients (Menon et al., 2020; Zhang et al., 2020a; Chen et al., 2020; Koloskova et al., 2023; Yang et al., 2022). However, prior works on $L_2$ clipping of gradients or model updates (e.g., Yang et al. (2022)) generally do not adapt to gradient geometry, due to proportionally and uniformly downscaling each gradient coordinate. Therefore, smaller signals can become even more difficult to detect and propagate.

**Interpolating Adaptivity: $BiClip$.** Adaptive optimizers have consistently demonstrated superior performance for training modern architectures (Zhang et al., 2020b; Reddi et al., 2021; Lee et al., 2024). Key among adaptive methods such as Adam (Kingma & Ba, 2015) and Adagrad (Duchi et al., 2011; Streeter & McMahan, 2010) is the use of preconditioning, where preconditioners synthesized from historical gradient statistics help to procure a per-coordinate learning rate. This process dynamically modulates model updates: rare gradient coordinates are amplified, while uninformative gradients are scaled down, speeding up convergence. The trade-off, however, lies in the increased systems requirements to maintain preconditioners. For instance, deploying Adam can instantly triple the memory demand to host model parameters during minibatch backpropagation compared to vanilla SGD, due to the inclusion of first/second moment exponentially decaying moving averages of the gradient.

To take advantage of adaptivity without incurring additional memory or communication overhead, we propose a new clipping mechanism, $BiClip$, that performs coordinate-wise clipping from both above and below. $BiClip$ is motivated by an interpolation between clipped-SGD and adaptive methods, employing a stabilizing absolute-value clipping mechanism that modulates model updates while eliminating the overhead of preconditioner maintenance. Formally, we define $BiClip(\cdot)$ as follows[1] :

$$
\begin{aligned}
BiClip(u, d, x)_j := \text{sign}(x_j) \; & [d \, \chi \, (|x_j| \le d)] \\
+ \text{sign}(x_j) & [u \, \chi \, (|x_j| \ge u) + |x_j| \chi \, (d < |x_j| < u)],
\end{aligned}
\tag{2}
$$

where $\chi$ is the indicator function, $x \in \mathbb{R}^m$, $j \in [m]$, and $0 \le d \le u$ are the lower and upper clipping thresholds. $BiClip$ draws on the intuition of adaptive methods by selectively amplifying smaller gradient values while tempering larger gradients. When combined with an outer (potentially adaptive) optimizer, this approach leverages sensitive, amplified gradient updates from the participating compute nodes, thus emulating the advantages of adaptive optimization without preconditioner maintenance. This serves as the main building blocks of Algorithm 1.

---

**Algorithm 1** Heavy-Tailed Optimization (TailOPT)

---

**Require:** Initial model $x_1$, learning rate schedule $\eta_t$
    Clipping schedules $u_t \ge d_t \ge 0$,
    Synchronization timestep $z \in \mathbb{Z}_{>0}$
1: **for** $t = 1, \ldots, T$ **do**
2:     **for** each node $i \in [N]$ in parallel **do**
3:         $x_{i,0}^t \leftarrow x_t$
4:         **for** each local step $k \in [z]$ **do**
5:             Draw gradient $g_{i,k}^t = \nabla F_k(x_{i,k}^t, \xi_{i,k}^t)$
6:             $x_{i,k}^{t+1} \leftarrow x_{i,k}^t - \eta_t \cdot TailClip(u_t, d_t, g_{i,k}^t)$
7:         **end for**
8:     **end for**
9:     $\Delta_t = \frac{1}{N} \sum_{i \in [N]} \left( x_{i,z}^t - x_{t-1} \right)$
10:     $x_t = Outer\_Optimizer \left( x_{t-1}, \Delta_t \right)$
11: **end for**

---

**TailOPT.** In the TailOPT framework (Algorithm 1), the inner optimization strategy, TailClip, refers to either $BiClip$ or $L_2Clip$. In Line 10, the outer optimization can be adaptive or non-adaptive, applying clipping, adaptivity, or momentum to the aggregate pseudogradients ($\Delta_t$). Notably, $Bi^2Clip$,

---

[1]For clarity in notation, we define $0/0 := 0$.

which applies $BiClip$ in both inner and outer optimization, achieves strong empirical performance (Algorithm 4, Appendix D.3). While our focus is on the distributed setting, which aligns with practical applications, we note that $BiClip$ can also be effectively applied in centralized settings. Throughout the paper, we list the outer optimizer followed by the inner optimizer when referencing algorithms. For example, 'Adam-$BiClip$' instantiates Adam as the outer optimizer and $BiClip$ as the inner optimizer.

## 3.1 CONVERGENCE OF THE TAILOPT FRAMEWORK

In Appendix C, we provide a summary of the convergence results attained in TailOPT under various algorithmic instantiations. In particular, several variants (Adagrad/RMSProp-$TailClip$, Algorithms 5, 6) achieve the state-of-the-art convergence rate of $\mathcal{O}(1/\sqrt{T})$ (Li et al., 2024; Arjevani et al., 2023; Pillutla et al., 2024) even under the presence of infinite variance, heavy-tailed noise with local updates. In addition, to leverage the benefits of adaptivity while strictly enforcing almost identical memory and compute resources as *vanilla* SGD, we instantiate all optimizer strategies as $BiClip$ across all nodes, resulting in $Bi^2Clip$ (Algorithm 4). Theorem 2 shows convergence under heavy-tailed noise. We present convergence results for only a subset of TailOPT algorithms in the main text. For a comprehensive analysis, Appendices D.1, D.2 provide detailed convergence bounds for Avg-$L_2Clip$, and Appendices D.3, D include additional convergence analyses and precise pseudocodes for various (adaptive) instantiations of the TailOPT framework incorporating Adagrad, RMSProp, or Adam. Convergence results for certain instantiations are also extended to allow for *node drop or failures* at each round (Appendix D.2).

## 4 EXPERIMENTS

We assess the performance of various TailOPT instantiations across a range of empirical tasks, benchmarking against state-of-the-art algorithms from the literature. Extended details of the experimental setup, dataset descriptions, and extensive hyperparameter tuning procedures are provided in Appendix E. Our experiments include synthetic tests with carefully controlled heavy-tailed noise injection, as well as evaluations of real-world benchmarks on generative models.

## 4.1 CONVEX MODELS

We designed our convex, synthetic setup to explicitly control and inject heavy-tailed noise, enabling a focused study of its effects. In language tasks, the frequencies of words or tokens typically follows a heavy-tailed distribution, where a small subset of tokens occurs with high frequency, while the majority appear infrequently yet carry significant contextual information. To mirror this phenomenon, emulating a similar setup in Li et al. (2022), we partitioned the input feature space into common and rare features. Specifically, we set the first $p = 10\%$ features (or tokens) from data $X$ as common features, with each feature activated according to a Bernoulli distribution Bern(0.9). The remaining 90% of the features are configured as rare, each sampled from Bern(0.1). The weight vector $w_*$ is drawn from a standard multivariate normal distribution, $w_* \sim \mathcal{N}(0, I_m)$, and the labels are generated as $\hat{y} = Xw_* + \xi_{noise}$. A neural network with model weight $\hat{w}$ is then trained to learn the ground truth $w_*$. A comprehensive explanation of the dataset construction and experimental setup is provided in Appendix E.1. We inject noise $\xi_{noise}$ sampled from a heavy-tailed distribution, which induces

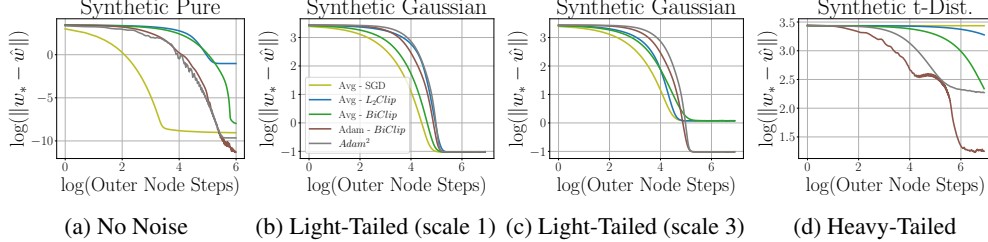

| (a) No Noise | (b) Light-Tailed (scale 1) | (c) Light-Tailed (scale 3) | (d) Heavy-Tailed |

Figure 1: The impact of heavy-tailed noise. When injected gradient noise is absent, Avg - SGD achieves the best performance (c.f., (a)). However, as the noise tails grow heavier, the performance of Avg - SGD deteriorates considerably. By contrast, both clipping mechanisms and adaptive updates demonstrate considerable performance in locating the ground truth $w_*$, and effectively mitigates the adverse effects of heavy-tailed noise (d). Light tailed noise (b-c) may not significantly destabilize the dynamics of non-adaptive Avg - SGD. The scale parameter in (c) represents the multiplier applied to $\xi_{noise}$, sampled from Gaussian and heavy-tailed distributions.

heavy-tailed stochastic gradients under MSE loss. In Figure 1, we sample from the Gaussian and Student $t$ distributions for the non-heavy-tailed and heavy-tailed $\xi_{noise}$. By default, we multiply the noise by scale 1 unless otherwise specified (Figure 1 (c)).

We observe that while SGD demonstrates strong performance in non-noisy settings, its effectiveness diminishes as noise tails become heavier—a scenario where adaptive methods and $BiClip$ excel. Similarly, $L_2Clip$ shows some ability to mitigate heavy-tailed noise but exhibits a comparable decline in performance under heavy-tailed conditions.

## 4.2 TRANSFORMER ENCODERS

Table 1: Evaluation results on GLUE Benchmark datasets during test time. Metric descriptions are given in Appendix E.3, and the full table is given as Table 10. Entries marked with 0.0 indicate failure to learn, where the performance metrics are averaged across the granularity of each datapoint. Top **first**, **second**, and **third** best-performing algorithms are highlighted. For $Adam^2$, preconditioners are transmitted between the inner and outer optimizers, whereas DiLoCo requires maintaining preconditioners on the inner optimizers, both of which incur significant communication or memory overhead. Our experiments show that $Bi^2Clip$ achieves the best aggregate performance with the minimal overhead.

| Algorithm | MNLI | QNLI | QQP (Acc/F1) | RTE | SST-2 | MRPC (Acc/F1) | CoLA | STS-B (S/P) | Average |
|---|---|---|---|---|---|---|---|---|---|
| **Avg-SGD** (McMahan et al., 2017) | 81.13 | 83.21 | 78.71/78.69 | 57.40 | 90.94 | 67.30/80.52 | 0.0 | 26.76/28.20 | 61.17 |
| **Avg-**$L_2Clip$ (Yang et al., 2022) | 81.82 | 85.68 | 80.00/79.82 | 54.51 | 91.97 | 68.38/81.22 | 0.0 | 41.27/40.96 | 64.15 |
| **Avg-Adagrad** | 84.70 | 88.79 | 87.09/83.34 | 64.26 | 93.34 | 71.56/82.63 | 27.72 | 81.93/81.26 | 76.97 |
| **Avg-Adam** | 84.97 | 89.47 | 87.66/84.09 | 64.62 | **93.80** | 81.86/87.74 | 41.41 | **86.21**/**86.55** | 80.76 |
| **Avg-**$BiClip$ | 85.08 | 89.45 | 87.83/84.12 | 66.06 | **94.03** | 71.32/82.45 | 41.40 | 84.08/84.48 | 79.12 |
| **Adagrad-SGD** (Reddi et al., 2021) | 82.40 | 86.61 | 82.51/77.68 | 71.48 | 92.08 | 85.53/89.52 | 47.80 | 40.37/42.24 | 72.69 |
| **Adagrad-**$BiClip$ | **85.54** | **90.02** | 88.60/**85.05** | **73.36** | 93.23 | 85.78/89.86 | 48.87 | 84.03/85.90 | 82.75 |
| **RMSProp-SGD** (Reddi et al., 2021) | 84.20 | 88.46 | 87.12/83.30 | **72.56** | 91.85 | 85.50/89.17 | 52.39 | 45.72/41.80 | 74.73 |
| **RMSProp-**$BiClip$ | **85.56** | **89.82** | 88.50/84.44 | 70.75 | **93.69** | 84.80/88.92 | 50.99 | **87.65**/**87.79** | **82.99** |
| **Adam-SGD** (Reddi et al., 2021) | 82.93 | 86.98 | 85.99/80.87 | 66.78 | 90.71 | 87.01/90.09 | 49.93 | 44.48/41.26 | 73.37 |
| **Adam-**$L_2Clip$ | 82.54 | 86.69 | 85.88/80.72 | 59.92 | 89.67 | 85.29/89.90 | 48.54 | 69.19/67.16 | 76.86 |
| **Adam-**$BiClip$ | 84.26 | 89.20 | **88.64**/84.74 | 69.67 | 92.43 | 86.52/90.09 | **56.12** | 82.83/79.71 | 82.20 |
| $Adam^2$ (Wang et al., 2021b) | 85.11 | 88.87 | **89.04**/**85.51** | 71.48 | 92.66 | **87.50**/**91.03** | 52.70 | 84.47/83.82 | 82.93 |
| DiLoCo (Douillard et al., 2024) | **85.68** | **89.87** | **88.78**/**85.19** | 67.87 | 91.89 | **87.99**/**91.20** | **54.77** | 85.93/84.76 | **83.08** |
| $Bi^2Clip$ | 85.06 | 89.73 | 84.93/83.97 | **76.53** | **93.80** | **89.21**/**92.44** | **60.08** | **87.07**/**86.89** | **84.52** |

To evaluate the effectiveness of our approach, we fine-tuned RoBERTa (Liu et al., 2019) on the General Language Understanding Evaluation (GLUE) benchmark (Wang et al., 2019), a widely-used suite of natural language understanding tasks. Detailed discussions for each task are provided in Appendix E.3. Table 1 presents the performance of the various state-of-the-art algorithms and TailOPT instantiations on the GLUE benchmark. Our results indicate that $L_2Clip$ enhances performance on real-world data, but adaptive methods further improve upon these results, consistently outperforming $L_2Clip$ (e.g., convergence curves in Figure 2). Notably, the clipping mechanism in TailOPT, $BiClip$, demonstrates superior performance compared to $L_2Clip$ and even surpasses Adam in aggregate during test time (c.f., $Bi^2Clip$ and $Adam^2$), highlighting its potential as an efficient and effective optimizer in real-world applications. Additionally, algorithmic instantiations achieving $\geq 80\%$ average accuracy generally employ adaptive or adaptive-approximating optimizers across all nodes. In particular, adaptivity on the inner optimizer appears crucial for performance, as SGD-based methods perform considerably worse ($\leq 75\%$). By contrast, both $BiClip$ or Adam reach $\sim 80\%$ even when combined with a simple averaging outer optimizer strategy.

## 4.3 GENERATIVE MODELS

We also evaluate TailOPT on machine translation tasks utilizing the WMT datasets, a widely used benchmark for translation research (Foundation, 2019). Specifically, we fine-tune the T5 (Raffel et al., 2020) generative model on the TED Talks and News Commentary parallel training datasets. The TED Talks dataset, originally sourced from IWSLT 2017 (Cettolo et al., 2017), comprises multilingual translations of TED Talk transcripts, while the News Commentary dataset includes parallel text from news articles across various languages. We report both Bleu and Meteor scores across several variants of source and target language translations in Table 2.

**Discussion.** For language reasoning benchmarks (i.e., GLUE datasets), the performance differences across algorithmic instantiations are particularly pronounced. While $L_2$ clipping is a common stabilization strategy, it exhibits limited effectiveness. In contrast, coordinate-wise $BiClip$ demonstrates significantly better stability and performance. Moreover, frameworks aiming to utilize or mimic

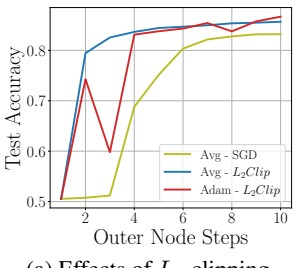
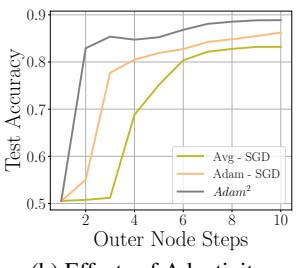
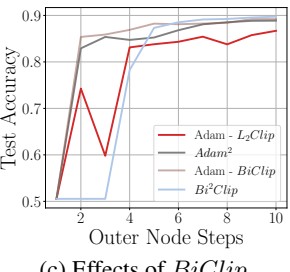

$\qquad$ (a) Effects of $L_2$ clipping $\qquad$ (b) Effects of Adaptivity $\qquad$ (c) Effects of $BiClip$

Figure 2: Convergence curves on the QNLI dataset. In (a), we see that $L_2Clip$ (one option of $TailClip$) can help to improve performance under different outer optimizers. (b) demonstrates that adaptivity further helps to mitigate the negative effects of heavy-tailed noise. In all (a-c), $L_2Clip$ performs worse than adaptive methods, but the coordinate-wise $BiClip$ optimizer performs comparably or even better than adaptive optimization frameworks, manifesting Adam-like performance. We note that the $Adam^2$ baseline, which applies Adam both in inner and outer optimization, requires transmitting preconditioners of the same size as the model weights to inner optimizers, resulting in substantial communication and memory overhead to deploy. By contrast, $Bi^2Clip$ removes the necessity of preconditioner maintenance, sidestepping this bottleneck entirely.

Table 2: Evaluation results on machine translation benchmarks. Metrics reported are BLEU and METEOR scores for various language pairs across the TED Talks and News Commentary datasets. The final column represents the average score across all metrics for each algorithm.

| Algorithm | TED Talks (en-de) | | TED Talks (en-fr) | | News Commentary (en-fr) | | Average |
|---|---|---|---|---|---|---|---|
| | BLEU | METEOR | BLEU | METEOR | BLEU | METEOR | |
| **Avg + SGD** | 28.02 | 58.52 | 27.48 | 54.67 | 30.07 | 54.13 | 42.15 |
| **Avg +** $L_2Clip$ | 28.99 | 58.94 | 29.66 | 57.40 | 31.02 | 56.73 | 43.79 |
| $Bi^2Clip$ | **29.41** | **59.18** | 30.70 | **58.13** | **31.79** | **57.69** | **44.48** |
| $Adam^2$ | 28.06 | 58.05 | **30.94** | 57.48 | 30.97 | 55.85 | 43.56 |

adaptivity in both the inner and outer optimizers generally achieve superior results, surpassing $80\%$ average performance across all benchmarks. Notably, performance is highly sensitive to the choice of inner optimizers, with SGD and $L_2$ clipping yielding the lowest results. For machine translation fine-tuning tasks however, the performance variance across different optimizer strategies is relatively small when optimal hyperparameters are selected. An expanded table with a more extensive evaluation is provided in Appendix F as Table 10.

In resource-constrained settings, $BiClip$ emerges as a strong candidate, where $Bi^2Clip$ outperforms even $Adam^2$ and DiLoCo. While its design aims to emulate adaptivity under heavy-tailed noise, $BiClip$ exhibits characteristics that can interpolate between non-adaptive and adaptive methods, capturing benefits from both without necessarily fully belonging to either paradigm (Figure 4, Appendix F). $Bi^2Clip$ retains the same memory requirements as standard vanilla SGD, which cements a highly resource-efficient adaptive approximation while strictly adhering to resource constraints.

## 5 CONCLUSION

In this work, we introduce TailOPT, a framework for scalable and efficient heavy-tailed optimization. We also propose the $BiClip$ optimizer, which utilizes nearly identical memory and compute resources to vanilla SGD yet manifests Adam-like performance. We establish convergence guarantees for our framework and provide a thorough empirical evaluation with synthetic as well as real-world datasets. Our experiments indicate that coordinate-wise $BiClip$ which clips from above and below, rather than standard $L_2Clip$, stabilizes training under heavy-tailed noise and achieves the benefits of efficient adaptive optimization, exceeding the state-of-the-art performance. Future work could explore the autonomous selection of $u_t$ and $d_t$ based on initial statistics or bespoke estimators, which could provide practical solutions. Alternatively, allowing the clipping thresholds to vary depending on coordinate partition subsets (e.g., across tensor slices), similar to compressed preconditioners such as SM3 (Anil et al., 2019), may further enhance performance. An extended conclusion with possible future directions is included in Appendix B.

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

# A ADDITIONAL RELATED WORKS

## A.1 CHALLENGES OF TRAINING TRANSFORMERS & LLMS

Training transformers and LLMs is complicated by heavy-tailed stochastic gradient distributions with very large variance, often theoretically and empirically modeled as Lévy $\alpha$-stable processes (Ahn et al., 2024; Nguyen et al., 2019; Simsekli et al., 2019; 2020; Gorbunov et al., 2020; Kunstner et al., 2024; Chezhegov et al., 2024). In such settings, non-adaptive optimization methods have been shown to destabilize during training due to the heavy-tailed nature of stochastic gradients inherent in large-scale models (Gorbunov et al., 2020; Zhang et al., 2020b). Similarly, such gradient behaviors also provably destabilize traditional averaging methods in distributed settings (Lee et al., 2024), highlighting the need for novel optimization algorithms tailored to these environments.

Recent advancements have explored centralized adaptive optimization techniques and robust gradient aggregation methods to mitigate the adverse effects of heavy-tailed noise, including gradient clipping (Simsekli et al., 2019; Juditsky et al., 2019a; Gorbunov et al., 2024a; Sadiev et al., 2023; Cutkosky & Mehta, 2021; Nguyen et al., 2023a) or adaptive clipping strategies (Chezhegov et al., 2024). However, the complexities of handling heavy-tailed noise in distributed optimization settings often prevent these algorithms and their convergence bounds from extending to scenarios with multiple nodes training in parallel, which are essential for scalable training beyond the capacity of individual nodes. Consequently, there remains a critical need for distributed optimization algorithms that are both computationally efficient and inherently robust to heavy-tailed stochastic gradients, particularly for training large-scale neural architectures. To our knowledge, developing an adaptive distributed algorithm with local updates (i.e., allowing multiple inner optimizer updates prior to outer optimizer synchronization) that converges under heavy-tailed stochastic gradient noise with theoretical in-expectation convergence guarantees has remained an open challenge.

## A.2 CLIPPING APPROACHES FOR STABILIZING TRAINING DYNAMICS

Due to its success in stabilizing model updates, gradient clipping has been extensively studied empirically (Gehring et al., 2017; Merity et al., 2018; Peters et al., 2018; Mikolov, 2012) and theoretically (Chezhegov et al., 2024; Zhang et al., 2020b; Menon et al., 2020; Zhang et al., 2020a; Chen et al., 2020; Koloskova et al., 2023; Gorbunov et al., 2020; Cutkosky & Mehta, 2021). The majority of results study the centralized setting (e.g., Liu et al. (2023b); Zhang & Cutkosky (2022); Parletta et al. (2024); Li & Liu (2022); Puchkin et al. (2024); Nguyen et al. (2023b); Gorbunov et al. (2024a)), as moving to the distributed setting provides significant challenges such as multiple inner optimizer updates prior to outer optimizer synchronization. Additionally, it was shown that using a constant clip threshold can induce gradient bias, preventing the algorithm from ever converging (Koloskova et al., 2023; Chen et al., 2020). Therefore, some works have attempted to circumvent this issue by debiasing via error feedback (Khirirat et al., 2023; Zhang et al., 2024). Other works in distributed optimization have imposed strong distributional stochastic gradient structures in the analysis. For instance, Qian et al. (2021) assume a well-behaved angular dependence between the stochastic and deterministic gradients throughout training, and Liu et al. (2022) assumes symmetric gradient noise, almost surely bounded stochastic gradients, as well as homogeneous data. By contrast, in the analysis of TailOPT, we do not impose any conditions on the noise nor data distributions except for finite

noise $\alpha$-moment for some $\alpha \in (1, 2)$. This also sharpens the sensitivity of our bounds to gradient distributions, as $\alpha$ may be selected as the minimal (or close to infimum) $\alpha$-moment value such that the moment is bounded. Moreover, our proposed clipping mechanism, realized as an instantiation of TailOPT (i.e., $BiClip$), fundamentally differs from prior approaches by integrating per-coordinate clipping in a nested setting. The inner optimization steps employ clipping operations to adapt to the gradient geometry, complemented by the outer optimizers which enhance rarified signals through adaptivity or adaptive approximations. Additionally, our algorithm and analysis accommodate local updates and allow for potentially unbounded stochastic gradient variance.

Very recently, some results studying the dynamics of heavy-tailed clipped-SGD in the distributed setting have been provided in the literature. The works Sun & Chen (2024); Gorbunov et al. (2024b); Yu et al. (2024) study distributed optimization with no local updates, where global synchronization is done after every update which has connections with batched centralized training. In particular, Sun & Chen (2024) studies the convergence of distributed clipped-sgd in the absence global synchronization, where smaller nodes communicate with their neighbors according to a strongly connected graph. Under clipping of the stochastic gradients given an $L_2$-norm constraint, projection, and averaging weights from nearby nodes, convergence is shown for strongly convex objectives. By contrast, Yu et al. (2024) proposes 'smooth-clipping' the difference between a local gradient estimator and the local stochastic gradient (using a custom smoothed $L_2$ clipping function), which is shown to converge under only the integrability condition (finite first moment) for strongly convex objectives when assuming symmetric noise distributions.

Finally, the work by Yang et al. (2022) studies the case with local updates, and is the closest in comparison to our algorithm. There, a so-called 'FAT-Clipping' algorithmic framework is proven to attain convergence under $L_2$ clipping for heavy-tailed stochastic gradients. Two variants are studied, clipping per every local iteration as well as clipping once prior to global synchronization. It is shown that per-iteration clipping achieves faster speedup and better performance (evaluated in our paper as the 'Avg + $L_2Clip$' baseline in Table 1). Our proposed clipping mechanism, $BiClip$, differs from these approaches by incorporating clipping in conjunction with adaptivity in a nested setting. The clipping operations on the inner optimizers in TailOPT temper large gradient updates while amplifying smaller ones, complemented by the outer optimizer which enhances rare covariates through adaptive mimicry or adaptivity. An added advantage of TailOPT is significant communication efficiency, as we do not transmit preconditioners from the inner and outer optimizers under iterative local updates.

### A.3 Federated Learning

Federated learning (FL) is a distributed learning paradigm designed to train machine learning models across multiple clients without requiring the transmission of raw data (McMahan et al., 2017; Li et al., 2020a; Wang et al., 2021a). This decentralized approach is particularly relevant in privacy-sensitive domains, such as healthcare and finance (Wang et al., 2022a; Liu et al., 2023a; Huang et al., 2019), where data-sharing restrictions make centralized data aggregation impractical. In its basic form, FL involves a central server that coordinates the training process by distributing a global model to a subset of clients, which can range from a dozen in cross-silo settings (e.g., hospitals (Silva et al., 2019), research institutions (Ramaswamy et al., 2019; Jiang et al., 2022), or datacenters (Douillard et al., 2024; Liu et al., 2024a; Jaghouar et al., 2024)) to millions in cross-device scenarios (e.g., mobile phones (Li et al., 2020a)). Each client performs local updates using stochastic gradient descent (SGD) on its own data and, after several local training steps, sends the aggregated models back to the server. The server then averages these updates to refine the global model. This training paradigm, commonly referred to as *FedAvg*, has become the foundation for many federated learning algorithms (McMahan et al., 2017; Reddi et al., 2021; Wang et al., 2020). Despite its effectiveness, FedAvg faces significant challenges, especially in heterogeneous environments where client data is non-IID (Wang et al., 2021a). Cross-device settings, for example, often exhibit highly diverse data distributions and stochastic gradients, as each client has access to only a small, biased subset of the overall data. These issues have motivated a rich body of research aimed at analyzing the behavior of learning algorithms under federated settings (e.g., Reddi et al. (2021)) to determine whether they can handle the complexities of real-world federated training, particularly in the presence of data heterogeneity and heavy-tails (Sun & Chen, 2024; Gorbunov et al., 2024b; Yu et al., 2024; Yang et al., 2022).

### A.4 CONVERGENCE BOUNDS

In general, there are two primary types of convergence bounds: in-probability bounds (Juditsky et al., 2019b; Davis et al., 2021; Gorbunov et al., 2022; 2020; Sadiev et al., 2023; Cutkosky & Mehta, 2021; Gorbunov et al., 2024a;b) and in-expectation bounds (Wang et al., 2022b; Li et al., 2020b; Reddi et al., 2021; Xie et al., 2020; Wang et al., 2020; Karimireddy et al., 2021; Li et al., 2023; 2022). Each type has distinct characteristics that complement the other. In-probability bounds provide an upper limit on the number of timesteps required to achieve model parameters $x$ such that $\mathbb{P}\{\mathcal{M}(x) \leq \varepsilon\} \geq 1 - \delta$ for a given evaluation metric $\mathcal{M}(x)$ (e.g., $\min_{t \in 1, \dots, T} |\nabla F(x_t)|$). Here, $\delta$ represents the failure probability, or confidence level, of the bound. As $\delta \to 0^+$, the required communication complexity or number of timesteps diverges, as expected. The key challenge is to mitigate this divergence as effectively as possible through novel algorithm designs or refined mathematical analysis, such as by deriving a polylogarithmic dependence on $\delta$ rather than a more severe inverse power-law dependence.

By contrast, in-expectation bounds complement in-probability bounds by ensuring that convergence to an optimal point is guaranteed under expectations, without a confidence level that determines the success or failure of the algorithm. However, the majority of such analyses assume a bounded noise variance, typically denoted by an upper bound $G$ or $\sigma$, which appears as constants in the upper bound of the communication complexity required for convergence (Gorbunov et al., 2020; Parletta et al., 2024; Li & Liu, 2022). Relaxing this assumption is particularly challenging because unbounded noise adds significant uncertainty to controlling model updates, as stochastic gradients with infinite variance interfere with taking expectations. Due to this dependence, some works (e.g., those studying high-probability results (Davis et al., 2021; Gorbunov et al., 2020; 2024a)) argue that in-expectation bounds are insensitive to the underlying distributional structures of the stochastic gradients, due to being compressed or approximated away by $G$. Furthermore, works such as (Lee et al., 2024) have demonstrated that under stochastic gradient descent, unbounded noise is instantaneously transmitted to the model parameters in both centralized and distributed settings, leading to severe instability and ensuring divergence in expectation. Such results elucidate the additional difficulties induced by efforts to remove the bounded gradient condition.

A recent work by Sadiev et al. (2023) provides the first high-probability results under unbounded variance for clipped-SGD applied to star-convex or quasi-convex objectives in a distributed setting without local updates. Their analysis reveals an inverse logarithmic dependence on the confidence level, prompting the question of what happens as $\delta$ approaches zero: will the method stabilize or diverge? This naturally motivates a broader inquiry—can we derive in-expectation results without the bounded variance assumption, *with* local updates, providing a complementary counterpart to the high-probability bounds? In this paper, we affirmatively answer this question by studying the dynamics of TailOPT under heavy-tailed stochastic gradient distributions. Specifically, we provide the in-expectation convergence guarantees under infinite variance and delayed synchronization, offering novel bounds that are more sensitive to distributional structures of mini-batch noise. Unlike traditional in-expectation bounds, which rely on the bounded noise variance $G$, our bounds are based on $\alpha$-moment conditions for $\alpha \in (1, 2)$. Our analysis is carried out in the distributed setting with local updates, ensuring communication efficiency.

## B FUTURE DIRECTIONS AND POSSIBLE EXTENSIONS

Efficient estimation of the clipping thresholds $d_t$ and $u_t$ in $BiClip$ remains an open avenue for research. One potential approach is to segment the thresholds into coordinate subsets (e.g., row-wise or column-wise), similar to the memory-efficient partitioning strategies employed in approximate optimizers such as SM3 (Anil et al., 2019). Alternatively, autonomous selection of $u_t$ and $d_t$ based on initial statistics or bespoke estimators could provide practical solutions. Our experiments indicate that coordinate-wise $BiClip$, rather than standard $L_2$ clipping, achieves the benefits of adaptive optimization without incurring any additional memory overhead compared to SGD. Notably, methods like Adam at least double memory usage, whereas $BiClip$ maintains parity with non-adaptive methods. This suggests that uniformly amplifying small updates can contribute to optimization efficiency. Furthermore, layer-wise $BiClip$ can be readily generalized, with proofs extending straightforwardly.

The challenges posed by heavy-tailed noise are further exacerbated in settings with non-IID data shards and diverse tokenization strategies, which introduce additional variability in gradient distribu-

tions. These challenges have catalyzed a growing body of research aimed at developing algorithms that provably converge under heavy-tailed conditions. Establishing a robust theoretical framework for such algorithms is critical for aligning with the optimization dynamics observed in modern architectures, such as transformers, where heavy-tailedness is a prominent characteristic. Our framework is also closely related to, and has applications in, federated learning. We provide a self-contained literature review in Appendix A.

Another interesting direction for future work is incorporating Adam on top of $BiClip$ instead of $L_2$ clipping to enhance stability. Notably, when each inner optimizer synthesizes a single data point, this approach effectively reduces to centralized batched Adam-$BiClip$. Thus, another potential extension is to apply $BiClip$ before passing updates to the adaptive optimizer, both at the inner optimizer, which could improve stability while potentially reducing reliance on the adaptivity parameter.

## C  CONVERGENCE OF THE TAILOPT FRAMEWORK

For the convenience of any interested readers, we provide a quick summary and overview of a selection of upcoming convergence results provided in the appendix. We carry out our analysis where the model weights $x_t \in \mathcal{X}$ are contained within a sufficiently large, compact set $\mathcal{X} \subset \mathbb{R}^d$. In such settings, finding the global minimum is known to be NP-Hard, and the standard convergence metric is the stabilization of the minimum gradient (Liu et al., 2022). We then obtain the following theorems, where the pseudocode for each optimizer instantiation is detailed in Appendix D. Up to $\mathcal{O}(d)$, the presented convergence bounds hold for both gradient-wise $L_2$ clipping as well as coordinate-wise clipping. Generalization to layer-wise clipping with varying thresholds specific to each layer or model weight tensor slice is straightforward.

**Theorem 1.** *Let assumptions 1-2 hold. Instantiate the outer optimizer in Algorithm 1 with RMSProp, giving Algorithm 6 (RMSProp-$TailClip$). Let the clipping and learning rate thresholds satisfy $\eta_t = \Theta(t^\omega)$, $\eta_\ell^t = \Theta(t^\nu)$, $d_t = \Theta(t^\gamma)$, and $u_t = \Theta(t^\zeta)$ for the conditions*

$$\nu < \min\left\{-\frac{1}{6} - \frac{4}{3}\zeta, -\frac{1}{4} - \frac{3}{2}\zeta - \frac{1}{2}\omega, -\frac{1}{2} + (\alpha - 2)\zeta\right\},$$

$$0 < \zeta < \min\left\{\frac{1}{4}, \omega + \frac{1}{2}\right\}, \quad -\frac{1}{2} < \omega \le 0,$$

$$\gamma < \min\left\{0, -\nu - \zeta - \frac{1}{2}\right\}.$$

*Then, we have that*

$$\min_{t \in [T]} \mathbb{E}\left\|\nabla F(x_t)\right\|^2 \le \sum_{i=1}^{6} \Psi_i,$$

*where the $\Psi_i$ are upper bounded by*

$$\Psi_1 \le \mathcal{O}(T^{-\omega+\zeta-\frac{1}{2}}), \quad \Psi_2 \le \mathcal{O}(T^{\omega+2\nu+3\zeta+\frac{1}{2}}),$$

$$\Psi_3 \le \mathcal{O}(T^{4\zeta+3\nu+\frac{1}{2}}), \Psi_4 \le \mathcal{O}(T^{2\nu+2\zeta+\frac{1}{2}}),$$

$$\Psi_5 \le \mathcal{O}(T^{\nu+\gamma+\zeta+\frac{1}{2}}), \quad \Psi_6 \le \mathcal{O}(T^{\nu+(2-\alpha)\zeta+\frac{1}{2}}),$$

*which guarantees convergence via an inversely proportional power law decay with respect to $T$. Here, the exponential moving average parameter of the second pseudogradient moment is fixed within the range $\widetilde{\beta}_2 \in [0, 1)$.*

In particular, the proof of this result immediately implies the following summarizing corollary.

**Corollary 1.** *Algorithm 6 (RMSProp-$TailClip$) convergences under heavy-tailed stochastic gradient noise. The maximal convergence rate can be attained in the limit $\zeta \to 0^+$ for an asymptotically near-constant upper clip threshold $u_t = \Theta(t^\zeta)$ as $\mathcal{O}(1/\sqrt{T})$.*

The full proofs of all results in this section are given in Appendix D, which holds for both convex and non-convex functions. This achieves the state-of-the-art convergence rate of $\mathcal{O}(1/\sqrt{T})$ (Li et al., 2024; Arjevani et al., 2023; Pillutla et al., 2024) even in the presence of heavy-tailed noise with

local updates. We also obtain a $\mathcal{O}(1/\sqrt{T})$ rate for an alternate instantiation (Adagrad-$TailClip$) and provide the exact algorithm in Algorithm 5 and convergence result in Theorem 6 of the appendix.

When deploying distributed optimization, adaptive optimizers such as Adam can considerably increase the memory requirements on each compute node due to preconditioner storage, which matches the model parameter tensor size. For instance, $Adam^2$ (Wang et al., 2021b), which applies Adam across all compute nodes, increases overhead by transmitting preconditioners from outer to inner optimizers to maximize performance, posing significant communication and memory challenges. Algorithm 6 (RMSProp-$TailClip$) eliminates this bottleneck by removing both preconditioner transmission and maintenance on all inner optimizers, while imitating adaptivity through $BiClip$. This naturally intuits the question of whether TailOPT can incorporate further efficient adaptive approximations on the outer optimizer, while ensuring convergence under heavy-tailed noise. This motivates $Bi^2Clip$, which leverages $BiClip$ at both inner and outer optimizers, retaining the benefits of adaptivity with minimal overhead. Convergence results are given below.

**Theorem 2.** *Let the learning rate and clipping schedules satisfy $\eta_t = \Theta(t^\omega)$, $\eta_\ell^t = \Theta(t^\nu)$, $d_t = \Theta(t^\gamma)$, $u_t = \Theta(t^\zeta)$, $\widetilde{d}_t = \Theta(t^{\widetilde{\gamma}})$, and $u_t = \Theta(t^{\widetilde{\zeta}})$. For $Bi^2Clip$ (Algorithm 4), we have that the minimum gradient satisfies*

$$\min_{t \in [T]} \mathbb{E}[\|\nabla F(x_{t-1})\|^2] \lesssim \sum_{i=1}^{7} \Psi_i,$$

*where the $\Psi_i$ are given*

$$\Psi_1 = \mathcal{O}\left(T^{-\omega-\nu-1}\right), \quad \Psi_2 = \mathcal{O}\left(T^{\omega+2\widetilde{\zeta}-\nu}\right), \quad \Psi_3 = \mathcal{O}\left(T^\gamma\right),$$

$$\Psi_4 = \mathcal{O}\left(T^{\widetilde{\gamma}-\nu}\right), \quad \Psi_5 = \mathcal{O}\left(T^{(\alpha-1)\nu+(1-\alpha)\widetilde{\zeta}}\right),$$

$$\Psi_6 = \mathcal{O}\left(T^{(1-\alpha)\zeta}\right), \quad \Psi_7 = \mathcal{O}\left(T^{\nu+\zeta}\right).$$

*To attain convergence, we impose $\zeta, \widetilde{\zeta} > 0 > \gamma, \widetilde{\gamma}$, for $\omega, \nu \leq 0$, as well as the following conditions*

$$-1 < \omega + \nu, \quad \nu + \zeta < 0, \quad \max\{\omega + 2\widetilde{\zeta}, \widetilde{\gamma}\} < \nu.$$

*Then, $Bi^2Clip$ converges with maximal rate at least $\mathcal{O}(T^{-r})$, where for $\widetilde{\varepsilon} \in (0, 1/8)$ and $\alpha > 1$,*

$$r := \min\left\{\frac{(\alpha-1)\alpha}{4}, \ \widetilde{\varepsilon}, \ \frac{\alpha-1}{4} - (1-\alpha)(\frac{1}{8} - \widetilde{\varepsilon})\right\}.$$

This gives the following corollary.

**Corollary 2.** *Algorithm 4 ($Bi^2Clip$) converges with respect to heavy-tailed stochastic gradient noise ($\alpha > 1$). For instance, if the moment is further constrained by $\alpha > 1.5$, the algorithm converges with a maximal rate of at least $\mathcal{O}(T^{-r})$ for $r = 1/8$.*

Similar to RMSProp-$TailClip$, the results hold for both convex and non-convex functions as long as the assumptions are satisfied. The convergence rate given in Corollary 2 represents a lower bound on the maximal achievable rate, obtained by a fixed selection of hyperparameters. Interestingly, our empirical results demonstrate that $Bi^2Clip$ outperforms other methods, suggesting that the current convergence bounds could be further refined.

**Discussion.** To ensure convergence and mitigate bias in the derived bound, it is necessary for the upper clipping threshold $u_t \to \infty$ and the lower clipping threshold $d_t \to 0$ as $t \to \infty$, consistent with established counterexamples that occur due to unmitigated clipping bias (Koloskova et al., 2023; Chen et al., 2020). In cases where stochastic gradients are sampled from large-variance distributions, this necessitates a continual warm-up phase that is continuously relaxed, akin to learning rate *warm-up* schemes that conclude after a finite period (Kosson et al., 2024). This may help to explain why learning rate warm-ups are observed to significantly improve training (Kalra & Barkeshli, 2024; Ma & Yarats, 2021) in the presence of heavy-tailed stochastic gradients. Finally, as the maximal bounded moment condition $\alpha$ approaches the integrability threshold ($\alpha = 1$), or as $\gamma$ nears $0^-$, the convergence bound is mollified. Despite this, in our experiments, we set $\nu = \zeta = \gamma = 0$, which yielded strong empirical performance. Intuitively, this setup corresponds to a continual amplification of informative coordinates and attenuation of uninformative covariates.

**Other Instantiations and Extensions.** For a brief overview, we have presented convergence results for only a subset of TailOPT algorithms in Appendix C. For a more comprehensive analysis, Appendices D.1, D.2 provide detailed convergence bounds for Avg-$L_2Clip$, and Appendices D.3-D.6 include additional convergence analyses and precise pseudocodes for various (adaptive) instantiations of the TailOPT framework incorporating Adagrad, RMSProp, or Adam. Convergence results for certain instantiations are also extended to allow for *node drop or failures* at each round (Appendix D.2).

## D   CONVERGENCE OF TAILOPT

In this section, we rigorously analyze the convergence of TailOPT under heavy-tailed noise, beginning with the simpler case of Avg-$L_2Clip$ to enhance readability before progressively advancing to more sophisticated TailOPT variants incorporating $BiClip$ and other adaptive outer optimizers. We first establish the foundational convergence proof for Avg-$L_2Clip$ in Appendix D.1, which serves as the basis for subsequent analyses. The proof for Avg-$L_2Clip$ studies a virtual history of model weights synthesized by inner optimizers, which is inaccessible in real-world settings except when the model updates are communicated to the outer optimizer. However, by analyzing the virtual history, we are able to attain convergence of a moving average of accessible model weights to the optimum, which can be materialized in practice. In Appendix D.2, we extend this proof to settings with partial participation and failing compute nodes, examining the resulting dynamics under heavy-tailed noise.

In Appendix D.3, we further generalize the analysis to the $Bi^2Clip$ instantiation, where $BiClip$ is applied to both the inner and outer optimizers. Notably, $Bi^2Clip$ encompasses Avg-$BiClip$ as a special case under specific hyperparameter choices, which in turn subsumes Avg-$L_2Clip$. Finally, in Appendices D.4, D.5, and D.6, we investigate the convergence properties of TailOPT when the outer optimizer is instantiated with Adagrad, RMSProp, and Adam, respectively.

### D.1   CONVERGENCE OF AVG-$L_2Clip$

We aim to model contemporary, large-scale neural network training across multiple powerful compute nodes (datacenters or GPU clusters), in which data is typically preprocessed IID to optimize for training. However, for fullest generality, we conduct our theoretical analysis in the more challenging, non-IID setting. Our setup is identical to Section 2, with some added notation. We denote $x^*$ to represent the global optimum of $F(x)$ with a minimum value $F^* = F(x^*)$, and additionally, we let $x_i^*$ be the global optimum of $F_i(x) = \mathbb{E}_\xi[F_i(x, \xi)]$, with a minimum value $F_i^* = F(x_i^*)$.

For model weight or stochastic gradient averages, we use the following notation

$$\overline{x}_t = \sum_{i=1}^{N} p_i x_{i,0}^t, \quad g_t = \sum_{i=1}^{N} p_i \cdot Clip(c_t, \nabla F_i\left(x_{i,0}^t, \xi_{i,0}^t\right)), \quad Clip(c, y) := \min\left\{1, \frac{c}{\|y\|}\right\} y.$$

The use of the notation $x_{i,0}^t$ instead of $x_i^t$ carefully reflects the flow of the proof, which studies a 'virtual synchronization' of the model weights synthesized by the inner optimizer at each time $t \in [T]$ (see Algorithm 2). In other words, we first analyze the virtual average $\overline{x}_t$ which is not materially realized except at outer optimizer synchronization steps, before modifying the proof to procure a moving average of weights which is solely dependent on those communicated to the outer optimizer, which can now be obtained.

We now present some assumptions used in the convergence analysis for this section. We take the model weight projection domain to be $\mathcal{X} = \mathcal{B}(0, B) \subset \mathbb{R}^d$, where $\mathcal{B}(0, B)$ is the closed ball centered at the origin with radius $B$. Clearly, $B > 0$ needs to be large enough to contain $x^*, x_i^* \in \mathcal{X}$ for convergence. However, we note that the convergence analysis holds for $\mathcal{X}$ any large enough compact, convex set.

**Assumption 3 ($\mu$-strong convexity).** For all $x, y \in \mathcal{X}$ and $i \in [N]$, $F_i(x)$ satisfies $F_i(x) \geq F_i(y) + \langle x - y, \nabla F_i(y)\rangle + \mu_i\|x - y\|^2/2$.

One motivation behind Assumption 3 is that while the optimization of DNNs is a non-convex problem Choromanska et al. (2015), Goodfellow et al. (2015) observe that loss surfaces are often approximately convex in practice, over a single optimization trajectory. Additionally, modern training paradigms, such as the fine-tuning of foundation models, have been empirically reported to belong to

a shared convex loss basin Wortsman et al. (2022); Izmailov et al. (2018). We note that Proposition 1 shows that gradient perturbations do not affect dominance of nor over second order approximations, which preserves the values of $L, \mu$.

Gradient clipping is a widely adopted technique to stabilize model updates by mitigating the impact of large gradients Menon et al. (2020); Zhang et al. (2020a); Chen et al. (2020); Koloskova et al. (2023). The $Clip(\cdot)$ operator rescales the gradient uniformly to ensure it remains below a predefined threshold. This procedure is mathematically equivalent to applying a dynamically adjusted, lower learning rate when large stochastic gradients are encountered. Another related technique is projection, which operates in the model weight space rather than the gradient space, effectively stabilizing the model parameters themselves instead of acting on the updates. These observations motivate Algorithm 2, which may be interpreted as dynamically modulating the learning rates as well as backtracking toward the model origin $\overline{0}$ when heavy-tailed stochastic gradient updates are realized.

---

**Algorithm 2** Avg-$L_2 Clip$

---

**Require:** Initial model $x_1$, learning rate schedule $\eta_t$, clipping schedule $c_t$
    Synchronization timestep $z \in \mathbb{Z}_{>0}$, projection domain $\mathcal{X}$
1: **for** $t = 1, \ldots, T$ **do**
2:     **for** each node $i \in [N]$ **do**
3:         Draw minibatch gradient $g_{i,0}^t = \nabla F_i(x_{i,0}^t, \xi_{i,0}^t)$
4:         $x_{i,0}^{t+1} \leftarrow x_{i,0}^t - \eta_t \cdot Clip(c_t, g_{i,0}^t)$
5:     **end for**
6:     **if** $t - 1 \in z \cdot \mathbb{Z}_{\geq 0}$ :
7:         $x_{i,0}^{t+1} \leftarrow \mathrm{Proj}_{\mathcal{X}} \left( \sum_{i \in [N]} p_i x_{i,0}^{t+1} \right)$, for $\forall i \in [N]$
8: **end for**

---

Theorem 3 demonstrates that distributed Avg-$L_2 Clip$ converges in expectation under heavy-tailed noise, despite potential clipping-induced bias. We also offer the first proof demonstrating convergence under an extension of these results to accommodate failing nodes (e.g., partial datacenter participation) for additional utility in Appendix D.2. To proceed with the analysis, we first provide a simple proposition:

**Proposition 1.** *If $F_i(x)$ is $\mu$-strongly convex (or $L$-smooth), then so is $F_i(x, \xi)$ for the identical $\mu$ (or $L$).*

*Proof.* The proof is simple. By $\mu$-strong convexity or $L$-smoothness, we have

$$F_i(x) \geq F_i(y) + \langle x - y, \nabla F_i(y) \rangle + \frac{\mu}{2} \|x - y\|^2,$$

$$F_i(x) \leq F_i(y) + \langle x - y, \nabla F_i(y) \rangle + \frac{L}{2} \|x - y\|^2.$$

Then, note the following equations for $\langle \xi, x \rangle$:

$$\langle \xi, x \rangle \geq \langle \xi, y \rangle + \langle x - y, \xi \rangle,$$
$$\langle \xi, x \rangle \leq \langle \xi, y \rangle + \langle x - y, \xi \rangle.$$

Collecting these inequalities give the result. $\qquad\square$

While clipping offers the benefit of stabilization, it introduces complexities that significantly complicate the convergence analysis. In particular, clipping induces a non-zero bias on the stochastic gradients, rendering them to be no longer unbiased estimators of the true gradient. Prior work, such as Chen et al. (2020), presents illustrative examples where using a fixed clipping threshold can bias the gradient dynamics to the extent that the optimum is no longer a steady state, preventing SGD from ever converging. Furthermore, unlike in previous analyses, our work also considers scenarios involving distributions with infinite variance, where the clipping bias is exacerbated by the presence of heavy tails. Despite these challenges, Theorem 3 demonstrates that with appropriately chosen (increasing) clipping and (decreasing) learning rate schedules, convergence of Algorithm 2 is nevertheless attainable in expectation.

**Theorem 3.** *Let Assumptions 1-3 hold, and the clipping threshold in Avg-$L_2$Clip (Algorithm 2) satisfy $c_t = c\eta_t^\gamma$ for $c > 0$ and $1/2 > \gamma > 0$. Decay the learning rate with schedule $\eta_t = r/(t+1)$ for $r > 2/\mu$, where $\mu = \min_{k \in [N]} \mu_k$ and $L = \max_{k \in [N]} L_k$. Then, we have for $\tilde{x}_T := \sum_{t=1}^{T} t\mathbb{E}[\overline{x}_t]/T(T+1)$ that*

$$F(\tilde{x}_T) - F(x^*) \leq \Psi_1 + \Psi_2 + \Psi_3 + \Psi_4,$$

*where*

$$\Psi_1 = \frac{rc^2 T^{2\gamma+1}}{(4\gamma+2)T(T+1)},$$

$$\Psi_2 = \frac{(M^\alpha + B^\alpha)^2 c^{2-2\alpha}(T^{(2-2\alpha)\gamma+1}+1)}{2(\mu - 2/r)((2-2\alpha)\gamma+1)T(T+1)},$$

$$\Psi_3 = \frac{c^{2-\alpha}rzu(M^\alpha + B^\alpha)LT^{(2-\alpha)\gamma+1}}{(\mu - 2/r)((2-\alpha)\gamma+1)T(T+1)},$$

$$\Psi_4 = \frac{r^2 c^2 z^2 u^2 L^2 (T^{2\gamma}+1)}{4\gamma(\mu - 2/r)T(T+1)}.$$

*Here, we have used the notation*

$$M = \sqrt{\max_{k \in [N], x \in \tilde{\mathcal{X}}} \frac{2L^2}{\mu}(F_i(x) - F_i(x_i^*))}, \quad \alpha = \min_{k \in [N]} \alpha_k, \quad B = \max_{k \in [N]} B_k, \quad u = \frac{z+1}{2},$$

*where $\tilde{\mathcal{X}}$ is a compact domain constructed by a uniformly closed extension of $\mathcal{X}$ with $L_2$ distance $\sum_{t=1}^{z} rct^{\gamma-1}$.*

*Proof.* Let us bound the distance between the averaged model weights $\overline{x}_t$ and the global optimum $x^*$. Assume that $t \in z \cdot \mathbb{Z}$. We consider the following function

$$f(t) = \|x^* - \text{Proj}_\mathcal{X}(\overline{x}_t - \eta_t g_t) + t(-\overline{x}_t + \eta_t g_t + \text{Proj}_\mathcal{X}(\overline{x}_t - \eta_t g_t))\|^2,$$

for which

$$f'(0) = 2\langle x^* - \text{Proj}_\mathcal{X}(\overline{x}_t - \eta_t g_t), -\overline{x}_t + \eta_t g_t + \text{Proj}_X(\overline{x}_t - \eta_t g_t)\rangle.$$

Now, consider the function

$$g(t) = \|(1-t)\text{Proj}_\mathcal{X}(\overline{x}_t - \eta_t g_t) + t\text{Proj}_\mathcal{X}(x^*) - \overline{x}_t + \eta_t g_t\|$$

By the projective property,

$$g(t) \geq \|\text{Proj}_X(\overline{x}_t - \eta_t g_t) - (\overline{x}_t - \eta_t g_t)\|.$$

holds for $t \in [0, 1]$ via convexity of $\mathcal{X}$. Additionally, $g(t)^2$ meets its minimum at $t = 0$. Therefore, we have that $\mathrm{d}g(t)^2/\mathrm{d}t_{t=0} \geq 0$ due to $g(t)^2$ being quadratic with respect to $t$. Noting that $f'(0) = \mathrm{d}g(t)^2/\mathrm{d}t|_{t=0}$, we have that $f(t)$ is monotonically increasing for $t \geq 0$, again due to properties of a quadratic. Then, $f(1) \geq f(0)$ gives that

$$\|\text{Proj}_\mathcal{X}(\overline{x}_t - \eta_t g_t) - x^*\|^2 \leq \|\overline{x}_t - \eta_t g_t - x^*\|^2.$$

Therefore, we may conclude

$$\|\overline{x}_{t+1} - x^*\|^2 = \left\|\sum_{i=1}^{N} p_i \text{Proj}_\mathcal{X}(\overline{x}_t - \eta_t g_t) - x^*\right\|^2 = \|\text{Proj}_\mathcal{X}(\overline{x}_t - \eta_t g_t) - x^*\|^2$$

$$\leq \|\overline{x}_t - \eta_t g_t - x^*\|^2 = \|\overline{x}_t - x^*\|^2 - 2\eta_t\langle\overline{x}_t - x^*, g_t\rangle + \eta_t^2\|g_t\|^2$$

$$= \|\overline{x}_t - x^*\|^2 \underbrace{-2\eta_t\langle\overline{x}_t - x^*, g_t - \nabla F(\overline{x}_t)\rangle}_{A_1} \underbrace{-2\eta_t\langle\overline{x}_t - x^*, \nabla F(\overline{x}_t)\rangle}_{A_2} + \underbrace{\eta_t^2\|g_t\|^2}_{A_3}.$$

Note that the final inequality LHS $\leq$ RHS also holds for $t \notin z \cdot \mathbb{Z}$. In bounding $A_2$, we aim to derive a term that decays $\|\overline{x}_t - x^*\|^2$ by inducing a coefficient $(1 - \tilde{c}\eta_t)\|\overline{x}_t - x^*\|^2$ for some $\tilde{c} > 0$ to be determined. By $\mu$-strong convexity of $F(x)$,

$$F(x^*) \geq F(\overline{x}_t) - \langle\overline{x}_t - x^*, \nabla F_i(\overline{x}_t)\rangle + \frac{\mu}{2}\|x^* - \overline{x}_t\|^2$$

$$\implies -(F(\overline{x}_t) - F(x^*)) - \frac{\mu}{2}\|\overline{x}_t - x^*\|^2 \geq -\langle\overline{x}_t - x^*, \nabla F(\overline{x}_t)\rangle.$$

To bound $A_1$, we consider conditional expectations

$$-2\eta_t \langle \overline{x}_t - x^*, \mathbb{E}_t[g_t] - \nabla F(\overline{x}_t) \rangle \leq 2\eta_t \|\overline{x}_t - x^*\| \|\mathbb{E}_t[g_t] - \nabla F(\overline{x}_t)\|,$$

where $\mathbb{E}_t[\cdot]$ conditions on all realizations up to time $t$. Unraveling definitions gives

$$
\begin{aligned}
\|\mathbb{E}_t[g_t] - \nabla F(\overline{x}_t)\| = \| &\sum_{i \in [N]} p_i (\mathbb{E}_t[Clip(c_t, \nabla F_i(x_{i,0}^t, \xi_{i,0}^t))] - \nabla F_i(x_{i,0}^t) + \nabla F_i(x_{i,0}^t) - \nabla F_i(\overline{x}_t))\| \\
\leq &\sum_{i \in [N]} p_i \|\mathbb{E}_t[Clip(c_t, \nabla F_i(x_{i,0}^t, \xi_{i,0}^t)) - \nabla F_i(x_{i,0}^t, \xi_{i,0}^t))]\| + \sum_{i \in [N]} p_i \|\nabla F_i(x_{i,0}^t) - \nabla F_i(\overline{x}_t)\| \\
\leq &\sum_{i \in [N]} p_i \underbrace{\mathbb{E}_t[\|Clip(c_t, \nabla F_i(x_{i,0}^t, \xi_{i,0}^t)) - \nabla F_i(x_{i,0}^t, \xi_{i,0}^t))\|]}_{A_4} + \sum_{i \in [N]} p_i L \|x_{i,0}^t - \overline{x}_t\|,
\end{aligned}
$$
(3)

where the second line used Jensen and triangle inequality, and the third line used $L$-smoothness as well as Jensen. Now, we note that clipping biases the expectation in $A_4$, and we seek to ease out a measure of the clipping bias. For this purpose, we quantify the $\alpha$-moment of the stochastic gradient:

$$2^\alpha \mathbb{E}_t \left\| \frac{\nabla F_i(x) + \xi_{i,0}^t}{2} \right\|^\alpha \leq 2^{\alpha-1} \left( \mathbb{E}_t \|\nabla F_i(x)\|^\alpha + \mathbb{E}_t \|\xi_{i,0}^t\|^\alpha \right) \leq 2^{\alpha-1} \left( \|\nabla F_i(x)\|^\alpha + B_i^\alpha \right).$$

Here, we have used the notation $B_i < \infty$ for readability, but strictly speaking this is not identical to the $B_i$ given in Assumption 2 as $\alpha := \min_{i \in [N]} \alpha_i$. Finally, the projection in each outer optimizer synchronization step ensures that the $x_{i,0}^t$ remain in a compact set $\tilde{\mathcal{X}}$. Therefore, to bound gradients, we use $L$-smoothness and $\mu$-strong convexity of $F_i(x)$ as follows:

$$\|\nabla F_i(x)\|^2 \leq L^2 \|x - x_i^*\|^2,$$

where $x_i^*$ is the optimum of $F_i(x)$. Then, convexity gives that

$$F_i(x) \geq F_i(x_i^*) + \frac{\mu}{2} \|x - x_i^*\|^2,$$

from which we conclude

$$\|\nabla F_i(x)\|^2 \leq \frac{2L^2}{\mu} (F_i(x) - F_i(x_i^*)) \leq M^2 := \max_{k \in [N], x \in \tilde{\mathcal{X}}} \frac{2L^2}{\mu} (F_i(x) - F_i(x_i^*)). \qquad (4)$$

Piecewise continuity of $F_i(x)$ is clear due to the existence of $\nabla F_i(x)$. Therefore,

$$\mathbb{E}_t \|\nabla F_i(x_{i,0}^t) + \xi_{i,0}^t\|^\alpha \leq \frac{(M^\alpha + B^\alpha)}{2}.$$

Now, note that if $\|\nabla F_i(x_{i,0}^t, \xi_{i,0}^t)\| \leq c_t$, clipping has no effect in $A_4$. Thus, we focus on the case $\|\nabla F_i(x_{i,0}^t, \xi_{i,0}^t)\| > c_t$. Additionally, clipping only downscales each stochastic gradient by a scalar, which preserves direction. Therefore,

$$
\begin{aligned}
A_4 = &\mathbb{E}_t \left[ \|Clip(c_t, \nabla F_i(x_{i,0}^t, \xi_{i,0}^t)) - \nabla F_i(x_{i,0}^t, \xi_{i,0}^t))\| \cdot \chi \left( \|\nabla F_i(x_{i,0}^t, \xi_{i,0}^t)\| > c_t \right) \right] \\
\leq &\mathbb{E}_t \left[ \|\nabla F_i(x_{i,0}^t, \xi_{i,0}^t)\| \cdot \chi \left( \|\nabla F_i(x_{i,0}^t, \xi_{i,0}^t)\| > c_t \right) \right] \\
\leq &\mathbb{E}_t \left[ \|\nabla F_i(x_{i,0}^t, \xi_{i,0}^t)\|^\alpha \cdot \|\nabla F_i(x_{i,0}^t, \xi_{i,0}^t))\|^{1-\alpha} \cdot \chi \left( \|\nabla F_i(x_{i,0}^t, \xi_{i,0}^t)\| > c_t \right) \right] \leq (M^\alpha + B^\alpha) c_t^{1-\alpha}.
\end{aligned}
$$
(5)

Putting these inequalities together, we obtain as an intermediary step for $a > 0$:

$$
\begin{aligned}
A_1 \leq &2\eta_t \|\overline{x}_t - x^*\| ((M^\alpha + B^\alpha) c_t^{1-\alpha} + \sum_{i \in [N]} p_i L \|x_{i,0}^t - \overline{x}_t\|) \\
\leq &\mu a \eta_t \|\overline{x}_t - x^*\|^2 + \frac{\eta_t}{\mu a} ((M^\alpha + B^\alpha) c_t^{1-\alpha} + L \sum_{i \in [N]} p_i \|x_{i,0}^t - \overline{x}_t\|)^2.
\end{aligned}
$$

Thus, our next step is to ease out $\|x_{i,0}^t - \overline{x}_t\| = \mathcal{O}(\eta_t)$. For this purpose, our intuition is that the drift in model weights from local updates are bounded by the update size, as well as by taking a maximum of $z$ local steps after global synchronization. Therefore, we naturally consider the timestep $t_s(t)$ of

the latest synchronization round up to $t$, and observe that if the random variable $X := x_{i,0}^t - \overline{x}_{t_s}$, then $\mathbb{E}_k[X] = \overline{x}_t - \overline{x}_{t_s}$. Noting that the variance of $X$ is no greater than its second moment, we proceed as follows via telescoping:

$$
\begin{aligned}
\mathbb{E}_k[\|x_{i,0}^t - \overline{x}_t\|^2] = \sum_{i=1}^N p_i \|x_{i,0}^t - \overline{x}_t\|^2 &= \mathbb{E}_k[\|X - \mathbb{E}_k[X]\|^2] \\
&\leq \mathbb{E}_k[\|X\|^2] = \sum_{i=1}^N p_i \|x_{i,0}^t - \overline{x}_{t_s}\|^2 \\
&= \sum_{i=1}^N p_i \left\| x_{i,0}^t + \sum_{\tilde{t}=t_s+1}^{t-1} (-x_{\tilde{t}}^k + x_{\tilde{t}}^k) - \overline{x}_{t_s} \right\|^2 \\
&\leq \sum_{i=1}^N p_i (t - t_s - 1)^2 \max_{t' \in [t_s, t]} \eta_{t'}^2 \|Clip(c_t', \nabla F_i(x_{i,0}^t, \xi_{i,0}^t))\|^2 \\
&\leq \sum_{i=1}^N p_i z^2 \eta_{t_s}^2 c_t^2 = z^2 \eta_{t_s}^2 c_t^2 \leq z^2 u^2 \eta_t^2 c_t^2.
\end{aligned}
\tag{6}
$$

The final inequality was obtained by noting that $\eta_t \to 0^+$ monotonically from above and that $c_t \geq c_{t-1}$. The above holds for all $t \in \mathbb{Z}_{\geq 0}$, as if $t$ is a synchronization step, $\mathbb{E}_k\|x_{i,0}^t - \overline{x}_t\|^2 = 0$. The final inequality used that the monotonic near-harmonic decay of $\eta_t$ allows $\eta_{t_s} \leq u\eta_t$ for $u = (z+1)/2$. Finally, by Cauchy-Schwartz,

$$
\left( \sum_{i=1}^N p_i \|\overline{x}_t - x_{i,0}^t\| \right)^2 \leq \left( \sum_{i=1}^N p_i \right) \left( \sum_{i=1}^N p_i \|\overline{x}_t - x_{i,0}^t\|^2 \right),
$$

from which we conclude

$$
A_1 \leq \mu a \eta_t \|\overline{x}_t - x^*\|^2 + \frac{\eta_t}{\mu a} ((M^\alpha + B^\alpha) c_t^{1-\alpha} + \eta_t c_t z u L)^2
\tag{7}
$$

It now remains to bound $A_3$, which can be done straightforwardly via Jensen:

$$
A_3 = \eta_t^2 \|g_t\|^2 \leq \eta_t^2 \sum_{i=1}^N p_i \left\| Clip(c_t, \nabla F_i(x_{i,0}^t, \xi_{i,0}^t)) \right\|^2 \leq \eta_t^2 c_t^2.
$$

Collecting all inequalities gathered thus far gives the simple form

$$
\mathbb{E}_t[\|\overline{x}_{t+1} - x^*\|^2] \leq (1 - (1-a)\mu\eta_t) \|\overline{x}_t - x^*\|^2 - 2\eta_t (F(\overline{x}_t) - F(x^*)) + \eta_t^2 c_t^2 + \frac{\eta_t}{\mu a}((M^\alpha + B^\alpha) c_t^{1-\alpha} + \eta_t c_t z u L)^2,
$$

which under tower law of expectations is amenable to telescoping. Intuitively, we want to control the learning rate and form a quadratically decaying average on the LHS, which by Jensen and convexity will give a desired near-optimal point. The rest is a matter of carefully easing out a rate schedule that enables averaging, which also converges. Rearranging gives

$$
\begin{aligned}
\mathbb{E}[F(\overline{x}_t)] - F(x^*) \leq & \frac{(\eta_t^{-1} - (1-a)\mu)}{2} \mathbb{E}[\|\overline{x}_t - x^*\|^2] - \frac{1}{2\eta_t} \mathbb{E}[\|\overline{x}_{t+1} - x^*\|^2] + \frac{\eta_t c_t^2}{2} \\
& + \frac{1}{2\mu a}((M^\alpha + B^\alpha)^2 c_t^{2-2\alpha} + 2(M^\alpha + B^\alpha) c_t^{2-\alpha} \eta_t z u L + \eta_t^2 c_t^2 z^2 u^2 L^2).
\end{aligned}
\tag{8}
$$

Letting $\eta_t = r/(t+1)$, $a = 1 - 2/(r\mu)$ for $r > 2/\mu$, we have

$$
\begin{aligned}
t\mathbb{E}[F(\overline{x}_t)] - tF(x^*) \leq & \frac{t(t-1)}{2} \mathbb{E}[\|\overline{x}_t - x^*\|^2] - \frac{(t+1)t}{2} \mathbb{E}[\|\overline{x}_{t+1} - x^*\|^2] + \frac{t\eta_t c_t^2}{2} \\
& + \frac{t}{2\mu a}((M^\alpha + B^\alpha)^2 c_t^{2-2\alpha} + 2(M^\alpha + B^\alpha) c_t^{2-\alpha} \eta_t z u L + \eta_t^2 c_t^2 z^2 u^2 L^2)
\end{aligned}
\tag{9}
$$

Setting $c_t = ct^\gamma$ for $1/2 > \gamma > 0$, $c > 0$ gives after telescoping

$$\frac{\sum_{t=1}^{T} t\mathbb{E}[F(\overline{x}_t)]}{T(T+1)} - F(x^*) \leq \frac{rc^2 \sum_{t=1}^{T} t^{2\gamma}}{2T(T+1)} + \frac{(M^\alpha + B^\alpha)^2 c^{2-2\alpha} \sum_{t=1}^{T} t^{(2-2\alpha)\gamma}}{2(\mu - 2/r)T(T+1)}$$
$$+ \frac{c^{2-\alpha}rzu(M^\alpha + B^\alpha)L \sum_{t=1}^{T} t^{(2-\alpha)\gamma}}{(\mu - 2/r)T(T+1)} + \frac{r^2 c^2 z^2 u^2 L^2 \sum_{t=1}^{T} t^{2\gamma-1}}{2(\mu - 2/r)T(T+1)}.$$

Standard integral bounds give

$$\frac{\sum_{t=1}^{T} t\mathbb{E}[F(\overline{x}_t)]}{T(T+1)} - F(x^*) \leq \frac{rc^2 T^{2\gamma+1}}{(4\gamma + 2)T(T+1)} + \frac{(M^\alpha + B^\alpha)^2 c^{2-2\alpha}(T^{(2-2\alpha)\gamma+1} + 1)}{2(\mu - 2/r)((2 - 2\alpha)\gamma + 1)T(T+1)}$$
$$+ \frac{c^{2-\alpha}rzu(M^\alpha + B^\alpha)LT^{(2-\alpha)\gamma+1}}{(\mu - 2/r)((2 - \alpha)\gamma + 1)T(T+1)} + \frac{r^2 c^2 z^2 u^2 L^2(T^{2\gamma} + 1)}{4\gamma(\mu - 2/r)T(T+1)}.$$

Finally, note that by Jensen and convexity, the left hand side is lower bounded by

$$0 \leq F(\tilde{x}_T) - F(x^*) \leq \frac{\sum_{t=1}^{T} t\mathbb{E}[F(\overline{x}_t)]}{T(T+1)} - F(x^*)$$

where $\tilde{x}_T := \sum_{t=1}^{T} t\mathbb{E}[\overline{x}_t]/T(T+1)$ is a quadratically decaying average. This concludes the proof. It is straightforward to extend to the case in which the learning rate is scheduled to decay in each outer optimizer synchronization step instead of at each local step, by letting $\eta_t = r/(\lceil t/z \rceil + 1)$ in equation equation 8. $\square$

The value of the tail-index parameter $\alpha$ has a significant impact on the convergence behavior. When $\alpha$ is close to 1, the convergence becomes substantially slower due to the heavy-tailed nature of the induced stochastic gradients and the increased variance they introduce. Conversely, when $\alpha$ approaches 2, the variance is more controlled, leading to faster convergence rates. Importantly, our results demonstrate that even in the presence of infinite variance (i.e., $\alpha < 2$), convergence can still be achieved, showcasing the robustness of the clipping approach under extreme heavy-tailed conditions.

The averages $\overline{x}_t$ are virtual constructs used for theoretical analysis of Algorithm 2, which are not accumulated during the execution phase. That is, these quantities are only available at the outer optimizer synchronization steps, $t \in z \cdot \mathbb{Z}_{\geq 0}$, and are not collected otherwise (as models are not saved for every local timestep prior to synchronization). As a result, the application of Avg-$L_2 Clip$ creates a virtual history on the compute node models, where the aggregation of ephemeral model weights can theoretically induce convergence. However, in practice, this conflicts with the use of local epochs for communication efficiency, necessitating adjustments to the convergence theorem. This leads to the development of Corollary 3.

**Corollary 3.** *Let the conditions of Theorem 3 hold. Then, we have that*

$$\mathbb{E}\left[F\left(\frac{\sum_{t \in Z}(t-1)\overline{x}_t}{\sum_{t \in Z}(t-1)}\right)\right] - F(x^*) \leq \frac{(T+1)z}{(T-z)}(\psi_1 + \psi_2 + \psi_3 + \psi_4),$$

*where the $\psi_i$ are defined as in the statement of Theorem 3 and $Z$ is the set of all outer optimizer synchronization steps.*

*Proof.* We may start with equation equation 9, where we use the same notation as the proof of Theorem 3. Recall that $0 \leq F(x) - F(x^*)$ for all $x$. Therefore, we have for $Z = \{1, z + 1, \ldots, z\lfloor T/z \rfloor + 1\}$ for $T \notin z \cdot \mathbb{Z}$ and $Z = \{1, z + 1, \ldots, z(\lfloor T/z \rfloor - 1) + 1\}$ otherwise,

$$\sum_{t \in Z} t\left(\mathbb{E}[F(\overline{x}_t)] - F(x^*)\right) \leq \sum_{t \in [T]}\left(\frac{t(t-1)}{2}\mathbb{E}[\|\overline{x}_t - x^*\|^2] - \frac{(t+1)t}{2}\mathbb{E}[\|\overline{x}_{t+1} - x^*\|^2]\right)$$
$$+ \sum_{t \in [T]} \frac{t\eta_t c_t^2}{2} + \sum_{t \in [T]} \frac{t}{2\mu a}\left((M^\alpha + B^\alpha)^2 c_t^{2-2\alpha} + 2(M^\alpha + B^\alpha)c_t^{2-\alpha}\eta_t zuL + \eta_t^2 c_t^2 z^2 u^2 L^2\right).$$

Noting that

$$\sum_{t \in Z}(t-1)\left(\mathbb{E}[F(\overline{x}_t)] - F(x^*)\right) \leq \sum_{t \in Z} t\left(\mathbb{E}[F(\overline{x}_t)] - F(x^*)\right),$$

$$\frac{(T-z)T}{2z} \leq \frac{z(\lceil T/z \rceil - 1)\lceil T/z \rceil}{2} \leq \frac{z(\lfloor T/z \rfloor + 1)\lfloor T/z \rfloor}{2},$$

we obtain

$$\mathbb{E}\left[ F\left( \frac{\sum_{t \in Z}(t-1)\overline{x}_t}{\sum_{t \in Z}(t-1)} \right) \right] - F(x^*) \leq \frac{(T+1)z}{(T-z)}(\psi_1 + \psi_2 + \psi_3 + \psi_4).$$

As before, extension to the case where the learning rate decays at each outer optimizer synchronization step is straightforward. Therefore, the asymptotic convergence rate is identical that give in Theorem 3. □

In particular, we immediately deduce the following corollary.

**Corollary 4.** *Let the conditions of Theorem 3 hold. Then, Avg-$L_2Clip$ converges under heavy-tailed noise with rate $\mathcal{O}(T^{-1/2})$. That is, the algorithm recovers a point $\widetilde{x}_T$ which is materialized during training such that*

$$\mathbb{E}[F(\widetilde{x}_T)] - F(x^*) \lesssim \mathcal{O}(T^{-1/2}).$$

*Proof.* The maximal rate of convergence is immediately attained in the limit $\gamma \to 0^+$, where the dominating terms are $\Psi_i$ for $i = 1, 2, 3$. □

### D.2    DYNAMICS OF AVG-$L_2Clip$ UNDER FAILING COMPUTE NODES

Distributed optimization operates in two primary modes: full participation or partial participation (known in some fields such as federated learning as cross-silo or cross-device). Full participation distributed optimization is relevant for scenarios such as training language models in datacenters or healthcare models across hospitals Liu et al. (2024a); Douillard et al. (2024); Huang et al. (2019); Silva et al. (2019), where bypassing legislative geolocation restrictions enables access to larger datasets and promotes fairer, balanced model training Zhang et al. (2022). In contrast, partial participation involves training small-scale, personalized models on restricted compute nodes such as mobile devices Li et al. (2020a). In such settings, local data shards are often highly heterogeneous and non-IID, leading to diverse gradient distributions induced by the distributed outer global model weights synthesized by the outer optimizer. Consequently, it is crucial to conduct a theoretical performance analysis of Avg-$L_2Clip$ within environments to accommodate the presence of failing compute nodes or partial participation.

In this setting, line 2 of Avg-$L_2Clip$ is modified to sample a subset of participating nodes, $\mathcal{S} \subset [N]$, rather than selecting $S = [N]$. Additionally, normalized averaging is performed across only the participating compute nodes in line 7. Typically, extending the analysis from full to partial participation introduces additional complexities due to the randomness of node subsampling and the fact that most compute nodes remain idle. However, we can leverage elements of our previous analysis by considering a highly resource-inefficient algorithm that mimics full participation Avg-$L_2Clip$, in which all compute nodes remain active. We refer to this algorithm as $SludgeClip$ to emphasize its impracticality, despite being functionally equivalent to Avg-$L_2Clip$. By analyzing $SludgeClip$, we are able to establish convergence of Avg-$L_2Clip$ in when several datacenters or compute nodes fail to partake in training.

**Theorem 4.** *Let the clipping threshold in $SludgeClip$ (Algorithm 3) satisfy $c_t = c\eta_t^\gamma$ for $c > 0$ and $1/2 > \gamma > 0$. Decay the learning rate with schedule $\eta_t = r/(t+1)$ for $r > 2/\mu$. If the sampling scheme preserves the global objective[2], that is,*

$$\mathbb{E}_S\left[ \sum_{i \in [S]} p_i F_i(x) \right] = \sum_{i \in [N]} p_i F_i(x) = F(x),$$

*then we have for $Z$ the set of synchronization steps up to $T$ that*

$$\mathbb{E}\left[ F\left( \widetilde{x}_T' \right) \right] - F(x^*) := \mathbb{E}\left[ F\left( \frac{\sum_{t \in Z}(t-1)\overline{x}_t}{\sum_{t \in Z}(t-1)} \right) \right] - F(x^*) \leq z \cdot \mathcal{O}\left( t^{-\omega} \right),$$

---

[2]For example, $p_i = 1/N$ satisfies this condition. That is, given any selection of $p_i$ and $F_i(x)$, we may rescale the local objectives $F_i(x)$ such that $p_i = 1/N$ by controlling the influence of each local gradient update.

---

**Algorithm 3** $SludgeClip$

---

**Require:** Initial model $x_1$, learning rate schedule $\eta_t$, clipping schedule $c_t$
    Synchronization timestep $z \in \mathbb{Z}_{>0}$, projection domain $\mathcal{X}$
1: **for** $t = 1, \ldots, T$ **do**
2:     Sample participating compute nodes $S \subset [N]$ according to $p_i$
3:     **for** each node $i \in [N]$ **do**
4:         Draw minibatch gradient $g_{i,0}^t = \nabla F_i(x_{i,0}^t, \xi_{i,0}^t)$
5:         $x_{t+1}^k \leftarrow x_t^k - \eta_t \cdot L_2 Clip(c_t, g_{i,0}^t)$
6:     **end for**
7:     **if** $t - 1 \in z \cdot \mathbb{Z}_{\geq 0}$ :
8:         $x_{t+1}^k \leftarrow \mathrm{Proj}_{\mathcal{X}}\left((\sum_{i' \in S} p_{i'})^{-1} \sum_{i' \in S} p_{i'} x_{t+1}^{i'}\right)$, for $\forall k \in [N]$
9: **end for**

---

*where now $\omega$ satisfies*

$$\omega = \min\{1 - 2\gamma, 1 - (2 - 2\alpha)\gamma, 1 - (2 - \alpha)\gamma, 2 - 2\gamma, 2\gamma(\alpha - 1)\}.$$

*If the subsampling scheme fails to preserve the global objective (e.g., by sampling only a strict subset of avaliable nodes repeatedly), then Algorithm 3 asymptotes toward biased minimizer points within an increasing region determined by the clipping threshold $\mathbb{E}\left[F\left(\tilde{x}_T'\right)\right] - F(x^*) \lesssim \mathcal{O}(t^{2\gamma})$.*

We note that convergence is not clearly guaranteed when subsampling procedures violate the global objective in expectation. Specifically, we evaluate the algorithm's output relative to $x^*$, the global optimum of the true objective $F(x)$. However, when subsampling alters the objective, the algorithm no longer optimizes for $F(x)$, thereby clearly undermining convergence toward $x^*$. We then measure the propensity of the algorithm output to $x^*$, the global optimum of the true objective $F(x)$ which is no longer the objective of the subsampled algorithm.

*Proof.* We first analyze the case in which the subsampling strategy preserves the correct global objective, which allows for convergence to $x^*$. Recall that $SludgeClip$-SGD was constructed to allow the analysis for non-synchronization steps to be analogous to full-participation Avg-$L_2Clip$. Therefore, we focus on outer optimizer synchronization steps while incorporating the elements of the previous analysis for Theorem 3. We now use the following notation for subsampled averages of participating compute node devices:

$$\tilde{x}_t = \frac{\sum_{i \in S} p_i x_{i,0}^t}{\sum_{i \in S} p_i}, \quad \tilde{g}_t = \frac{\sum_{i \in S} p_i \cdot Clip(c_t, \nabla F_i\left(x_{i,0}^t, \xi_{i,0}^t\right))}{\sum_{i \in S} p_i}.$$

For added clarity, we denote $g_t$ as $\overline{g}_t$ to indicate that normalized averages are taken over all inner compute nodes, and not solely participating nodes as in $\tilde{g}_t$. Then for $t + 1$ a synchronization step, we have that

$$\|\tilde{x}_{t+1} - x^*\|^2 \leq \|\tilde{x}_t - x^* - \eta_t \tilde{g}_t\|^2 = \|\overline{x}_t + (\tilde{x}_t - \overline{x}_t) - x^* - \eta_t \tilde{g}_t + (\eta_t \overline{g}_t - \eta_t \overline{g}_t)\|^2$$

$$= \|\overline{x}_t - x^*\|^2 + 2\langle \overline{x}_t - x^*, \underbrace{\tilde{x}_t - \overline{x}_t - \eta_t \tilde{g}_t + (\eta_t \overline{g}_t - \eta_t \overline{g}_t)}_{B_1}\rangle + B_1^2$$

$$\leq \|\overline{x}_t - x^*\|^2 \underbrace{-2\eta_t \langle \overline{x}_t - x^*, \overline{g}_t - \nabla F(\overline{x}_t)\rangle}_{A_1} \underbrace{-2\eta_t \langle \overline{x}_t - x^*, \nabla F(\overline{x}_t)\rangle}_{A_2}$$

$$\underbrace{+2\langle \overline{x}_t - x^*, \tilde{x}_t - \overline{x}_t\rangle}_{B_2} \underbrace{+2\eta_t\langle \overline{x}_t - x^*, \overline{g}_t - \tilde{g}_t\rangle}_{B_3} + \underbrace{\|\tilde{x}_t - \overline{x}_t - \eta_t \tilde{g}_t\|^2}_{B_4}.$$

In this form, the $A_i$ terms are therefore shared with the previous analysis, and $A_2$ may be bounded by $\mu$-strong convexity as before. This gives that

$$A_2 \leq -\mu\eta_t\|\overline{x}_t - x^*\|^2 - 2\eta_t\left(F(\overline{x}_t) - F(x^*)\right).$$

$A_1$ is once again bounded under conditional expectations $\mathbb{E}_t[\cdot]$ by equation equation 7, though with a different value of $a' > 0$ than in the previous proof,

$$A_1 \leq \mu a' \eta_t\|\overline{x}_t - x^*\|^2 + \frac{\eta_t}{\mu a'}((M^\alpha + B^\alpha)c_t^{1-\alpha} + \eta_t c_t z u L)^2. \tag{7}$$

Now, as $B_2$ is eliminated under expectations under subsampling, we focus on the remaining terms. It is clear that we must bound and $\|\bar{g}_t - \tilde{g}_t\|$ to proceed. Intuitively, this is controlled by normalized averages and model drift across participating nodes. Therefore, we consider the nearest or most recent synchronization timestep $t_s(t)$ as before and rearrange to incorporate elements of our previous analysis. Assuming interchangeability between the integrals $\mathbb{E}_S$ (integrating over the randomness of node subsampling) and $\mathbb{E}_t$ (integrating over randomness of $\xi_{i,0}^t$),

$$\|\mathbb{E}_t\left[\mathbb{E}_S[\tilde{g}_t] - \bar{g}_t\right]\| = \left\|\mathbb{E}_t\left[\mathbb{E}_S\left[\sum_{i \in S}\frac{p_i}{\sum_{i' \in S}p_{i'}}(Clip(c_t, \nabla F_i\left(x_{i,0}^t, \xi_{i,0}^t\right)) - \nabla F_i(\bar{x}_t))\right] - (\bar{g}_t - \nabla F(\bar{x}_t))\right]\right\|$$

$$= \left\|\mathbb{E}_S\left[\mathbb{E}_t\left[\sum_{i \in S}\frac{p_i}{\sum_{i' \in S}p_{i'}}(Clip(c_t, \nabla F_i\left(x_{i,0}^t, \xi_{i,0}^t\right)) - \nabla F_i(\bar{x}_t, \xi_{i,0}^t))\right]\right] - \mathbb{E}_t\left[\bar{g}_t - \nabla F(\bar{x}_t)\right]\right\|$$

$$\leq \mathbb{E}_S\left[\sum_{i \in S}\frac{p_i}{\sum_{i' \in S}p_{i'}}\mathbb{E}_t[\|Clip(c_t, \nabla F_i\left(x_{i,0}^t, \xi_{i,0}^t\right)) - \nabla F_i(\bar{x}_t, \xi_{i,0}^t)\|]\right] + \mathbb{E}_t[\|\bar{g}_t - \nabla F(\bar{x}_t)\|] \leq 2(M^\alpha + B^\alpha)c_t^{1-\alpha}$$

where to obtain the final line we used Jensen and an analogous reasoning as in equation equation 5.

Therefore, we have for $b > 0$ that

$$B_3 \leq b\eta_t\|\bar{x}_t - x^*\|^2 + 4\eta_t(M^\alpha + B^\alpha)^2 c_t^{2(1-\alpha)}.$$

It now remains to bound $B_4$, which can be done straightforwardly:

$$B_4 \leq 2\|\tilde{x}_t - \bar{x}_t\|^2 + 2\eta_t^2\|\tilde{g}_t\|^2 \leq 4z^2u^2\eta_t^2 c_t^2 + 2\eta_t^2 c_t^2.$$

Collecting all inequalities gathered under the tower law of expectation, we have

$$\mathbb{E}[\|\tilde{x}_{t+1} - x^*\|^2] \leq (1 - ((1-a)\mu + b)\eta_t)\mathbb{E}[\|\bar{x}_t - x^*\|^2] - 2\eta_t\mathbb{E}\left[F(\bar{x}_t) - F(x^*)\right]$$
$$+ \frac{\eta_t}{\mu a}((M^\alpha + B^\alpha)c_t^{1-\alpha} + \eta_t c_t zuL)^2 + 4z^2u^2\eta_t^2 c_t^2 + 2\eta_t^2 c_t^2 + 4\eta_t(M^\alpha + B^\alpha)^2 c_t^{2(1-\alpha)}.$$

Recall the learning rate schedule $\eta_t = r/(t+1)$, while setting $a', b$ such that $r((1-a')\mu + b) = 2$. Then, we have for $Z$ the set of all synchronization steps,

$$\sum_{t+1 \in Z} t(\mathbb{E}[F(\bar{x}_t)] - F(x^*)) \leq \sum_{t+1 \in Z}\left[\frac{t(t-1)}{2}\mathbb{E}[\|\bar{x}_t - x^*\|^2] - \frac{(t+1)t}{2}\mathbb{E}[\|\tilde{x}_{t+1} - x^*\|^2]\right]$$
$$+ \underbrace{\sum_{t+1 \in Z} 2(M^\alpha + B^\alpha)^2 t c_t^{2(1-\alpha)}}_{B_5} + \underbrace{\sum_{t+1 \in Z}\frac{1}{2\mu a}((M^\alpha + B^\alpha)c_t^{1-\alpha} + \eta_t c_t zuL)^2}_{\sim \Psi_2 + \Psi_3 + \Psi_4} + \underbrace{\sum_{t+1 \in Z} t\eta_t c_t^2(2z^2u^2 + 1)}_{\sim \Psi_1}.$$

For $t+1 \notin Z$, we use the standard telescoping sum in equation equation 9 while noting that $\tilde{x}_{t+1} = \bar{x}_{t+1}$ due to the synchronization step. We do not repeat mechanical calculation steps here to not obscure the intuitions behind the proof, and instead indicate asymptotically equivalent terms to $\Psi_i$ under $1/(T^2 + T)$ averaging on the right hand side. It remains to bound the residual term $B_5$ under the averaging step, which gives

$$\frac{B_5}{T(T+1)} \lesssim \mathcal{O}(t^{2\gamma(1-\alpha)}),$$

which concludes the proof for the first case.

In the setting in which the subsampling procedure fails to preserve the global objective, we bound $\|\tilde{x}_t - \bar{x}_t\|$ as follows:

$$\|\tilde{x}_t - \bar{x}_t\| = \left\|\sum_{i \in [S]}\left(\frac{\sum_{\bar{k} \notin [S]}p_{\bar{k}}}{\sum_{i' \in [S]}p_{i'}}\right)p_i x_{i,0}^t - \sum_{i \notin [S]}p_i x_{i,0}^t\right\|$$

$$\leq \sum_{i \in [S]}\left(\frac{\sum_{\bar{k} \notin [S]}p_{\bar{k}}}{\sum_{i' \in [S]}p_{i'}}\right)p_i\|x_{i,0}^t - \bar{x}_{t_s}\| + \sum_{i \notin [S]}p_i\|x_{i,0}^t - \bar{x}_{t_s}\| \leq 2zu\eta_t c_t,$$

due to triangle inequality and Jensen. That is, by the synchronization step, we have $x_{t_s}^k = \overline{x}_{t_s}$, $\forall k \in [N]$ via to full available node activation in $SludgeClip$. This gives

$$\|x_{i,0}^t - \overline{x}_{t_s}\| = \left\| x_{i,0}^t + \sum_{t'=t_s+1}^{t-1} (-x_{t'}^k + x_{t'}^k) - \overline{x}_{t_s} \right\| \leq \sum_{t'=t_s+1}^{t-1} \|x_{t'}^k - x_{t'-1}^k\| \leq z u \eta_t c_t$$

as in equation equation 6. Similarly, we have by Jensen and convexity of the norm that

$$\|\tilde{g}_t - \overline{g}_t\| \leq 2c_t.$$

Therefore, we obtain for $b_1, b_2 > 0$

$$B_2 \leq b_1 \eta_t \|\overline{x}_t - x^*\|^2 + \frac{1}{b_1 \eta_t} \|\tilde{x}_t - \overline{x}_t\|^2 \leq b_1 \eta_t \|\overline{x}_t - x^*\|^2 + \frac{2z^2 u^2 c_t^2 \eta_t}{b_1},$$

$$B_3 \leq b_2 \eta_t \|\overline{x}_t - x^*\|^2 + 4\eta_t c_t^2.$$

Following analogous calculations as in the case where the subsampling does not violate the global objective, we arrive at a new residual term

$$\frac{B_6}{T(T+1)} \lesssim \mathcal{O}(t^{2\gamma}),$$

which controls the expansion of the bias due to the incorrect sampling strategy. □

## D.3 Convergence of $Bi^2Clip$

In this section, we analyze the convergence of $Bi^2Clip$ under heavy-tailed noise. By employing $BiClip$ at both the inner and outer optimizers, $Bi^2Clip$ can represent a highly competitive algorithm realized by TailOPT that utilizes adaptive mimicry, aiming to adjust to gradient distributional statistics while strictly maintaining resource efficiency. Unlike $Adam^2$, which applies Adam at both the inner and outer optimizers, $Bi^2Clip$ achieves comparable empirical performance while requiring no additional memory or computational overhead beyond standard SGD (Table 1). This highlights its efficiency and practicality, particularly in resource-constrained settings. We begin with the pseudocode for $Bi^2Clip$, Algorithm 4.

---

**Algorithm 4** $Bi^2Clip$

---

**Require:** Initial model $x_1$, learning rate schedule $\eta_t$, clipping schedules $u_t, d_t, \widetilde{u}_t, \widetilde{d}_t$
     Synchronization timestep $z \in \mathbb{Z}_{>0}$
1: **for** $t = 1, \ldots, T$ **do**
2:     **for** each node $i \in [N]$ in parallel **do**
3:         $x_{i,0}^t \leftarrow x_t$
4:         **for** each local step $k \in [z]$ **do**
5:             Draw minibatch gradient $g_{i,k}^t = \nabla F_i(x_{i,k}^t, \xi_{i,k}^t)$
6:             $x_{i,k}^{t+1} \leftarrow x_{i,k}^t - \eta_t \cdot BiClip(u_t, d_t, g_{i,k}^t)$
7:         **end for**
8:     **end for**
9:     $\Delta_t = \frac{1}{N} \sum_{i \in [N]} \left( x_{i,z}^t - x_{t-1} \right), \quad \widetilde{m}_t \leftarrow \Delta_t$
10:     $x_t = x_{t-1} + \eta BiClip(\widetilde{u}_t, \widetilde{d}_t, \widetilde{m}_t)$
11: **end for**

---

**Bounded domain.** We carry out the analysis over a sufficiently large, compact domain $\mathcal{X}$. Let $\nabla F(x)$ be the deterministic gradient, obtained by integrating over $\nabla F(x, \xi)$, the stochastic gradient with a heavy-tailed distribution. The existence of $\nabla F(x)$ implies $F(x)$ is continuous, which gives boundedness via the extremal value theorem. Therefore, from now onward, we formally assume $\nabla F(x)$ is coordinatewise bounded by $G$ in absolute value. We have the following theorem.

**Theorem 5.** *Let assumptions 1-2 hold, and the learning rate and clipping schedules satisfy $\eta_t = \Theta(t^\omega)$, $\eta_\ell^t = \Theta(t^\nu)$, $d_t = \Theta(t^\gamma)$, $u_t = \Theta(t^\zeta)$, $\widetilde{d}_t = \Theta(t^{\widetilde{\gamma}})$, and $u_t = \Theta(t^{\widetilde{\zeta}})$. Imposing $\zeta, \widetilde{\zeta} > 0 > \gamma, \widetilde{\gamma}$, for $\omega, \nu \leq 0$, as well as the following conditions*

$$-1 < \omega + \nu, \quad \nu + \zeta < 0, \quad \max\{\omega + 2\widetilde{\zeta}, \widetilde{\gamma}\} < \nu,$$

for $Bi^2Clip$ (Algorithm 4), we have that

$$\min_{t\in[T]} \mathbb{E}[\|\nabla F(x_{t-1})\|^2] \lesssim \Psi_1 + \Psi_2 + \Psi_3 + \Psi_4 + \Psi_5 + \Psi_6 + \Psi_7,$$

where the $\Psi_i$ are given

$$\Psi_1 = \mathcal{O}\left(T^{-\omega-\nu-1}\right), \quad \Psi_2 = \mathcal{O}\left(T^{\omega+2\widetilde{\zeta}-\nu}\right), \quad \Psi_3 = \mathcal{O}\left(T^{\widetilde{\gamma}-\nu}\right), \quad \Psi_4 = \mathcal{O}\left(T^{\gamma}\right),$$

$$\Psi_5 = \mathcal{O}\left(T^{(\alpha-1)\nu+(1-\alpha)\widetilde{\zeta}}\right), \quad \Psi_6 = \mathcal{O}\left(T^{(1-\alpha)\zeta}\right), \quad \Psi_7 = \mathcal{O}\left(T^{\nu+\zeta}\right).$$

*Proof.* We provide the proof for $L_2$-wise $BiClip(\cdot)$ for illustrative purposes and notational convenience. The extension to coordinate-wise $BiClip(\cdot)$ is straightforward as described in the comments following the proof of Theorem 6, Remark 2. For completeness and readability, we formally provide the definition of $L_2$-wise $BiClip(\cdot)$ as

$$BiClip(u_t, d_t, x) = x \cdot \frac{d_t}{\|x\|} \chi\left(\|x\| \le d_t\right)$$

$$+ x \cdot \frac{u_t}{\|x\|} \chi\left(\|x\| \ge u_t\right) + x \cdot \chi\left(d_t < \|x\| < u_t\right).$$

Here, $\chi$ is the indicator function, and $u_t \ge d_t \ge 0$ are the clipping thresholds. By default, we take $a/0 := 0$ for $\forall a \in \mathbb{R}$. Now, we begin by noting that due to $L$-smoothness, we have where $\mathbb{E}_t[\,\cdot\,]$ takes expectation up to $x_{t-1}$ that

$$\mathbb{E}_t[F(x_t)] - F(x_{t-1}) \le \langle \nabla F(x_{t-1}), \mathbb{E}_t[x_t - x_{t-1}]\rangle + \frac{L}{2}\mathbb{E}_t[\|x_t - x_{t-1}\|^2]$$

$$\le \eta_t \underbrace{\left\langle \nabla F(x_{t-1}), -\mathbb{E}_t[BiClip(\widetilde{u}_t, \widetilde{d}_t, -\Delta_t)]\right\rangle}_{A_1} + \frac{L\eta_t^2}{2}\mathbb{E}_t\left[\left\|BiClip(\widetilde{u}_t, \widetilde{d}_t, \Delta_t)\right\|^2\right].$$

Now, we expand to obtain the following form

$$A_1 = -\left\langle \nabla F(x_{t-1}), \mathbb{E}_t[BiClip(\widetilde{u}_t, \widetilde{d}_t, -\Delta_t) \pm \Delta_t] \mp \eta_\ell^t \sum_{i\in[N]}\sum_{\nu\in[K]-1} p_i\mathbb{E}_t[\nabla F_i(x_{i,\nu}^t)] \mp K\eta_\ell^t\nabla F(x_{t-1})\right\rangle$$

$$= -\underbrace{\left\langle \nabla F(x_{t-1}), \mathbb{E}_t[BiClip(\widetilde{u}_t, \widetilde{d}_t, -\Delta_t) + \Delta_t]\right\rangle}_{B_1} - \underbrace{\left\langle \nabla F(x_{t-1}), -\eta_\ell^t \sum_{i\in[N]}\sum_{\nu\in[K]-1} p_i\mathbb{E}_t[\nabla F_i(x_{i,\nu}^t)] - \mathbb{E}_t[\Delta_t]\right\rangle}_{B_2}$$

$$- \underbrace{\left\langle \nabla F(x_{t-1}), \eta_\ell^t \sum_{i\in[N]}\sum_{\nu\in[K]-1} p_i\mathbb{E}_t[\nabla F_i(x_{i,\nu}^t)] - K\eta_\ell^t\nabla F(x_{t-1})\right\rangle}_{B_3} - K\eta_\ell^t\|\nabla F(x_{t-1})\|^2.$$

Using the convexity of compositions (via $\alpha \ge 1$) and Jensen, we deduce

$$\mathbb{E}_t[\|\Delta_t\|^\alpha] = \mathbb{E}_t[\|\eta_\ell^t \sum_{i\in[N]}\sum_{\nu\in[K]-1} p_i \cdot BiClip(u_t, d_t, \nabla F_i(x_{i,\nu}^t, \xi_{i,\nu}^t))\|^\alpha]$$

$$\le (\eta_\ell^t)^\alpha K^\alpha \mathbb{E}_t\left[\left\|\frac{1}{K}\cdot\sum_{i\in[N]}\sum_{\nu\in[K]-1} p_i \cdot BiClip(u_t, d_t, \nabla F_i(x_{i,\nu}^t, \xi_{i,\nu}^t))\right\|^\alpha\right]$$

$$\le (\eta_\ell^t)^\alpha K^{\alpha-1} \sum_{i\in[N]}\sum_{\nu\in[K]-1} p_i\mathbb{E}_t[\|BiClip(u_t, d_t, \nabla F_i(x_{i,\nu}^t, \xi_{i,\nu}^t))\|^\alpha]$$

$$\le (\eta_\ell^t)^\alpha K^{\alpha-1} \sum_{i\in[N]}\sum_{\nu\in[K]-1} p_i(d_t^\alpha + \mathbb{E}_t[\|\nabla F_i(x_{i,\nu}^t, \xi_{i,\nu}^t)\|^\alpha])$$

$$\le (\eta_\ell^t)^\alpha K^{\alpha-1} \sum_{i\in[N]}\sum_{\nu\in[K]-1} p_i d_t^\alpha + (\eta_\ell^t)^\alpha K^{\alpha-1} \underbrace{\sum_{i\in[N]}\sum_{\nu\in[K]-1} p_i\mathbb{E}_t[\|\nabla F_i(x_{i,\nu}^t, \xi_{i,\nu}^t)\|^\alpha]}_{C}.$$

Note that the term $C$ can be bounded as

$$
\begin{aligned}
C &\le (\eta_\ell^t)^\alpha K^{\alpha-1} \sum_{i\in[N]} \sum_{\nu\in[K]-1} p_i 2^\alpha \mathbb{E}_t\left[\frac{\|\nabla F_i(x_{i,v}^t)\|^\alpha}{2} + \frac{\|\xi_{i,v}^t\|^\alpha}{2}\right] \\
&\le (\eta_\ell^t)^\alpha K^{\alpha-1} \sum_{i\in[N]} \sum_{\nu\in[K]-1} p_i 2^{\alpha-1}(M^\alpha + B^\alpha) = (\eta_\ell^t)^\alpha K^{\alpha-1} \sum_{\nu\in[K]-1} 2^{\alpha-1}(M^\alpha + B^\alpha),
\end{aligned}
$$

where $M := \max_{x\in\mathcal{X}, i\in[N]} \|\nabla F_i(x)\|$ and $B^\alpha := \max_{i\in[N],\ \nu\in[K]-1} \mathbb{E}_t[\|\xi_{i,v}^t\|^\alpha] \le \sup_{i\in[N]}(B_i)^{\alpha_i}$. We note that this results holds also under distribution shift for the stochastic noise $\xi_i^t$, where $t\in[T]$ and $i\in[N]$, as long as the $\alpha$-moment remains universally bounded. Therefore, we conclude

$$
\mathbb{E}_t[\|\Delta_t\|^\alpha] \le (\eta_\ell^t)^\alpha K^{\alpha-1} \sum_{\nu\in[K]-1} d_t^\alpha + (\eta_\ell^t)^\alpha K^{\alpha-1} 2^{\alpha-1} \sum_{\nu\in[K]-1}(M^\alpha + B^\alpha) =: (\eta_\ell^t)^\alpha \widetilde{M}.
$$

This gives by the Cauchy-Schwartz inequality that

$$
\begin{aligned}
B_1 &\le \|\nabla F(x_{t-1})\| \|\mathbb{E}_t[BiClip(\widetilde{u}_t, \widetilde{d}_t, -\Delta_t)] + \Delta_t\| \\
&\le G \cdot \mathbb{E}_t[\chi(\|\Delta_t\| \le \widetilde{d}_t)\, \widetilde{d}_t + \chi\,(\widetilde{u}_t \le \|\Delta_t\|)\, \|\Delta_t\|^\alpha \|\Delta_t\|^{1-\alpha}] \\
&\le G\left[\mathbb{P}(\|\Delta_t\| \le \widetilde{d}_t)\, \widetilde{d}_t + \mathbb{P}\,(\widetilde{u}_t \le \|\Delta_t\|)\,(\eta_\ell^t)^\alpha \widetilde{u}_t^{1-\alpha} \widetilde{M}\right].
\end{aligned}
$$

Now, $B_2$ may be bounded as follows:

$$
\begin{aligned}
B_2 &\le G \left\| \eta_\ell^t \sum_{i\in[N]} \sum_{\nu\in[K]-1} p_i \mathbb{E}_t[\nabla F_i(x_{i,\nu}^t)] + \mathbb{E}_t[\Delta_t] \right\| \\
&= G \left\| \mathbb{E}_t[\eta_\ell^t \sum_{i\in[N]} \sum_{\nu\in[K]-1} p_i \nabla F_i(x_{i,\nu}^t, \xi_{i,\nu}^t) + \Delta_t] \right\| \\
&\le G \mathbb{E}_t \left[ \left\| \eta_\ell^t \sum_{i\in[N]} \sum_{\nu\in[K]-1} p_i \nabla F_i(x_{i,\nu}^t, \xi_{i,\nu}^t) + \Delta_t \right\| \right],
\end{aligned}
$$

where we used convexity, Jensen, and that the stochastic gradient noise is unbiased. Unraveling the definition of the pseudogradient $\Delta_t$ gives

$$
\begin{aligned}
B_2 &\le G\eta_\ell^t \mathbb{E}_t \left[ \left\| \sum_{i\in[N]} \sum_{\nu\in[K]-1} p_i \nabla F_i(x_{i,\nu}^t, \xi_{i,\nu}^t) - \sum_{i\in[N]} \sum_{\nu\in[K]-1} p_i BiClip(u_t, d_t, \nabla F_i(x_{i,\nu}^t, \xi_{i,\nu}^t)) \right\| \right] \\
&\le G\eta_\ell^t \sum_{i\in[N]} \sum_{\nu\in[K]-1} p_i \mathbb{E}_t\left[ \left\| \nabla F_i(x_{i,\nu}^t, \xi_{i,\nu}^t) - BiClip(u_t, d_t, \nabla F_i(x_{i,\nu}^t, \xi_{i,\nu}^t)) \right\| \right] \\
&\le G\eta_\ell^t \sum_{i\in[N]} \sum_{\nu\in[K]-1} p_i \left[ d_t \mathbb{P}(\|\nabla F_i(x_{i,\nu}^t, \xi_{i,\nu}^t)\| \le d_t) + \mathbb{P}(\|\nabla F_i(x_{i,\nu}^t, \xi_{i,\nu}^t)\| \ge u_t) u_t^{1-\alpha} 2^{\alpha-1}(M^\alpha + B^\alpha) \right] \\
&\le G\eta_\ell^t \sum_{\nu\in[K]-1} \left[ d_t + u_t^{1-\alpha} 2^{\alpha-1}(M^\alpha + B^\alpha) \right].
\end{aligned}
$$

Additionally, $B_3$ may be bounded via $L$-smoothness and telescoping:

$$B_3 \le \eta_\ell^t G \left\| \sum_{i\in[N]} \sum_{\nu\in[K]-1} p_i \nabla F_i(x_{i,\nu}^t) - K\nabla F(x_{t-1}) \right\|$$

$$\le \eta_\ell^t G \left\| \sum_{i\in[N]} \sum_{\nu\in[K]-1} p_i \nabla F_i(x_{i,\nu}^t) - \sum_{i\in[N]} \sum_{\nu\in[K]-1} p_i \nabla F_i(x_{i,0}^t) \right\|$$

$$\le \eta_\ell^t G \sum_{i\in[N]} \sum_{\nu\in[K]-1} p_i L \|x_{i,\nu}^t - x_{i,0}^t\|$$

$$\le \eta_\ell^t G \sum_{i\in[N]} \sum_{\nu\in[K]-1} p_i L \left\| x_{i,\nu}^t + \sum_{r=1}^{v-1}(x_{i,r}^t - x_{i,r}^t) - x_{i,0}^t \right\|$$

$$\le \eta_\ell^t GL \sum_{i\in[N]} p_i \cdot \left( \sum_{\nu\in[K]-1} \sum_{r=1}^{v-1} \|x_{i,r}^t - x_{i,r-1}^t\| \right) \le \frac{(\eta_\ell^t)^2 GLK^2 u_t}{2}.$$

Collecting all inequalities gathered thus far, we have

$$\mathbb{E}_t[F(x_t)] - F(x_{t-1}) \le \frac{L\eta_t^2 \widetilde{u}_t^2}{2} - K\eta_\ell^t \eta_t \|\nabla F(x_{t-1})\|^2 + G\eta_t \widetilde{d}_t + G\eta_t(\eta_\ell^t)^\alpha \widetilde{u}_t^{1-\alpha} \widetilde{M}$$

$$+ G\eta_\ell^t \eta_t \sum_{\nu\in[K]-1} \left[ d_t + u_t^{1-\alpha} 2^{\alpha-1}(M^\alpha + B^\alpha) \right] + \frac{\eta_t(\eta_\ell^t)^2 GLK^2 u_t}{2}.$$

Telescoping under the law of iterated expectations gives

$$\sum_{t\in[T]} K\eta_\ell^t \eta_t \mathbb{E}[\|\nabla F(x_{t-1})\|^2] \le F(x_0) - \mathbb{E}[F(x_T)] + \sum_{t\in[T]} \left( \frac{L\eta_t^2 \widetilde{u}_t^2}{2} + G\eta_t \widetilde{d}_t + G\eta_t(\eta_\ell^t)^\alpha \widetilde{u}_t^{1-\alpha} \widetilde{M} \right)$$

$$+ G \sum_{t\in[T]} \eta_\ell^t \eta_t \sum_{\nu\in[K]-1} \left[ d_t + u_t^{1-\alpha} 2^{\alpha-1}(M^\alpha + B^\alpha) \right] + \sum_{t\in[T]} \frac{\eta_t(\eta_\ell^t)^2 GLK^2 u_t}{2}.$$

Now, we move to the asymptotic regime. Let $\eta_t = \Theta(t^\omega)$, $\eta_\ell^t = \Theta(t^\nu)$, $d_t = \Theta(t^\gamma)$, $u_t = \Theta(t^\zeta)$, $\widetilde{d}_t = \Theta(t^{\widetilde\gamma})$, and $u_t = \Theta(t^{\widetilde\zeta})$. This gives after routine calculations that

$$\min_{t\in[T]} \mathbb{E}[\|\nabla F(x_{t-1})\|^2] \lesssim \mathcal{O}\left( T^{-\omega-\nu-1} + T^{\omega+2\widetilde\zeta-\nu} + T^{\widetilde\gamma-\nu} + T^{(\alpha-1)\nu+(1-\alpha)\widetilde\zeta} + T^\gamma + T^{(1-\alpha)\zeta} + T^{\nu+\zeta} \right).$$

To attain convergence of the RHS, it is clear that we must impose $\zeta, \widetilde\zeta > 0 > \gamma, \widetilde\gamma$, for $\omega, \nu \le 0$. Additionally, we have further constrained

$$-1 < \omega + \nu, \quad \nu + \zeta < 0, \quad \max\{\omega + 2\widetilde\zeta, \widetilde\gamma\} < \nu,$$

which ensures that the LHS diverges at a scale faster than logarithmic, validating the asymptotic regime and concluding the proof. To obtain the rate of convergence, we may let for $\widetilde\varepsilon \in (0, 1/8)$,

$$\omega = -\frac{1}{2}, \quad \nu = -\frac{1}{4}, \quad \widetilde\zeta = \frac{1}{8} - \widetilde\varepsilon, \quad \widetilde\gamma = -\frac{1}{8} - \widetilde\varepsilon, \quad \zeta = \frac{\alpha(1-\alpha)}{4}.$$

This gives that $Bi^2Clip$ converges with maximal rate at least $\mathcal{O}(T^{-r})$, where for $\widetilde\varepsilon \in (0, 1/8)$ and $\alpha > 1$,

$$r := \min\left\{ \frac{(\alpha-1)\alpha}{4}, \widetilde\varepsilon, \frac{\alpha-1}{4} - (1-\alpha)(\frac{1}{8} - \widetilde\varepsilon) \right\}.$$

$\square$

**Remark 1.** *We note that setting $\widetilde{d}_t = 0$, $\widetilde{u}_t = \infty$, and $\eta_t = 1$ recovers the simple averaging operation that can be done at the outer optimizer as a special case of $Bi^2Clip$, procuring Avg-BiClip. Therefore, one perspective of viewing $Bi^2Clip$ may be the addition of computation and memory efficient adaptive mimicry into traditional SGD-Averaging distributed training frameworks, that aims to dynamically adjust to the gradient distributional geometry. Similarly, for specific hyperparameter choices, $Bi^2Clip$ collapses into $BiClip$-SGD, with upper and lower thresholding applied by the outer optimizers only to accumulated model updates from the inner compute nodes.*

Now, in the following subsections, we further analyze the convergence behavior of TailOPT under additional varying adaptive optimizer instantiations. The Adagrad instantiation (Algorithm 5) collects pseudogradients and sums their squares, effectively implementing a form of implicit clipping. However, it aggressively decays coordinate-wise learning rates, which can limit performance. To address this, we introduce RMSProp-$TailClip$ (Algorithm 6), which relaxes the preconditioning by employing an exponentially decaying moving average of the second moment. In both cases, we prove that the minimum expected gradient converges to $0$. Additionally, by incorporating a moving average of the first pseudogradient moment as a form of momentum, we derive Algorithm 7. For this variant, we show that the expected minimal gradient does not diverge even under restarting of the algorithm, which in practice translates to the update of any singular step not diverging in expectation. As in the main paper, $TailClip$ refers to either $BiClip$ or $L_2Clip$, and we provide our proofs for $BiClip$ for added generality over $L_2Clip$.

### D.4 CONVERGENCE OF ADAGRAD-$TailClip$

We begin by providing the pseudocode of Adagrad-$TailClip$ (Algorithm 5). Then, we have the following result.

---

**Algorithm 5** Adagrad-$TailClip$

---

**Require:** Initial model $x_1$, learning rate schedule $\eta_t$, clipping schedules $u_t$, $d_t$
 Synchronization timestep $z \in \mathbb{Z}_{>0}$, adaptivity parameter $\tau > 0$
 1: **for** $t = 1, \ldots, T$ **do**
 2:     **for** each node $i \in [N]$ in parallel **do**
 3:         $x_{i,0}^t \leftarrow x_t$
 4:         **for** each local step $k \in [z]$ **do**
 5:             Draw minibatch gradient $g_{i,k}^t = \nabla F_i(x_{i,k}^t, \xi_{i,k}^t)$
 6:             $x_{i,k}^{t+1} \leftarrow x_{i,k}^t - \eta_t \cdot TailClip(u_t, d_t, g_{i,k}^t)$
 7:         **end for**
 8:     **end for**
 9:     $\Delta_t = \frac{1}{N}\sum_{i\in[N]}\left(x_{i,z}^t - x_{t-1}\right), \quad \widetilde{m}_t \leftarrow \Delta_t$
10:     $\widetilde{v}_t = \widetilde{v}_{t-1} + \Delta_t^2$
11:     $x_t = x_{t-1} + \eta\frac{\widetilde{m}_t}{\sqrt{\widetilde{v}_t}+\tau}$
12: **end for**

---

**Theorem 6.** *Let the clipping and learning rate thresholds satisfy $\eta_t = \Theta(t^\omega)$, $\eta_\ell^t = \Theta(t^\nu)$, $d_t = \Theta(t^\gamma)$, and $u_t = \Theta(t^\zeta)$ for the conditions*

$$0 < \zeta < \min\left\{\frac{1}{4}, \omega + \frac{1}{2}\right\}, \quad -\frac{1}{2} < \omega \le 0, \quad \gamma < \min\left\{0, -\nu - \zeta - \frac{1}{2}\right\},$$

$$\nu < \min\left\{-\frac{1}{6} - \frac{4}{3}\zeta, -\frac{1}{4} - \frac{3}{2}\zeta - \frac{1}{2}\omega, -\frac{1}{2} + (\alpha - 2)\zeta\right\}.$$

*Then, we have that*

$$\min_{t\in[T]} \mathbb{E}\left\|\nabla F(x_t)\right\|^2 \le \Psi_1 + \Psi_2 + \Psi_3 + \Psi_4 + \Psi_5 + \Psi_6,$$

*where the $\Psi_i$ are upper bounded by*

$$\Psi_1 \le \mathcal{O}(T^{-\omega+\zeta-\frac{1}{2}}), \quad \Psi_2 \le \mathcal{O}(T^{\omega+2\nu+3\zeta+\frac{1}{2}}), \quad \Psi_3 \le \mathcal{O}(T^{4\zeta+3\nu+\frac{1}{2}}),$$

$$\Psi_4 \le \mathcal{O}(T^{2\nu+2\zeta+\frac{1}{2}}), \quad \Psi_5 \le \mathcal{O}(T^{\nu+\gamma+\zeta+\frac{1}{2}}), \quad \Psi_6 \le \mathcal{O}(T^{\nu+(2-\alpha)\zeta+\frac{1}{2}}),$$

*which guarantees convergence via an inversely proportional power law decay with respect to $T$. The maximal convergence rate is given by $\mathcal{O}(1/\sqrt{T})$.*

*Proof.* We analyze the convergence of the global objective, where model weights are updated in a distributed fashion via local $BiClip$ under heavy-tailed noise. By $L$-smoothness, we have

$$F(x_t) \leq F(x_{t-1}) + \langle \nabla F(x_{t-1}), x_t - x_{t-1} \rangle + \frac{L}{2} \|x_t - x_{t-1}\|^2$$

$$= F(x_{t-1}) + \underbrace{\eta_t \left\langle \nabla F(x_{t-1}), \frac{\Delta_t}{\sqrt{\widetilde{v}_t} + \tau} \right\rangle}_{A_1} + \frac{\eta_t^2 L}{2} \left\| \frac{\Delta_t}{\sqrt{\widetilde{v}_t} + \tau} \right\|^2,$$

which we further decompose via noting that

$$A_1 = \eta_t \left\langle \nabla F(x_{t-1}), \frac{\Delta_t(\sqrt{\widetilde{v}_{t-1}} - \sqrt{\widetilde{v}_t})}{(\sqrt{\widetilde{v}_t} + \tau)(\sqrt{\widetilde{v}_{t-1}} + \tau)} \right\rangle + \eta_t \left\langle \nabla F(x_{t-1}), \frac{\Delta_t}{\sqrt{\widetilde{v}_{t-1}} + \tau} \right\rangle$$

$$= \eta_t \left\langle \nabla F(x_{t-1}), \frac{-\Delta_t^3}{(\sqrt{\widetilde{v}_t} + \tau)(\sqrt{\widetilde{v}_{t-1}} + \tau)(\sqrt{\widetilde{v}_{t-1}} + \sqrt{\widetilde{v}_t})} \right\rangle + \eta_t \left\langle \nabla F(x_{t-1}), \frac{\Delta_t}{\sqrt{\widetilde{v}_{t-1}} + \tau} \right\rangle$$

$$\leq \eta_t \left\langle |\nabla F(x_{t-1})|, \frac{|\Delta_t|^3}{(\sqrt{\widetilde{v}_t} + \tau)(\sqrt{\widetilde{v}_{t-1}} + \tau)(\sqrt{\widetilde{v}_{t-1}} + \sqrt{\widetilde{v}_t})} \right\rangle + \underbrace{\eta_t \left\langle \nabla F(x_{t-1}), \frac{\Delta_t}{\sqrt{\widetilde{v}_{t-1}} + \tau} \right\rangle}_{B_1}.$$

To bound $B_1$, we extract a negative gradient norm

$$B_1 = \underbrace{\eta_t \left\langle \nabla F(x_{t-1}), \frac{\Delta_t}{\sqrt{\widetilde{v}_{t-1}} + \tau} + \frac{K\eta_\ell^t \nabla F(x_{t-1})}{\sqrt{\widetilde{v}_{t-1}} + \tau} \right\rangle}_{B_2} - K\eta_t\eta_\ell^t \left\| \frac{\nabla F(x_{t-1})}{\sqrt{\sqrt{\widetilde{v}_{t-1}} + \tau}} \right\|^2,$$

where $B_2$ decomposes further into

$$B_2 = \eta_t \left\langle \nabla F(x_{t-1}), \frac{\Delta_t}{\sqrt{\widetilde{v}_{t-1}} + \tau} + \frac{\sum_{i \in [N]} \sum_{v \in [K]-1} p_i \eta_\ell^t (\nabla F_i(x_{i,v}^t) - \nabla F_i(x_{i,v}^t))}{\sqrt{\widetilde{v}_{t-1}} + \tau} + \frac{K\eta_\ell^t \nabla F(x_{t-1})}{\sqrt{\widetilde{v}_{t-1}} + \tau} \right\rangle$$

Here, we use the convention $[K] - 1 = \{0, \dots, K-1\}$, and that summation over null indices are zero (e.g. $\sum_{j=K}^{K-1} [\cdot] = 0$). Now, recall

$$\Delta_t := \sum_{i \in [N]} p_i \Delta_i^t = \sum_{i \in [N]} p_i (x_{i,K}^t - x_{i,0}^t) = -\sum_{i \in [N]} \sum_{v \in [K]-1} p_i \eta_\ell^t \cdot \hat{g}_{i,v}^t$$

$$= -\sum_{i \in [N]} \sum_{v \in [K]-1} p_i \eta_\ell^t \cdot BiClip(u_t, d_t, \nabla F_i(x_{i,v}^t) + \xi_{i,v}^t),$$

which implies $B_2 = C_1 + C_2$ for

$$C_1 = \eta_t \left\langle \nabla F(x_{t-1}), \frac{\sum_{i \in [N]} \sum_{v \in [K]-1} p_i \eta_\ell^t (\nabla F_i(x_{i,v}^t) - BiClip(u_t, d_t, \nabla F_i(x_{i,v}^t) + \xi_{i,v}^t))}{\sqrt{\widetilde{v}_{t-1}} + \tau} \right\rangle$$

$$C_2 = \eta_t \left\langle \nabla F(x_{t-1}), \frac{\sum_{i \in [N]} \sum_{v \in [K]-1} p_i \eta_\ell^t (\nabla F_i(x_{i,0}^t) - \nabla F_i(x_{i,v}^t))}{\sqrt{\widetilde{v}_{t-1}} + \tau} \right\rangle.$$

Letting $\mathbb{E}_t[\cdot]$ condition over all stochasticity up to global step $t$, we have that $\mathbb{E}_t[C_1]$ is equal to

$$\eta_t \left\langle \nabla F(x_{t-1}), \frac{\sum_{i \in [N]} \sum_{v \in [K]-1} p_i \eta_\ell^t (\mathbb{E}_t[\nabla F_i(x_{i,v}^t) + \xi_{i,v}^t - BiClip(u_t, d_t, \nabla F_i(x_{i,v}^t) + \xi_{i,v}^t)])}{\sqrt{\widetilde{v}_{t-1}} + \tau} \right\rangle.$$

For $D_1 := \mathbb{E}_t[\nabla F_i(x_{i,v}^t) + \xi_{i,v}^t - BiClip(u_t, d_t, \nabla F_i(x_{i,v}^t) + \xi_{i,v}^t)]$, we have by convexity and Jensen that

$$\|D_1\| \leq \mathbb{E}_t[\|\nabla F_i(x_{i,v}^t) + \xi_{i,v}^t - BiClip(u_t, d_t, \nabla F_i(x_{i,v}^t) + \xi_{i,v}^t)\|]$$

$$\leq d_t \mathbb{P}(\|\nabla F_i(x_{i,v}^t) + \xi_{i,v}^t\| \leq d_t)$$

$$+ \underbrace{\mathbb{E}_t[\|\nabla F_i(x_{i,v}^t) + \xi_{i,v}^t - BiClip(u_t, d_t, \nabla F_i(x_{i,v}^t) + \xi_{i,v}^t)\| \chi(\|\nabla F_i(x_{i,v}^t) + \xi_{i,v}^t\| \geq u_t)]}_{D_2}.$$

Piecewise continuity of $F_i(x)$ is clear via the existence of $\nabla F_i(x)$. This gives that

$$\mathbb{E}_t[\|\nabla F_i(x_{i,v}^t) + \xi_{i,v}^t\|^\alpha \chi\left(\|\nabla F_i(x_{i,v}^t) + \xi_{i,v}^t)\| \geq u_t\right)] \leq \mathbb{E}_t[\|\nabla F_i(x_{i,v}^t) + \xi_{i,v}^t\|^\alpha]$$

$$\leq 2^\alpha \mathbb{E}_t\left[\left\|\frac{\nabla F_i(x_{i,v}^t) + \xi_{i,v}^t}{2}\right\|^\alpha\right] \leq 2^\alpha \mathbb{E}_t\left[\frac{\|\nabla F_i(x_{i,v}^t)\|^\alpha}{2} + \frac{\|\xi_{i,v}^t\|^\alpha}{2}\right] = 2^{\alpha-1}(M^\alpha + B^\alpha),$$

where now, $M := \max_{x \in \mathcal{X}, i \in [N]} \|\nabla F_i(x)\|$. Thus, we may bound $D_2$ via reduction to the $\alpha$-moment:

$$D_2 \leq 2^{\alpha-1}(M^\alpha + B^\alpha)\mathbb{E}_t[\|\nabla F_i(x_{i,v}^t) + \xi_{i,v}^t\|^{1-\alpha}\chi\left(\|\nabla F_i(x_{i,v}^t) + \xi_{i,v}^t)\| \geq u_t\right)]$$

$$\leq 2^{\alpha-1}(M^\alpha + B^\alpha)u_t^{1-\alpha}\mathbb{P}\left(\|\nabla F_i(x_{i,v}^t; \xi_{i,v}^t)\| \geq u_t\right).$$

Collecting inequalities gives

$$\|D_1\| \leq d_t\mathbb{P}(\|\nabla F_i(x_{i,v}^t; \xi_{i,v}^t)\| \leq d_t) + 2^{\alpha-1}(M^\alpha + B^\alpha)u_t^{1-\alpha}\mathbb{P}\left(\|\nabla F_i(x_{i,v}^t; \xi_{i,v}^t)\| \geq u_t\right).$$

Therefore,

$$\mathbb{E}_t[C_1] \leq \frac{\eta_t G d}{\tau}\sum_{i \in [N]}\sum_{v \in [K]-1} p_i\eta_\ell^t d_t\mathbb{P}(\|\nabla F_i(x_{i,v}^t; \xi_{i,v}^t)\| \leq d_t)$$

$$+ \frac{2^{\alpha-1}\eta_t G d}{\tau}\sum_{i \in [N]}\sum_{v \in [K]-1} p_i\eta_\ell^t(M^\alpha + B^\alpha)u_t^{1-\alpha}\mathbb{P}\left(\|\nabla F_i(x_{i,v}^t; \xi_{i,v}^t)\| \geq u_t\right).$$

To bound $C_2$, we note that via $L$-smoothness, we have

$$C_2 \leq \frac{\eta_t GLd}{\tau}\sum_{i \in [N]}\sum_{v \in [K]-1} p_i\eta_\ell^t\|x_{i,0}^t - x_{i,v}^t\|$$

$$\leq \frac{\eta_t GLd}{\tau}\sum_{i \in [N]}\sum_{v \in [K]-1} p_i\eta_\ell^t\|x_{i,0}^t + \sum_{r=1}^{v-1}(x_{i,r}^t - x_{i,r}^t) - x_{i,v}^t\|$$

$$\leq \frac{\eta_t GLd}{\tau}\sum_{i \in [N]}\sum_{v \in [K]-1}\sum_{r \in [v]} p_i\eta_\ell^t\|x_{i,r}^t - x_{i,r-1}^t\|$$

$$\leq \frac{\eta_t GLK^2 d}{2\tau}(\eta_\ell^t)^2 u_t.$$

Noting that $\|\Delta_t\| \leq \eta_\ell^t u_t K$, we thus obtain

$$\mathbb{E}_t[F(x_t)] \leq F(x_{t-1}) + \frac{\eta_t^2(\eta_\ell^t)^2 u_t^2 K^2 L}{2\tau^2} + \frac{\eta_t G d K^3 u_t^3(\eta_\ell^t)^3}{\tau^3} - K\eta_t\eta_\ell^t\left\|\frac{\nabla F(x_{t-1})}{\sqrt{\sqrt{\tilde{v}_{t-1}} + \tau}}\right\|^2$$

$$+ \frac{\eta_t GLK^2 d}{2\tau}(\eta_\ell^t)^2 u_t + \frac{\eta_t G d}{\tau}\sum_{i \in [N]}\sum_{v \in [K]-1} p_i\eta_\ell^t d_t\mathbb{P}(\|\nabla F_i(x_{i,v}^t; \xi_{i,v}^t)\| \leq d_t)$$

$$+ \frac{2^{\alpha-1}\eta_t G d}{\tau}\sum_{i \in [N]}\sum_{v \in [K]-1} p_i\eta_\ell^t(M^\alpha + B^\alpha)u_t^{1-\alpha}\mathbb{P}\left(\|\nabla F_i(x_{i,v}^t; \xi_{i,v}^t)\| \geq u_t\right).$$

Taking expectations on both sides and telescoping gives via the tower law of expectation,

$$\underbrace{\sum_{t \in [T]} K\eta_t\eta_\ell^t\mathbb{E}\left[\left\|\frac{\nabla F(x_{t-1})}{\sqrt{\sqrt{\tilde{v}_{t-1}} + \tau}}\right\|^2\right]}_{E_1} \leq \underbrace{\mathbb{E}[F(x_T) - F(x_0)]}_{E_2} + \underbrace{\sum_{t \in [T]}\frac{\eta_t^2(\eta_\ell^t)^2 u_t^2 K^2 L}{2\tau^2}}_{E_3} + \underbrace{\sum_{t \in [T]}\frac{\eta_t G d K^3 u_t^3(\eta_\ell^t)^3}{\tau^3}}_{E_4}$$

$$+ \underbrace{\sum_{t \in [T]}\frac{\eta_t GLK^2 d}{2\tau}(\eta_\ell^t)^2 u_t}_{E_5} + \underbrace{\sum_{t \in [T]}\frac{\eta_t G d}{\tau}\sum_{i \in [N]}\sum_{v \in [K]-1} p_i\eta_\ell^t d_t\mathbb{P}(\|\nabla F_i(x_{i,v}^t; \xi_{i,v}^t)\| \leq d_t)}_{E_6}$$

$$+ \underbrace{\sum_{t \in [T]}\frac{2^{\alpha-1}\eta_t G d}{\tau}\sum_{i \in [N]}\sum_{v \in [K]-1} p_i\eta_\ell^t(M^\alpha + B^\alpha)u_t^{1-\alpha}\mathbb{P}\left(\|\nabla F_i(x_{i,v}^t; \xi_{i,v}^t)\| \geq u_t\right)}_{E_7},$$

where we have enumerated each term from $E_1$ to $E_7$ for clarity. To simplify notation, we now move to the asymptotic regime. Letting $\eta_t = \Theta(t^\omega)$, $\eta_\ell^t = \Theta(t^\nu)$, $d_t = \Theta(t^\gamma)$, and $u_t = \Theta(t^\zeta)$, we have via standard integral bounds that

$$E_1 \geq \Omega\left(T^{\omega+\nu+1} \cdot T^{-\zeta-\nu-\frac{1}{2}} \cdot \min_{t\in[T]} \mathbb{E}[\|\nabla F(x_t)\|^2]\right) = \Omega\left(T^{\omega-\zeta+\frac{1}{2}} \cdot \min_{t\in[T]} \mathbb{E}[\|\nabla F(x_t)\|]\right),$$

$$E_2 \leq \max_{x\in\mathcal{X}} F(x) - \min_{y\in\mathcal{X}} F(y) = \mathcal{O}(1), \quad E_3 \leq \mathcal{O}(T^{2\omega+2\nu+2\zeta+1}), \quad E_4 \leq \mathcal{O}(T^{\omega+3\zeta+3\nu+1}),$$

$$E_5 \leq \mathcal{O}(T^{\omega+2\nu+\zeta+1}), \quad E_6 \leq \mathcal{O}(T^{\omega+\nu+\gamma+1}), \quad E_7 \leq \mathcal{O}(T^{\omega+\nu+(1-\alpha)\zeta+1})$$

where any $E_i$ residues of $\mathcal{O}(1)$ for $i \geq 2$ have been incorporated into the upper bound for $E_2$. We note that the bound may be sharpened as the probabilistic terms must necessarily decay if $d_t \to 0$, $u_t \to \infty$, which further diminishes $E_6$, $E_7$. Now, to attain convergence of the minimal gradient, we impose the conditions

$$\Lambda_1 : \zeta > 0 \quad \text{and} \quad \gamma < 0, \quad \Lambda_2 : \omega - \zeta + \frac{1}{2} > 0, \quad \Lambda_3 : \omega + 2\nu + 3\zeta + \frac{1}{2} < 0,$$

$$\Lambda_4 : 4\zeta + 3\nu + \frac{1}{2} < 0, \quad \Lambda_5 : 2\nu + 2\zeta + \frac{1}{2} < 0, \quad \Lambda_6 : \nu + \gamma + \zeta + \frac{1}{2} < 0,$$

$$\Lambda_7 : \nu + (2-\alpha)\zeta + \frac{1}{2} < 0.$$

We note that each condition $\Lambda_{i\geq 2}$ comes from $E_i/E_1 \to 0$, $T \to \infty$, as any residual terms are subsumed by $\mathcal{O}(1)$, which decays via $\Lambda_2$. Setting $0 < \zeta < 1/4$, we have

$$\nu < \min\{-\frac{1}{6} - \frac{4}{3}\zeta, -\frac{1}{4} - \frac{3}{2}\zeta - \frac{1}{2}\omega, -\frac{1}{2} + (\alpha-2)\zeta\}$$

$$\gamma < -\nu - \zeta - \frac{1}{2}, \quad \omega + \frac{1}{2} > \zeta, \quad -\frac{1}{2} < \omega \leq 0.$$

Therefore, any such selection stabilizes the minimum gradient, which guarantees convergence. It is straightforward to see that $\Lambda_2$ is the dominating condition, for which $\omega \leq 0$ and $\zeta \in (0, 1/4)$ gives the convergence rate $\mathcal{O}(1/\sqrt{T})$ as $\omega = 0$ and $\zeta \to 0^+$. $\qquad\square$

**Remark 2.** *In the case of coordinate-wise clipping, all major adjustments up to a scaling factor of $\sqrt{d}$ are made in the terms bounding $\mathbb{E}[C_1]$. In this case, the proof proceeds as follows.*

Defining $|\cdot|$ to act coordinatewise, $\mathbb{E}_t[C_1]$ is now less than or equal to

$$\eta_t \left\langle |\nabla F(x_{t-1})|, \frac{\sum_{i\in[N]}\sum_{v\in[K]-1} p_i \eta_\ell^t |\mathbb{E}_t[\nabla F_i(x_{i,v}^t) + \xi_{i,v}^t - BiClip(u_t, d_t, \nabla F_i(x_{i,v}^t) + \xi_{i,v}^t)]|}{\sqrt{\tilde{v}_{t-1}} + \tau} \right\rangle.$$

Therefore by Jensen,

$$\mathbb{E}_t[C_1] \leq \frac{\eta_t \eta_\ell^t G}{\tau} \sum_{i\in[N]} \sum_{v\in[K]-1} \sum_{j\in[d]} p_i \mathbb{E}_t[\underbrace{|\nabla F_i(x_{i,v}^t) + \xi_{i,v}^t - BiClip(u_t, d_t, \nabla F_i(x_{i,v}^t) + \xi_{i,v}^t)|_j}_{D_{1,j}}].$$

We note that $\mathbb{E}_t[D_{1,j}]$ can be upper bounded by $D_{2,j} + D_{3,j}$ where

$$D_{2,j} = \mathbb{E}_t\left[D_{1,j} \cdot \chi\left(|\nabla F_i(x_{i,v}^t; \xi_{i,v}^t)|_j \leq d_t\right)\right] \leq d_t \mathbb{P}\left(|\nabla F_i(x_{i,v}^t; \xi_{i,v}^t)|_j \leq d_t\right)$$

$$D_{3,j} = \mathbb{E}_t\left[|\nabla F_i(x_{i,v}^t; \xi_{i,v}^t)|_j \chi\left(|\nabla F_i(x_{i,v}^t; \xi_{i,v}^t)|_j \geq u_t\right)\right].$$

It follows that

$$D_{3,j} \leq \mathbb{E}_t\left[|\nabla F_i(x_{i,v}^t; \xi_{i,v}^t)|_j^\alpha |\nabla F_i(x_{i,v}^t; \xi_{i,v}^t)|_j^{1-\alpha} \chi\left(|\nabla F_i(x_{i,v}^t; \xi_{i,v}^t)|_j \geq u_t\right)\right]$$

$$\leq 2^{\alpha-1}(M^\alpha + B^\alpha)u_t^{1-\alpha}\mathbb{P}\left(|\nabla F_i(x_{i,v}^t; \xi_{i,v}^t)|_j \geq u_t\right).$$

Note that we used coordinate-wise bounded alpha moments for some $\alpha \in (1, 2)$, $\mathbb{E}[|\xi_i|_j^\alpha] \leq B_{i,j}^\alpha$. We therefore define the $M$ and $B$ to be

$$M := \max_{x\in\mathcal{X}, i\in[N], j\in[d]} |\nabla F_i(x)|_j \quad \text{and} \quad B = \max_{i\in[N], j\in[d]} B_{i,j}.$$

Comparing terms gives the identical asymptotic order of convergence to $L_2$ clipping in Theorem 6.

---

**Algorithm 6** RMS-$TailClip$

---

**Require:** Initial model $x_1$, learning rate schedule $\eta_t$, clipping schedules $u_t, d_t$
    Synchronization timestep $z \in \mathbb{Z}_{>0}$, adaptivity/EMA parameters $\tau > 0, \widetilde{\beta}_2 \in [0, 1)$
1: **for** $t = 1, \dots, T$ **do**
2:    **for** each node $i \in [N]$ in parallel **do**
3:       $x_{i,0}^t \leftarrow x_t$
4:       **for** each local step $k \in [z]$ **do**
5:          Draw minibatch gradient $g_{i,k}^t = \nabla F_i(x_{i,k}^t, \xi_{i,k}^t)$
6:          $x_{i,k}^{t+1} \leftarrow x_{i,k}^t - \eta_t \cdot TailClip(u_t, d_t, g_{i,k}^t)$
7:       **end for**
8:    **end for**
9:    $\Delta_t = \frac{1}{N} \sum_{i \in [N]} \left( x_{i,z}^t - x_{t-1} \right), \quad \widetilde{m}_t \leftarrow \Delta_t$
10:   $\widetilde{v}_t = \widetilde{\beta}_2 \widetilde{v}_{t-1} + (1 - \widetilde{\beta}_2)\Delta_t^2$
11:   $x_t = x_{t-1} + \eta \frac{\widetilde{m}_t}{\sqrt{\widetilde{v}_t} + \tau}$
12: **end for**

---

### D.5   Convergence of RMSProp-$TailClip$

For Algorithm 6, we have the following convergence bound.

**Theorem 7.** *For clipping and learning rate thresholds satisfying $\eta_t = \Theta(t^\omega)$, $\eta_\ell^t = \Theta(t^\nu)$, $d_t = \Theta(t^\gamma)$, and $u_t = \Theta(t^\zeta)$, let the conditions listed in Theorem 6 hold. Then, local $BiClip$ with outer optimizer RMSProp stabilizes the expected minimum gradient $\min_{t \in [T]} \mathbb{E}[\|\nabla F(x_t)\|^2] \to 0^+$ with maximal rate $\mathcal{O}(1/\sqrt{T})$. Here, the exponential moving average parameter of the second pseudogradient moment is fixed within the range $\widetilde{\beta}_2 \in [0, 1)$.*

*Proof.* The proof for outer optimizer RMSProp builds on the prior proof for $BiClip$ with outer optimizer Adagrad. We skip repeated details for clarity of exposition, and concisely present only the main steps and ideas central to the proof for readability. $L$-smoothness gives as before

$$F(x_t) \le F(x_{t-1}) + \langle \nabla F(x_{t-1}), x_t - x_{t-1} \rangle + \frac{L}{2} \|x_t - x_{t-1}\|^2$$

$$= F(x_{t-1}) + \eta_t \left\langle \nabla F(x_{t-1}), \frac{\Delta_t}{\sqrt{\widetilde{v}_t} + \tau} \right\rangle + \frac{\eta_t^2 L}{2} \left\| \frac{\Delta_t}{\sqrt{\widetilde{v}_t} + \tau} \right\|^2. \tag{10}$$

We note the decomposition

$$\left\langle \nabla F(x_{t-1}), \frac{\Delta_t}{\sqrt{\widetilde{v}_t} + \tau} \right\rangle = \underbrace{\left\langle \nabla F(x_{t-1}), \frac{\Delta_t}{\sqrt{\widetilde{v}_t} + \tau} - \frac{\Delta_t}{\sqrt{\widetilde{\beta}_2 \widetilde{v}_{t-1}} + \tau} \right\rangle}_{B_1} + \underbrace{\left\langle \nabla F(x_{t-1}), \frac{\Delta_t}{\sqrt{\widetilde{\beta}_2 \widetilde{v}_{t-1}} + \tau} \right\rangle}_{B_2}.$$

To form an upper bound, we use that

$$B_2 = \underbrace{\left\langle \nabla F(x_{t-1}), \frac{\Delta_t}{\sqrt{\widetilde{\beta}_2 \widetilde{v}_{t-1}} + \tau} + \frac{K \eta_\ell^t \nabla F(x_{t-1})}{\sqrt{\widetilde{\beta}_2 \widetilde{v}_{t-1}} + \tau} \right\rangle}_{C_0} - K \eta_\ell^t \left\| \frac{\nabla F(x_{t-1})}{\sqrt{\sqrt{\widetilde{\beta}_2 \widetilde{v}_{t-1}} + \tau}} \right\|^2$$

where $C_0 = C_1 + C_2$ for

$$C_1 = \left\langle \nabla F(x_{t-1}), \frac{\sum_{i \in [N]} \sum_{v \in [K]-1} p_i \eta_\ell^t (\nabla F_i(x_{i,v}^t) - BiClip(u_t, d_t, \nabla F_i(x_{i,v}^t) + \xi_{i,v}^t))}{\sqrt{\widetilde{\beta}_2 \widetilde{v}_{t-1}} + \tau} \right\rangle$$

$$C_2 = \left\langle \nabla F(x_{t-1}), \frac{\sum_{i \in [N]} \sum_{v \in [K]-1} p_i \eta_\ell^t (\nabla F_i(x_{i,0}^t) - \nabla F_i(x_{i,v}^t))}{\sqrt{\widetilde{\beta}_2 \widetilde{v}_{t-1}} + \tau} \right\rangle.$$

By the tower law and conditioning on stochastic realizations up to $t - 1$, we have as before

$$\mathbb{E}[C_0] \leq \frac{Gd}{\tau} \sum_{i \in [N]} \sum_{v \in [K]-1} p_i \eta_\ell^t d_t \mathbb{P}(\|\nabla F_i(x_{i,v}^t; \xi_{i,v}^t))\| \leq d_t) + \frac{GLK^2 d}{2\tau}(\eta_\ell^t)^2 u_t$$

$$+ \frac{2^{\alpha-1} Gd}{\tau} \sum_{i \in [N]} \sum_{v \in [K]-1} p_i \eta_\ell^t (M^\alpha + B^\alpha) u_t^{1-\alpha} \mathbb{P}\left(\|\nabla F_i(x_{i,v}^t; \xi_{i,v}^t))\| \geq u_t\right)$$

$$\leq \frac{Gd}{\tau} K \eta_\ell^t d_t + \frac{GLK^2 d}{2\tau}(\eta_\ell^t)^2 u_t + \frac{2^{\alpha-1} Gd}{\tau} K \eta_\ell^t (M^\alpha + B^\alpha) u_t^{1-\alpha}.$$

To bound $B_1$, we have

$$B_1 = \left\langle \nabla F(x_{t-1}), \frac{\Delta_t}{\sqrt{\widetilde{v}_t} + \tau} - \frac{\Delta_t}{\sqrt{\widetilde{\beta}_2 \widetilde{v}_{t-1}} + \tau} \right\rangle$$

$$= \left\langle \nabla F(x_{t-1}), \frac{(\widetilde{\beta}_2 - 1)\Delta_t^3}{\left(\sqrt{\widetilde{v}_t} + \tau\right)\left(\sqrt{\widetilde{\beta}_2 \widetilde{v}_{t-1}} + \tau\right)\left(\sqrt{\widetilde{v}_t} + \sqrt{\widetilde{\beta}_2 \widetilde{v}_{t-1}}\right)} \right\rangle$$

We prepare the global inequality equation 10 for telescoping. It is straightforward to see that collecting inequalities gives

$$\mathbb{E}[F(x_t)] \leq \mathbb{E}[F(x_{t-1})] + \frac{\eta_t^2 LK^2 u_t^2 (\eta_\ell^t)^2}{2\tau^2} - K \eta_t \eta_\ell^t \left\| \frac{\nabla F(x_{t-1})}{\sqrt{\sqrt{\widetilde{\beta}_2 \widetilde{v}_{t-1}} + \tau}} \right\|^2$$

$$\frac{Gd}{\tau} K \eta_t \eta_\ell^t d_t + \frac{GLK^2 d}{2\tau} \eta_t (\eta_\ell^t)^2 u_t + \frac{2^{\alpha-1} Gd}{\tau} K \eta_t \eta_\ell^t (M^\alpha + B^\alpha) u_t^{1-\alpha} + \frac{dG(1 - \widetilde{\beta}_2)(u_t \eta_\ell^t)^3}{\tau^3}$$

Rearranging and telescoping gives

$$\sum_{t=1}^{T} K \eta_t \eta_\ell^t \mathbb{E}\left[\left\| \frac{\nabla F(x_{t-1})}{\sqrt{\sqrt{\widetilde{\beta}_2 \widetilde{v}_{t-1}} + \tau}} \right\|^2\right] \leq \mathbb{E}[F(x_0)] - \mathbb{E}[F(x_T)] + \sum_{t=1}^{T} \frac{\eta_t^2 LK^2 u_t^2 (\eta_\ell^t)^2}{2\tau^2}$$

$$+ \sum_{t=1}^{T} \left( \frac{Gd}{\tau} K \eta_t \eta_\ell^t d_t + \frac{GLK^2 d}{2\tau} \eta_t (\eta_\ell^t)^2 u_t + \frac{2^{\alpha-1} Gd}{\tau} K \eta_t \eta_\ell^t (M^\alpha + B^\alpha) u_t^{1-\alpha} + \frac{dG(1 - \widetilde{\beta}_2)(u_t \eta_\ell^t)^3}{\tau^3} \right)$$

By non-negativity of squared pseudogradients, we immediately obtain $\widetilde{\beta}_2 \widetilde{v}_{t-1} \leq \widetilde{v}_{t-1}$. Therefore up to constants, the convergence bound collapses to asymptotically equivalent bounds than that of Theorem 6, up to constant multiples from the exponentially decaying moving average of the second moment pseudogradient. The modification to coordinate-wise clipping instead of $L_2$ clipping follows analogous steps. $\qquad \square$

Incorporating momentum into the first pseudogradient moment further complicates the analysis, and yields the results presented in Section D.6.

### D.6 CONVERGENCE OF ADAM-$TailClip$

By incorporating a moving average of the first pseudogradient moment as a form of momentum, we derive Algorithm 7. For this variant, we demonstrate that the expected minimal gradient does not diverge, even when the algorithm undergoes restarts. Practically, this ensures that the located gradient value update of any single step remains bounded in expectation. The key challenge in proving convergence to 0 arises from the moving average applied to the first moment, which effectively retains historical gradient information, significantly complicating the proof structure. Investigating the conditions required to guarantee convergence under this framework presents a promising avenue for future research. Our bound highlights that the dominating terms are influenced by the upper clipping threshold $u_r$, suggesting that the algorithm's convergence behavior may be closely related the choice of this threshold and can be tuned in practice.

---

**Algorithm 7** Adam-$TailClip$

---

**Require:** Initial model $x_1$, learning rate schedule $\eta_t$, clipping schedules $u_t$, $d_t$
    Synchronization timestep $z \in \mathbb{Z}_{>0}$, adaptivity/EMA parameters $\tau > 0$, $\widetilde{\beta}_1, \widetilde{\beta}_2 \in [0, 1)$
1: **for** $t = 1, \dots, T$ **do**
2:    **for** each node $i \in [N]$ in parallel **do**
3:       $x_{i,0}^t \leftarrow x_t$
4:       **for** each local step $k \in [z]$ **do**
5:          Draw minibatch gradient $g_{i,k}^t = \nabla F_i(x_{i,k}^t, \xi_{i,k}^t)$
6:          $x_{i,k}^{t+1} \leftarrow x_{i,k}^t - \eta_t \cdot TailClip(u_t, d_t, g_{i,k}^t)$
7:       **end for**
8:    **end for**
9:    $\Delta_t = \frac{1}{N} \sum_{i \in [N]} \left( x_{i,z}^t - x_{t-1} \right)$
10:   $\widetilde{m}_t = \widetilde{\beta}_1 \widetilde{m}_{t-1} + (1 - \widetilde{\beta}_1) \Delta_t$
11:   $\widetilde{v}_t = \widetilde{\beta}_2 \widetilde{v}_{t-1} + (1 - \widetilde{\beta}_2) \Delta_t^2$
12:   $x_t = x_{t-1} + \eta \frac{\widetilde{m}_t}{\sqrt{\widetilde{v}_t} + \tau}$
13: **end for**

---

**Theorem 8.** *Let the exponentially decaying moving average parameters satisfy $\widetilde{\beta}_1 \in (0, 1)$, $\widetilde{\beta}_2 \in [0, 1)$ for the outer optimizer first and second order pseudogradient moments, respectively. Extremize the unbiased stochastic noise such that $\nexists \alpha_k \in (1, 2)$ for which $\mathbb{E}[\|\xi_k\|^{\alpha_k}] < B_k^{\alpha_k}$ for integrable $\xi_k$. Then, Algorithm 7 gives under constant upper clipping threshold invariant to global timestep $t$ ($\zeta = 0$) that*

$$\min_{t \in [T]} \mathbb{E}[\|\nabla F(x_t)\|^2] \lesssim \mathcal{O}(1),$$

*where for $\eta_t = \Theta(t^\omega)$, $\eta_\ell^t = \Theta(t^\nu)$, and $d_t = \Theta(t^\gamma)$, we impose*

$$\nu \in (-1, 0), \quad -\nu - 1 < \omega \le 0, \quad -(1 + \nu + \omega) < \gamma < 0. \tag{11}$$

*Proof.* As in the case of outer optimizer Adagrad, we analyze the convergence of the global objective. By $L$-smoothness, we have

$$F(x_t) \le F(x_{t-1}) + \langle \nabla F(x_{t-1}), x_t - x_{t-1} \rangle + \frac{L}{2} \|x_t - x_{t-1}\|^2$$

$$= F(x_{t-1}) + \eta_t \left\langle \nabla F(x_{t-1}), \underbrace{\frac{\widetilde{\beta}_1^t \widetilde{m}_0 + (1 - \widetilde{\beta}_1) \sum_{r=1}^t \widetilde{\beta}_1^{t-r} \Delta_r}{\sqrt{\widetilde{v}_t} + \tau}}_{A_1} \right\rangle + \frac{\eta_t^2 L}{2} \|A_1\|^2. \tag{12}$$

To proceed with the proof, we note that

$$\langle \nabla F(x_{t-1}), A_1 \rangle = \left\langle \nabla F(x_{t-1}), \frac{\widetilde{\beta}_1^t \widetilde{m}_0}{\sqrt{\widetilde{v}_t} + \tau} \right\rangle + (1 - \widetilde{\beta}_1) \sum_{r=1}^t \widetilde{\beta}_1^{t-r} \left\langle \nabla F(x_{t-1}), \frac{\Delta_r}{\sqrt{\widetilde{v}_t} + \tau} \right\rangle,$$

which we further decompose by using

$$\left\langle \nabla F(x_{t-1}), \frac{\Delta_r}{\sqrt{\widetilde{v}_t} + \tau} \right\rangle = \underbrace{\sum_{q=0}^{t-r} \left\langle \nabla F(x_{t-1}), \frac{\Delta_r}{\sqrt{\widetilde{\beta}_2^q \widetilde{v}_{t-q}} + \tau} - \frac{\Delta_r}{\sqrt{\widetilde{\beta}_2^{q+1} \widetilde{v}_{t-q-1}} + \tau} \right\rangle}_{A_{1,q}}$$

$$+ \underbrace{\left\langle \nabla F(x_{t-1}) - \nabla F(x_{r-1}), \frac{\Delta_r}{\sqrt{\widetilde{\beta}_2^{t-r+1} \widetilde{v}_{r-1}} + \tau} \right\rangle}_{B_1} + \underbrace{\left\langle \nabla F(x_{r-1}), \frac{\Delta_r}{\sqrt{\widetilde{\beta}_2^{t-r+1} \widetilde{v}_{r-1}} + \tau} \right\rangle}_{B_2}.$$

We have that

$$A_{1,q} = \sum_{q=0}^{t-r} \left\langle \nabla F(x_{t-1}), \frac{\Delta_r \left( \sqrt{\widetilde{\beta}_2^{q+1} \widetilde{v}_{t-q-1}} - \sqrt{\widetilde{\beta}_2^q \widetilde{v}_{t-q}} \right)}{\left( \sqrt{\widetilde{\beta}_2^q \widetilde{v}_{t-q}} + \tau \right) \left( \sqrt{\widetilde{\beta}_2^{q+1} \widetilde{v}_{t-q-1}} + \tau \right)} \right\rangle = \sum_{q=0}^{t-r} B_{1,q}$$

$$:= \sum_{q=0}^{t-r} \left\langle \nabla F(x_{t-1}), \frac{-(1-\widetilde{\beta}_2)\widetilde{\beta}_2^q \Delta_{t-q}^2 \Delta_r}{\left( \sqrt{\widetilde{\beta}_2^q \widetilde{v}_{t-q}} + \tau \right) \left( \sqrt{\widetilde{\beta}_2^{q+1} \widetilde{v}_{t-q-1}} + \tau \right) \left( \sqrt{\widetilde{\beta}_2^{q+1} \widetilde{v}_{t-q-1}} + \sqrt{\widetilde{\beta}_2^q \widetilde{v}_{t-q}} \right)} \right\rangle.$$

To upper bound $B_2$, we observe

$$B_2 = \underbrace{\left\langle \nabla F(x_{r-1}), \frac{\Delta_r}{\sqrt{\widetilde{\beta}_2^{t-r+1} \widetilde{v}_{r-1}} + \tau} + \frac{K\eta_\ell^r \nabla F(x_{r-1})}{\sqrt{\widetilde{\beta}_2^{t-r+1} \widetilde{v}_{r-1}} + \tau} \right\rangle}_{C_{0,r}} - K\eta_\ell^r \left\| \frac{\nabla F(x_{r-1})}{\sqrt{\sqrt{\widetilde{\beta}_2^{t-r+1} \widetilde{v}_{r-1}} + \tau}} \right\|^2$$

where $C_{0,r} = C_{1,r} + C_{2,r}$ for

$$C_{1,r} = \left\langle \nabla F(x_{r-1}), \frac{\sum_{i\in[N]} \sum_{v\in[K]-1} p_i \eta_\ell^r (\nabla F_i(x_{i,v}^r) - BiClip(u_r, d_r, \nabla F_i(x_{i,v}^r) + \xi_{i,v}^r))}{\sqrt{\widetilde{\beta}_2^{t-r+1} \widetilde{v}_{r-1}} + \tau} \right\rangle$$

$$C_{2,r} = \left\langle \nabla F(x_{r-1}), \frac{\sum_{i\in[N]} \sum_{v\in[K]-1} p_i \eta_\ell^r (\nabla F_i(x_{i,0}^r) - \nabla F_i(x_{i,v}^r))}{\sqrt{\widetilde{\beta}_2^{t-r+1} \widetilde{v}_{r-1}} + \tau} \right\rangle.$$

Noting that $\mathbb{E}[\,\cdot\,] = \mathbb{E}[\mathbb{E}_r[\,\cdot\,]]$ by the tower law, we have as before

$$\mathbb{E}[C_{0,r}] \le \frac{Gd}{\tau} \sum_{i\in[N]} \sum_{v\in[K]-1} p_i \eta_\ell^r d_r \mathbb{P}(\|\nabla F_i(x_{i,v}^r; \xi_{i,v}^r)\| \le d_r) + \frac{GLK^2 d}{2\tau} (\eta_\ell^r)^2 u_r$$

$$+ \frac{2^{\alpha-1} Gd}{\tau} \sum_{i\in[N]} \sum_{v\in[K]-1} p_i \eta_\ell^r (M^\alpha + B^\alpha) u_r^{1-\alpha} \mathbb{P}\left( \|\nabla F_i(x_{i,v}^r; \xi_{i,v}^r)\| \ge u_r \right)$$

$$\le \frac{Gd}{\tau} K\eta_\ell^r d_r + \frac{GLK^2 d}{2\tau} (\eta_\ell^r)^2 u_r + \frac{2^{\alpha-1} Gd}{\tau} K\eta_\ell^r (M^\alpha + B^\alpha) u_r^{1-\alpha}.$$

We retain the $\alpha$ for clarity and to draw comparision to previous proofs, however we note that $\alpha = 1$ as higher moments do not exist. Now, to bound $B_1$, we use $L$-smoothness:

$$\|B_1\| \le \frac{L\eta_\ell^r u_r K}{\tau} \|x_{t-1} - x_{r-1}\| \le \frac{L\eta_\ell^r u_r K \operatorname{diam}(\mathcal{X})}{\tau}.$$

Collecting all inequalities gathered thus far gives

$$\mathbb{E}[F(x_t)] \le \mathbb{E}[F(x_{t-1})] + \frac{\eta_t^2 L}{2} \mathbb{E}[\|A_1\|^2] + \widetilde{\beta}_1^t \eta_t \mathbb{E}\left[ \left\langle \nabla F(x_{t-1}), \frac{\widetilde{m}_0}{\sqrt{\widetilde{v}_t} + \tau} \right\rangle \right]$$

$$+ (1-\widetilde{\beta}_1)\eta_t \sum_{r=1}^{t} \widetilde{\beta}_1^{t-r} \left( \sum_{q=0}^{t-r} \mathbb{E}[B_{1,q}] - K\eta_\ell^r \mathbb{E}\left[ \left\| \frac{\nabla F(x_{r-1})}{\sqrt{\sqrt{\widetilde{\beta}_2^{t-r+1} \widetilde{v}_{r-1}} + \tau}} \right\|^2 \right] + \frac{L\eta_\ell^r u_r K \operatorname{diam}(\mathcal{X})}{\tau} \right)$$

$$+ (1-\widetilde{\beta}_1)\eta_t \sum_{r=1}^{t} \widetilde{\beta}_1^{t-r} \left( \frac{Gd}{\tau} K\eta_\ell^r d_r + \frac{GLK^2 d}{2\tau} (\eta_\ell^r)^2 u_r + \frac{2^{\alpha-1} Gd}{\tau} K\eta_\ell^r (M^\alpha + B^\alpha) u_r^{1-\alpha} \right).$$

We note the use of Jensen and convexity to ensure $\|\mathbb{E}[B_1]\| \leq \mathbb{E}[\|B_1\|]$. We now rearrange and telescope $t \in [1, T]$:

$$\underbrace{(1 - \widetilde{\beta}_1) \sum_{t=1}^{T} \eta_t \sum_{r=1}^{t} \widetilde{\beta}_1^{t-r} \left( K \eta_\ell^r \mathbb{E}\left[ \left\| \frac{\nabla F(x_{r-1})}{\sqrt{\sqrt{\widetilde{\beta}_2^{t-r+1} \widetilde{v}_{r-1}} + \tau}} \right\|^2 \right] \right)}_{F_1} \leq \underbrace{\mathbb{E}[F(x_0)] - \mathbb{E}[F(x_T)]}_{F_2} + \underbrace{\sum_{t=1}^{T} \frac{\eta_t^2 L}{2} \mathbb{E}[\|A_1\|^2]}_{F_3}$$

$$+ \underbrace{\sum_{t=1}^{T} \eta_t \widetilde{\beta}_1^t \mathbb{E}\left[ \left\langle \nabla F(x_{t-1}), \frac{\widetilde{m}_0}{\sqrt{\widetilde{v}_t} + \tau} \right\rangle \right]}_{F_4} + \underbrace{(1 - \widetilde{\beta}_1) \sum_{t=1}^{T} \eta_t \sum_{r=1}^{t} \widetilde{\beta}_1^{t-r}}_{F_5} \left( \underbrace{\sum_{q=0}^{t-r} \mathbb{E}[B_{1,q}]}_{F_6} + \underbrace{\frac{L \eta_\ell^r u_r K \operatorname{diam}(\mathcal{X})}{\tau}}_{F_7} \right)$$

$$+ \underbrace{(1 - \widetilde{\beta}_1) \sum_{t=1}^{T} \eta_t \sum_{r=1}^{t} \widetilde{\beta}_1^{t-r}}_{F_5} \left( \underbrace{\frac{Gd}{\tau} K \eta_\ell^r d_r}_{F_8} + \underbrace{\frac{GLK^2 d}{2\tau} (\eta_\ell^r)^2 u_r}_{F_9} + \underbrace{\frac{2^{\alpha-1} Gd}{\tau} K \eta_\ell^r (M^\alpha + B^\alpha) u_r^{1-\alpha}}_{F_{10}} \right).$$

We now aim to bound each term in the left hand side from below, and right hand side from above. Letting $\eta_t = \Theta(t^\omega)$, $\eta_\ell^t = \Theta(t^\nu)$, $d_t = \Theta(t^\gamma)$, and $u_t = \Theta(t^\zeta)$, we move to the asymptotic regime to simplify notation and suppress auxiliary constants for readability. We have that

$$(1 - \widetilde{\beta}_1) \sum_{t=1}^{T} \sum_{r=1}^{t} \eta_t \widetilde{\beta}_1^{t-r} \eta_\ell^r = (1 - \widetilde{\beta}_1) \sum_{t=1}^{T} \eta_t \widetilde{\beta}_1^t \left( \sum_{r=1}^{t} \widetilde{\beta}_1^{-r} \eta_\ell^r \right) \gtrsim (1 - \widetilde{\beta}_1) \sum_{t=1}^{T} \eta_t \widetilde{\beta}_1^t \int_1^t \widetilde{\beta}_1^{-r} r^\nu \, \mathrm{d}r. \tag{13}$$

Then, L'Hôpital's rule allows us to derive an asymptotically sharp bound as follows:

$$\int_1^t \widetilde{\beta}_1^{-r} r^\nu \, \mathrm{d}r = \left[ \frac{\widetilde{\beta}_1^{-r} r^\nu}{-\log_e(\widetilde{\beta}_1)} \right]_{r=1}^{t} - \int_1^t \frac{\nu \widetilde{\beta}_1^{-r} r^{\nu-1}}{-\log_e(\widetilde{\beta}_1)} \, \mathrm{d}r \gtrsim \frac{\widetilde{\beta}_1^{-t} t^\nu}{|\log_e(\widetilde{\beta}_1)|} \tag{14}$$

Here, we used that $\nu \leq 0$ and $0 < \widetilde{\beta}_1 < e$. Asymptotic equivalence is verified via

$$\lim_{t \to \infty} \frac{|\log_e(\widetilde{\beta}_1)| (\int_1^t \widetilde{\beta}_1^{-r} r^\nu \, \mathrm{d}r)}{\widetilde{\beta}_1^{-t} t^\nu} = \lim_{t \to \infty} \frac{|\log_e(\widetilde{\beta}_1)| \widetilde{\beta}_1^{-t} t^\nu}{-\log_e(\widetilde{\beta}_1) \widetilde{\beta}_1^{-t} t^\nu + \nu \widetilde{\beta}_1^{-t} t^{\nu-1}} = 1.$$

Therefore, the rightmost side of equation 14 is an asymptotically sharp approximation, relieving the condition $\nu \leq 0$ for validity of the approximation. Within $\widetilde{\beta}_1 \in (0, 1)$, the approximation diverges as expected, validating the asymptotic analysis. Recall that $|\Delta_r| \leq K \eta_\ell^r u_r$, which now gives via equation 14

$$\widetilde{\beta}_2^{t-r+1} \widetilde{v}_{r-1} \lesssim \sum_{z=1}^{r-1} \widetilde{\beta}_2^{r-1-z} \Delta_z^2 \lesssim \widetilde{\beta}_2^{r-1} \sum_{z=1}^{r-1} \widetilde{\beta}_2^{-z} (\eta_\ell^z)^2 u_z^2 \lesssim \max \left\{ \mathcal{O}(1), T^{2(\nu+\zeta)} \right\}. \tag{15}$$

Here, we used $\widetilde{\beta}_2 \leq 1$ and $r \leq T$. We thus obtain

$$(1 - \widetilde{\beta}_1) \sum_{t=1}^{T} \sum_{r=1}^{t} \eta_t \widetilde{\beta}_1^{t-r} \eta_\ell^r \gtrsim (1 - \widetilde{\beta}_1) \sum_{t=1}^{T} \eta_t \frac{t^\nu}{|\log_e(\widetilde{\beta}_1)|} \gtrsim (1 - \widetilde{\beta}_1) \int_1^T \frac{t^{\omega+\nu}}{\log_e(\widetilde{\beta}_1)} \mathrm{d}t \approx \frac{(1 - \widetilde{\beta}_1) T^{\omega+\nu+1}}{(\omega+\nu+1) |\log_e(\widetilde{\beta}_1)|}.$$

Therefore as $\nu + \zeta < 0$, we conclude that

$$F_1 \gtrsim \Omega \left( \frac{(1 - \widetilde{\beta}_1)}{(\omega+\nu+1) \log_e(\widetilde{\beta}_1)} \cdot T^{\omega+\nu+1} \cdot \min_{t \in [T]} \mathbb{E}[\|\nabla F(x_t)\|^2] \right).$$

Clearly, $F_2 \lesssim \mathcal{O}(1)$. To bound $F_3$, we have

$$F_3 = \sum_{t=1}^{T} \frac{\eta_t^2 L}{2} \left\| \frac{\widetilde{\beta}_1^t \widetilde{m}_0 + (1 - \widetilde{\beta}_1) \sum_{r=1}^{t} \widetilde{\beta}_1^{t-r} \Delta_r}{\sqrt{\widetilde{v}_t} + \tau} \right\|^2 \lesssim \sum_{t=1}^{T} \frac{t^{2\omega}}{\tau^2} \left( \widetilde{\beta}_1^{2t} \|\widetilde{m}_0\|^2 + (1 - \widetilde{\beta}_1)^2 \left\| \sum_{r=1}^{t} \widetilde{\beta}_1^{t-r} \Delta_r \right\|^2 \right)$$

$$\lesssim \frac{\mathcal{O}(1)}{\tau^2} + \frac{(1 - \widetilde{\beta}_1)^2 \sum_{t=1}^{T} t^{2\nu+2\zeta+2\omega}}{\tau^2 (\log_e(\widetilde{\beta}_1))^2} \lesssim \frac{\mathcal{O}(1)}{\tau^2} + \frac{(1 - \widetilde{\beta}_1)^2 \, T^{2(\nu+\zeta+\omega)+1}}{\tau^2 (\log_e(\widetilde{\beta}_1))^2}.$$

$F_4$ is bounded similarly after using Jensen,

$$|F_4| \leq \sum_{t=1}^T \eta_t \widetilde{\beta}_1^t \mathbb{E}\left[\left\langle |\nabla F(x_{t-1})|, \frac{|\widetilde{m}_0|}{\sqrt{\widetilde{v}_t} + \tau}\right\rangle\right] \leq \sum_{t=1}^T \eta_t \widetilde{\beta}_1^t dG \cdot \max_{j \in [d]} \frac{|\widetilde{m}_0|_j}{\sqrt{[\widetilde{v}_t]_j} + \tau} \lesssim \mathcal{O}(1).$$

Bounding $F_5$ and $F_6$ is more complex. We begin by noting that

$$\begin{aligned}
|\mathbb{E}[B_{1,q}]| &\leq \sum_{j=1}^d \frac{G(1-\widetilde{\beta}_2)\widetilde{\beta}_2^{\frac{q}{2}}}{\tau^2} \cdot \mathbb{E}\left[\frac{[\Delta_{t-q}^2 |\Delta_r|]_j}{\sqrt{[\widetilde{v}_{t-q}]_j}}\right] \\
&\leq \sum_{j=1}^d \frac{G(1-\widetilde{\beta}_2)\widetilde{\beta}_2^{\frac{q}{2}}}{\tau^2} \cdot \mathbb{E}\left[\frac{[\Delta_{t-q}^2 |\Delta_r|]_j}{\sqrt{\max\{[\widetilde{\beta}_2^{t-q}\widetilde{v}_0 + (1-\widetilde{\beta}_2)\sum_{o=1}^{t-q} \widetilde{\beta}_2^{t-q-o}\Delta_o^2]_j, \tau^2\}}}\right] \\
&\lesssim \sum_{j=1}^d \frac{(1-\widetilde{\beta}_2)\widetilde{\beta}_2^{\frac{q}{2}}}{\tau^3} \cdot \mathbb{E}\left[[\Delta_{t-q}^2 |\Delta_r|]_j\right].
\end{aligned}$$

Therefore,

$$\begin{aligned}
F_5 F_6 &\lesssim (1-\widetilde{\beta}_1) \sum_{t=1}^T \eta_t \sum_{r=1}^t \widetilde{\beta}_1^{t-r}(1-\widetilde{\beta}_2) \sum_{q=0}^{t-r} \widetilde{\beta}_2^{\frac{q}{2}} \cdot \mathbb{E}\left[\Delta_{t-q}^2 |\Delta_r|\right] \\
&\leq (1-\widetilde{\beta}_1) \sum_{t=1}^T \eta_t \sum_{r=1}^t \widetilde{\beta}_1^{t-r}(1-\widetilde{\beta}_2)\eta_\ell^r u_r \sum_{q=0}^{t-r} \widetilde{\beta}_2^{\frac{q}{2}} (\eta_\ell^{t-q} u_{t-q})^2.
\end{aligned}$$

Under the substitution $q \leftarrow t - \widetilde{q}$, we have that

$$\begin{aligned}
F_5 F_6 &\lesssim (1-\widetilde{\beta}_1) \sum_{t=1}^T \eta_t \sum_{r=1}^t \widetilde{\beta}_1^{t-r}(1-\widetilde{\beta}_2)\eta_\ell^r u_r \widetilde{\beta}_2^{\frac{t}{2}} \sum_{\widetilde{q}=r}^t \widetilde{\beta}_2^{\frac{-\widetilde{q}}{2}} (\eta_\ell^{\widetilde{q}} u_{\widetilde{q}})^2 \\
&\lesssim (1-\widetilde{\beta}_1) \sum_{t=1}^T \eta_t \sum_{r=1}^t \widetilde{\beta}_1^{t-r}(1-\widetilde{\beta}_2)\eta_\ell^r u_r \cdot 2^{\nu+\zeta} \frac{t^{2(\nu+\zeta)}}{|\log_e(\widetilde{\beta}_2)|} \\
&\lesssim (1-\widetilde{\beta}_1) \sum_{t=1}^T \frac{t^{\omega+2(\nu+\zeta)}}{|\log_e(\widetilde{\beta}_2)|} \widetilde{\beta}_1^t (1-\widetilde{\beta}_2) \sum_{r=1}^t \widetilde{\beta}_1^{-r} r^{\nu+\zeta} \\
&\lesssim (1-\widetilde{\beta}_1) \sum_{t=1}^T (1-\widetilde{\beta}_2) \frac{t^{\omega+3(\nu+\zeta)}}{|\log_e(\widetilde{\beta}_1)||\log_e(\widetilde{\beta}_2)|} \approx \frac{(1-\widetilde{\beta}_1)(1-\widetilde{\beta}_2)}{|\log_e(\widetilde{\beta}_1)||\log_e(\widetilde{\beta}_2)|} \cdot \max\left\{\mathcal{O}(1), T^{\omega+3(\nu+\zeta)+1}\right\}.
\end{aligned}$$

As $\mathcal{O}(1)$ terms are subsumed by $F_4$, $F_5 F_7$ is bounded via

$$\begin{aligned}
(1-\widetilde{\beta}_1) \sum_{t=1}^T \eta_t \sum_{r=1}^t \widetilde{\beta}_1^{t-r} \frac{L\eta_\ell^r u_r K \operatorname{diam}(\mathcal{X})}{\tau} &\lesssim (1-\widetilde{\beta}_1) \sum_{t=1}^T \eta_t \sum_{r=1}^t \widetilde{\beta}_1^t \frac{\eta_\ell^r u_r \widetilde{\beta}_1^{-r}}{\tau} \\
&\lesssim (1-\widetilde{\beta}_1) \sum_{t=1}^T \frac{t^{\nu+\zeta+\omega}}{\tau|\log_e(\widetilde{\beta}_1)|} \lesssim \frac{(1-\widetilde{\beta}_1)T^{\omega+\nu+\zeta+1}}{\tau|\log_e(\widetilde{\beta}_1)|}.
\end{aligned}$$

The remaining terms may also be bounded as follows:

$$\begin{aligned}
F_5 F_8 &\lesssim \frac{(1-\widetilde{\beta}_1)}{\tau} \sum_{t=1}^T \eta_t \sum_{r=1}^t \widetilde{\beta}_1^{t-r} \eta_\ell^r d_r \lesssim \frac{(1-\widetilde{\beta}_1)}{\tau} \sum_{t=1}^T \eta_t \sum_{r=1}^t \widetilde{\beta}_1^t \widetilde{\beta}_1^{-r} r^{\nu+\gamma} \\
&\lesssim \frac{(1-\widetilde{\beta}_1)}{|\log_e(\widetilde{\beta}_1)|} \sum_{t=1}^T t^\omega t^{\nu+\gamma} \lesssim \frac{(1-\widetilde{\beta}_1)}{|\log_e(\widetilde{\beta}_1)|} \max\{T^{\omega+\nu+\gamma+1}, \mathcal{O}(1)\}
\end{aligned}$$

where $F_9$ and $F_{10}$ can be bounded via

$$F_5 F_9 \lesssim (1-\widetilde{\beta}_1) \sum_{t=1}^T \sum_{r=1}^t \frac{\eta_t \widetilde{\beta}_1^{t-r}(\eta_\ell^r)^2 u_r}{\tau} \lesssim \frac{(1-\widetilde{\beta}_1)}{|\log_e(\widetilde{\beta}_1)|} \sum_{t=1}^T \sum_{r=1}^t \frac{\eta_t \widetilde{\beta}_1^{t-r} r^{2\nu+\zeta}}{\tau} \lesssim \frac{T^{2\nu+\zeta+1+\omega}}{\tau},$$

$$F_5 F_{10} \lesssim (1 - \widetilde{\beta}_1) \sum_{t=1}^{T} \sum_{r=1}^{t} \eta_t \widetilde{\beta}_1^{t-r} \frac{\eta_\ell^r u_r^{1-\alpha}}{\tau} \lesssim (1 - \widetilde{\beta}_1) \sum_{t=1}^{T} \sum_{r=1}^{t} t^\omega \widetilde{\beta}_1^t \frac{\widetilde{\beta}_1^{-r} r^{\nu + \zeta(1-\alpha)}}{\tau}$$

$$\lesssim \sum_{t=1}^{T} t^\omega \frac{(1 - \widetilde{\beta}_1)}{|\log_e(\widetilde{\beta}_1)|} t^{\nu + \zeta(1-\alpha)} \lesssim \frac{(1 - \widetilde{\beta}_1)}{|\log_e(\widetilde{\beta}_1)|} T^{\omega + \nu + \zeta(1-\alpha) + 1}.$$

Standard calculations imply that under the conditions equation 11, the dominating terms are $F_7$, $F_{10}$ with order $\mathcal{O}(1)$. Within the derived upper bound, $\zeta > 0$ destabilizes $F_7$ and decays $F_{10}$ to 0, while $\zeta < 0$ gives the analogous properties with $F_7$ and $F_{10}$ swapped. $\qquad\square$

## E  EXPERIMENT SETUP & FULL RESULTS

In this section, we present the experimental setups and results across two primary domains: synthetic data and natural language processing tasks. More precisely, we evaluate the performance of TailOPT instantiations with state-of-the-art benchmarks on convex models (with synthetic data), transformer encoders, as well as generative models. For convex, synthetic experiments, we construct datasets to emulate heavy-tailed stochastic gradients, focusing on linear regression models trained under contaminated label noise. The design includes generating feature matrices and labels while injecting noise from heavy-tailed distributions to study convergence behaviors. Additionally, we introduce the **SynToken** dataset, which models the heavy-tailed distribution of token frequencies observed in natural language processing. For brevity, we only include the results of the SynToken dataset, denoted 'Synthetic data', in the main text (Figure 1). This allows us to evaluate learning algorithms in controlled settings, easing out and exploring the effects of both common and rare features.

For assessing the optimization of transformer encoders on natural language processing tasks, we evaluate RoBERTa Liu et al. (2019) on the General Language Understanding Evaluation (GLUE) benchmark Wang et al. (2019), which encompasses a diverse range of tasks such as sentiment analysis, paraphrase detection, and natural language inference. By fine-tuning RoBERTa on GLUE, we assess its generalization capabilities and robustness. The benchmark's inclusion of multiple datasets ensures a comprehensive evaluation of model performance across various linguistic phenomena. Additionally, we also evaluate the capabilities of the T5 Raffel et al. (2020) generative model on WMT machine translation tasks Foundation (2019). These experiments provide insights into the behavior of optimization algorithms and pretrained models under realistic and challenging conditions. For RoBERTa, we optimize over GLUE across 10 simulated compute nodes, whereas for T5, we model 3 compute node fine-tuning on WMT benchmark datasets.

**Compute Resources.** We conducted our experiments on a compute cluster equipped with dozens of GPUs, with dynamic availability fluctuating based on overall cluster usage by other users. The cluster featured a set of GPU models, including H100, L40S, and A40 machines.

### E.1  CONVEX MODELS (SYNTHETIC EXPERIMENTS)

#### E.1.1  DATA GENERATION PROCESS

To simulate heavy-tailed stochastic gradients in a simple yet controlled linear regression setting, we generated a synthetic dataset as follows. The feature matrix $X \in \mathbb{R}^{M \times m}$ was constructed with entries drawn independently from a standard normal distribution, $X_{ij} \sim \mathcal{N}(0, 1)$. The true weight vector $w_{\text{true}} \in \mathbb{R}^m$ was sampled from $\mathcal{N}(0, I_m)$, where $I_m$ is the $m \times m$ identity matrix.

The true labels were computed using:

$$y_{\text{true}} = X w_{\text{true}}.$$

To induce heavy-tailed stochastic gradients, we injected noise into the label vector by adding a noise term $\xi$, resulting in contaminated labels:

$$\hat{y} = y_{\text{true}} + \xi,$$

where $\xi \in \mathbb{R}^M$ is a noise vector with entries drawn independently from a heavy-tailed distribution $\mathcal{D}$. For simplicity, we assume coordinate-wise independence of the noise components.

After generating the dataset, we distributed the data across $n = 10$ data centers in an IID fashion. Notably, the heavy-tailed noise was injected once prior to distribution, and no additional data were

generated afterward. This approach ensured that the same contaminated training data are used locally throughout the training process.

### E.1.2 Linear Regression Model

We consider a single-layer neural network without biases, parameterized by $w \in \mathbb{R}^m$, which is equivalent to linear regression. Training is performed using the contaminated labels $(X, \hat{y})$ with the mean-squared error (MSE) loss function:

$$\mathcal{L}(w) = \frac{1}{2}\|\hat{y} - Xw\|^2.$$

The gradient of the loss with respect to $w$ is given by:

$$\nabla_w \mathcal{L}(w) = -X^\top(\hat{y} - Xw).$$

Substituting $\hat{y} = y_{\text{true}} + \xi = Xw_{\text{true}} + \xi$, we have:

$$\nabla_w \mathcal{L}(w) = -X^\top(Xw_{\text{true}} + \xi - Xw) = -X^\top X(w_{\text{true}} - w) - X^\top \xi.$$

Simplifying, we obtain:

$$\nabla_w \mathcal{L}(w) = X^\top X(w - w_{\text{true}}) - X^\top \xi.$$

The term $-X^\top \xi$ reflects the influence of the heavy-tailed noise on the gradient. Given that $X$ has Gaussian entries and $\xi$ follows a heavy-tailed distribution, the stochastic gradients $\nabla_w \mathcal{L}(w)$ are also heavy-tailed.

### E.1.3 The SynToken Dataset

To model the heavy-tailed nature of token frequencies observed in natural language processing, we created the synthetic **SynToken** dataset. In natural language, word or token usage often follows a heavy-tailed distribution. That is, a small number of tokens appear very frequently, while a large number of tokens appear infrequently but carry significant contextual information.

In our dataset, we partitioned the feature space into common and rare features to reflect this phenomenon. Specifically, we designated the first $p = 10\%$ of the columns of $X$ as common features and the remaining $90\%$ as rare features. The common features were generated by sampling from a Bernoulli distribution with a high probability of success:

$$X_{\text{common}} \sim \text{Bernoulli}(0.9),$$

resulting in features that are frequently active. The rare features were sampled from a Bernoulli distribution with a low probability of success:

$$X_{\text{rare}} \sim \text{Bernoulli}(0.1),$$

introducing sparsity and emulating infrequently occurring tokens.

The complete feature matrix $X$ was formed by concatenating $X_{\text{common}}$ and $X_{\text{rare}}$:

$$X = [X_{\text{common}},\ X_{\text{rare}}].$$

The weight vector $w$ was sampled from a standard multivariate normal distribution, $w \sim \mathcal{N}(0, I_m)$, consistent with the previous setup. Noise injection was analogously applied to the labels as before. This approach was taken to mimic the key characteristics of tokenization and word embeddings in natural language processing, via a minimal yet effective model. One benefit of synthetic datasets is that by simulating the distribution of common and rare tokens, the **SynToken** dataset allows us to study the effects of heavy-tailed data distributions on learning algorithms in a controlled setting. Additionally, we note that the problem being studied is $\mu$-strongly convex with probability 1, as the setting is linear regression under Gaussian features.

### E.2 Synthetic Experiments Discussion

**Does the heavy-tailed distribution of covariates matter?** Figure 3 (a) and (c) illustrate that a heavy-tailed distribution of token frequencies has significant impacts on the performance of optimization strategies. In (a), RMSProp-$BiClip$ performs competitively under standard tokenization.

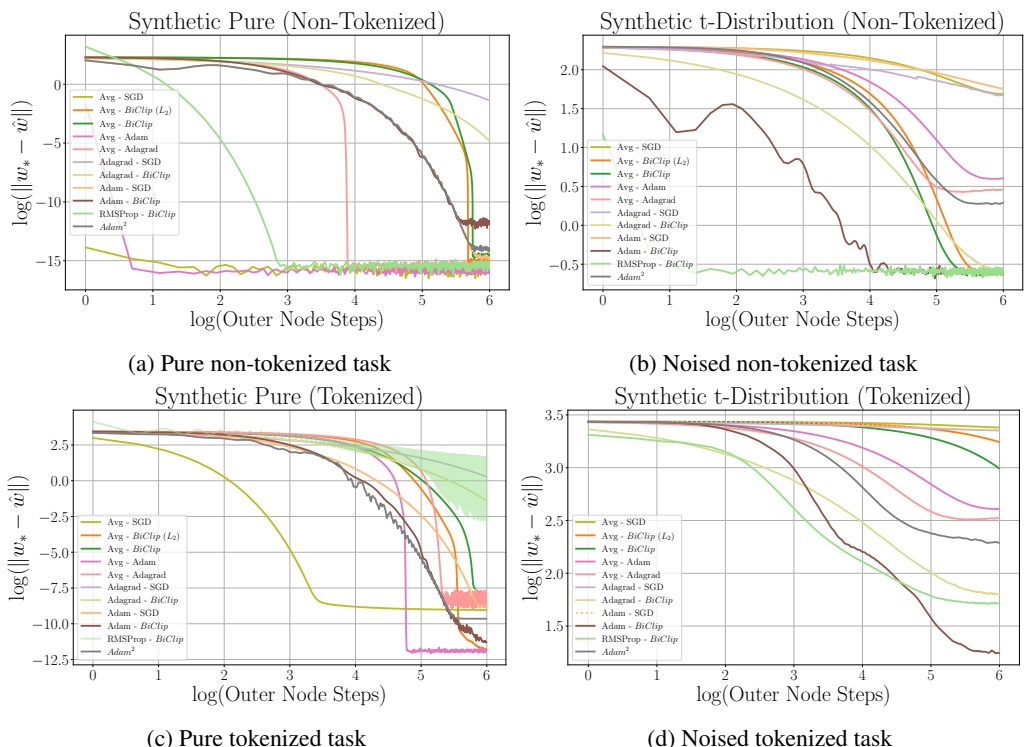

(a) Pure non-tokenized task

(b) Noised non-tokenized task

(c) Pure tokenized task

(d) Noised tokenized task

Figure 3: (Top) The results on the non-tokenized synthetic dataset are presented. In the absence of noise injection, Avg-Adam, Avg-SGD, and RMSProp-$BiClip$ demonstrate the most competitive performance. However, under heavy-tailed noise injection, RMSProp-$BiClip$ and Adam-$BiClip$ achieve the highest performance, while Avg-SGD exhibits among the poorest outcomes. Notably, oscillations observed in Adam-$BiClip$ may reflect the impact of amplified update step sizes in the outer optimizer, potentially enabling finer-grained exploration of the optimization landscape. (Bottom) Tokenization drastically alters algorithmic performance. Without noise, Avg-SGD decays the fastest, while Avg-Adam converges to a superior optimum. However, when synthetic, unbiased heavy-tailed noise is introduced, Avg-SGD becomes highly unstable, whereas Adam-$BiClip$ and RMSProp-$BiClip$ consistently deliver the best results.

However, in (c), heavy-tailed tokenization applied to the feature matrix destabilizes RMSProp-$BiClip$. Interestingly, under tokenized conditions without noise, RMSProp exhibits oscillatory behavior, whereas Adam maintains relative stability. This is consistent with the interpretation of Adam as incorporating an exponentially decaying moving average of the gradient's first moment, which augments optimization stability. Upon noise injection, best performing hyperparameters for RMSProp-$BiClip$ does not show oscillatory behavior, but is larger in terms of distance $\|w_* - \hat{w}\|$ than the case without noise.

**Does noise matter?** When noise is injected into the labels, the performance dynamics shift considerably. outer optimizer adaptive or non-adaptive methods *combined* with inner optimizer SGD perform poorly, which may indicate that inner optimizers should take a focal role in addressing the challenges posed by heavy-tailed noise. While the choice of the outer optimizer may appear to a limited impact on the binary question of learnability for this specific synthetic data (i.e., "Can the algorithm decrease distance to the true $w_*$ or not?"), under tokenized conditions with heavy-tailed noise (Figure 3(d)), outer optimizer Adam demonstrates the best performance. Figure 3 reveals that heavy-tailed noise generally destabilizes all algorithms, including adaptive methods, clipped approaches, and pure SGD (c.f., minimum values in (a) and (c) to (b) and (d)). Notably, coordinate-wise $BiClip$ consistently outperforms $L_2$ clipping, aligning with the results in Table 1.

**How far should these results generalize?** A word of caution is warranted against overgeneralization. These results are derived from a simplified regression model, limiting the ability to generalize the observed trends. Nevertheless, the experiments underscore the pronounced effects of heavy-tailed

noise in a controlled synthetic environment and highlight the noise-mitigating capabilities of optimizers such as Adam, RMSProp, and $BiClip$. Additionally, it is important to note that real-world transformer models often comprise tens of millions to billions of parameters.

### E.3 TRANSFORMER ENCODERS (ROBERTA & GLUE BENCHMARKS)

The General Language Understanding Evaluation (GLUE) benchmark Wang et al. (2019) serves as a comprehensive framework for evaluating natural language understanding (NLU) models across a diverse range of tasks. By incorporating datasets that span various linguistic challenges, GLUE provides a rigorous testbed for assessing the generalization capabilities of NLP models. Below, we summarize the datasets and tasks included in GLUE:

CoLA (Corpus of Linguistic Acceptability): A binary classification task that evaluates a model's ability to determine whether a given sentence is grammatically acceptable. Sentences are drawn from linguistic theory literature, with performance measured by the Matthews Correlation Coefficient (MCC). We fine-tune for 15 epochs (15 outer optimizer steps, where each inner optimizer performs 1 epoch on their allocated data).

SST-2 (Stanford Sentiment Treebank): This binary sentiment analysis task involves classifying movie reviews as expressing positive or negative sentiment. Accuracy is the primary evaluation metric. We fine-tune for 5 epochs.

MRPC (Microsoft Research Paraphrase Corpus): A paraphrase detection task where the goal is to identify whether two sentences, often drawn from news sources, have equivalent meanings. Performance is evaluated using both accuracy and F1 score. We fine-tune for 30 epochs.

STS-B (Semantic Textual Similarity Benchmark): A regression task that assesses the semantic similarity between two sentences on a continuous scale from 0 (unrelated) to 5 (identical in meaning). The dataset combines multiple sources, with evaluation based on Pearson and Spearman correlations. We fine-tune for 10 epochs.

QQP (Quora Question Pairs): Another paraphrase detection task, QQP focuses on identifying whether pairs of questions from the Quora platform are semantically equivalent. Metrics include accuracy and F1 score. We fine-tune for 5 epochs.

MNLI (Multi-Genre Natural Language Inference): A three-class classification task (entailment, neutral, contradiction) that evaluates a model's ability to perform natural language inference across multiple genres, including fiction, government reports, and spoken dialogue. We fine-tune for 7 epochs.

QNLI (Question Natural Language Inference): Adapted from the Stanford Question Answering Dataset (SQuAD), this binary classification task assesses whether a given sentence provides a valid answer to a question. We fine-tune for 10 epochs.

RTE (Recognizing Textual Entailment): Similar to MNLI but on a smaller scale, this binary classification task involves determining whether a hypothesis logically follows from a given premise. Data sources include news articles and Wikipedia. We fine-tune for 30 epochs.

WNLI (Winograd Natural Language Inference): A specialized task focusing on pronoun resolution in sentences. The dataset is based on the Winograd Schema Challenge, where resolving pronouns requires understanding contextual nuances. We note that it is standard to exclude the evaluation of WNLI when reporting GLUE results, due to the intrinsically adversarial nature of the dataset (i.e., validation data are constructed as subtle perturbations applied to the training data with opposite labels) Raffel et al. (2020).

**RobERTa.** RoBERTa is a state-of-the-art transformer-based model designed to enhance the performance of the original BERT architecture through improved pretraining strategies. Proposed by Liu et al. (2019), RoBERTa optimizes BERT by refining its training setup, enabling more robust natural language understanding (NLU) across diverse tasks. Key innovations introduced by RoBERTa include the removal of the next sentence prediction (NSP) objective, an increase in batch sizes and training data, and the use of longer training schedules. Additionally, RoBERTa employs dynamic masking during training, which prevents models from overfitting to static token masks.

Trained on significantly larger datasets (e.g., the BooksCorpus, CC-News, and OpenWebText), RoBERTa achieves superior performance on several benchmarks, including GLUE, SuperGLUE, and SQuAD. Its flexibility and robustness make it particularly effective for fine-tuning on a wide range of downstream tasks, from sentiment analysis to question answering. By refining BERT's pretraining process, RoBERTa underscores the importance of hyperparameter tuning and data utilization in achieving state-of-the-art results.

### E.4 GENERATIVE MODELS (T5 & WMT DATASET BENCHMARKS)

We additionally evaluate our method using T5 Raffel et al. (2020), a state-of-the-art text-to-text transformer model developed by Google Research. T5 unifies natural language processing tasks under a text-to-text framework, where both inputs and outputs are text strings, making it highly versatile across tasks such as summarization, translation, and classification. The model was pretrained on the Colossal Clean Crawled Corpus (C4) using a span corruption objective and is available in multiple sizes, ranging from T5-Small (60M parameters) to T5-XXL (11B parameters). This unified framework and scalability allow T5 to excel in a wide range of tasks, making it a strong baseline for evaluating our proposed method.

To evaluate machine translation tasks, we utilize the WMT datasets, a widely recognized benchmark for translation research Foundation (2019). Specifically, we fine-tune T5 on the TED Talks and News Commentary datasets. The TED Talks dataset, originally sourced from IWSLT 2017 Cettolo et al. (2017), provides multilingual translations of TED Talk transcripts, offering diverse linguistic and domain-specific challenges. In contrast, the News Commentary dataset contains parallel text derived from news articles in various languages, presenting a more formal and structured domain. These datasets represent distinct styles and linguistic features, providing a rigorous evaluation of algorithm agility in optimizing across various domains or tasks.

### E.5 HYPERPARAMETER SWEEP GRID

The sweep grids in Tables 3, 4 were determined by first performing a coarser sweep using an approximate grid, then localizing near the discovered well-performing hyperparameters.

Table 3: Hyperparameter Sweeps: Gradient Clipping Parameters. `i_u`, `i_d` = inner optimizer $u$, $d$, `o_u`, `o_d` = outer optimizer $u$, $d$.

| Algorithm | i_u | i_d | o_u | o_d |
|---|---|---|---|---|
| **Avg-SGD** | - | - | - | - |
| **Avg-$L_2$Clip SGD** | np.linspace($10^{-4}$, 1.5, 12) | 0.0 | - | - |
| **Avg-$BiClip$** | np.linspace($10^{-4}$, 1.5, 4) | np.linspace($10^{-7}$, i_u, 4) | - | - |
| **Avg-$BiClip$ ($L_2$)** | np.linspace($10^{-4}$, 1.5, 4) | np.linspace($10^{-7}$, i_u, 4) | - | - |
| **Avg-Adagrad** | - | - | - | - |
| **Avg-Adam** | - | - | - | - |
| **Adagrad-SGD** | - | - | - | - |
| **RMSProp-SGD** | - | - | - | - |
| **Adam-SGD** | - | - | - | - |
| **Adagrad-$BiClip$** | np.linspace($10^{-4}$, 1.5, 3) | np.linspace($10^{-7}$, i_u, 3) | - | - |
| **RMSProp-$BiClip$** | np.linspace($10^{-4}$, 1.5, 3) | np.linspace($10^{-7}$, i_u, 3) | - | - |
| **Adam-$L_2$Clip** | np.linspace($10^{-4}$, 1.5, 12) | 0.0 | - | - |
| **Adam-$BiClip$** | np.logspace($-2$, 1, 5) | np.linspace($10^{-7}$, i_u, 3) | - | - |
| **Adam-$BiClip$ ($L_2$)** | np.linspace($10^{-4}$, 1.5, 3) | np.linspace($10^{-7}$, i_u, 3) | - | - |
| $Adam^2$ | - | - | - | - |
| $Bi^2Clip$ **(Coordinate-wise)** | np.linspace($10^{-4}$, 1.5, 3) | np.linspace($10^{-7}$, i_u, 3) | np.linspace($10^{-4}$, 1.5, 3) | np.linspace($10^{-7}$, o_u, 3) |
| $Bi^2Clip$ ($L_2$) | np.logspace($-1$, 0.5, 3) | np.linspace($10^{-7}$, i_u, 3) | np.logspace($-1$, 0.5, 3) | np.linspace($10^{-7}$, o_u, 3) |
| **DiLoCo** | - | - | - | - |

### E.6 OPTIMAL HYPERPARAMETERS

In this subsection, we display the optimal hyperparameters located during our extensive sweep. For readability, we report the results as Tables 6-9.

Table 4: Hyperparameter Sweeps: Learning Rates and Adaptivity Parameters. `ilr` = inner optimizer learning rate, `olr` = outer optimizer learning rate, `ieps` = inner optimizer $\varepsilon$, `oeps` = outer optimizer $\varepsilon$. Additionally, DiLoCo swept over the nesterov learning rates $(0.9, 0.95)$, and inner optimizer weight decay parameters $(10^{-1}, 10^{-4})$, reported in prior works such as Douillard et al. (2024); Huo et al. (2020).

| Algorithm | ilr | olr | ieps | oeps |
|---|---|---|---|---|
| **Avg-SGD** | np.logspace$(-9, 1, 100)$ | - | - | - |
| **Avg-$L_2$Clip SGD** | np.linspace$(10^{-9}, 1, 10)$ | - | - | - |
| **Avg-$BiClip$** | np.linspace$(10^{-9}, 1, 10)$ | - | - | - |
| **Avg-$BiClip$ ($L_2$)** | np.linspace$(10^{-9}, 1, 10)$ | - | - | - |
| **Avg-Adagrad** | np.linspace$(10^{-9}, 1, 30)$ | - | $\{10^{-8}, 10^{-6}, 10^{-4}, 10^{-3}\}$ | - |
| **Avg-Adam** | np.linspace$(10^{-9}, 1, 30)$ | - | $\{10^{-8}, 10^{-6}, 10^{-4}, 10^{-3}\}$ | - |
| **Adagrad-SGD** | np.linspace$(10^{-5}, 0.1, 7)$ | np.logspace$(-5, -1, 7)$ | - | $\{10^{-7}, 10^{-5}, 10^{-3}\}$ |
| **RMSProp-SGD** | np.linspace$(10^{-5}, 0.1, 7)$ | np.linspace$(10^{-5}, 0.1, 7)$ | - | $\{10^{-7}, 10^{-5}, 10^{-3}\}$ |
| **Adam-SGD** | np.linspace$(10^{-5}, 0.1, 7)$ | np.logspace$(-5, -1, 7)$ | - | $\{10^{-7}, 10^{-5}, 10^{-3}\}$ |
| **Adagrad-$BiClip$** | np.linspace$(10^{-5}, 0.1, 4)$ | np.logspace$(-5, -1, 4)$ | - | $\{10^{-7}, 10^{-5}, 10^{-3}\}$ |
| **RMSProp-$BiClip$** | np.linspace$(10^{-5}, 0.1, 4)$ | np.logspace$(-5, -1, 4)$ | - | $\{10^{-7}, 10^{-5}, 10^{-3}\}$ |
| **Adam-$L_2$Clip** | np.linspace$(10^{-5}, 0.1, 4)$ | np.linspace$(10^{-5}, 0.1, 4)$ | - | $\{10^{-7}, 10^{-5}, 10^{-3}\}$ |
| **Adam-$BiClip$** | np.logspace$(-6, -1, 5)$ | np.logspace$(-6, -1, 5)$ | - | $\{10^{-7}, 10^{-5}, 10^{-3}\}$ |
| **Adam-$BiClip$ ($L_2$)** | np.linspace$(10^{-5}, 0.1, 4)$ | np.linspace$(10^{-5}, 0.1, 4)$ | - | $\{10^{-7}, 10^{-5}, 10^{-3}\}$ |
| $Adam^2$ | np.logspace$(-6, -1, 5)$ | np.logspace$(-6, -1, 5)$ | $\{10^{-7}, 10^{-5}, 10^{-3}\}$ | $\{10^{-7}, 10^{-5}, 10^{-3}\}$ |
| $Bi^2Clip$ (**Coordinate-wise**) | np.linspace$(10^{-9}, 1, 3)$ | np.linspace$(10^{-9}, 1, 3)$ | - | - |
| $Bi^2Clip$ ($L_2$) | np.logspace$(-1, 0.5, 3)$ | np.logspace$(-1, 0.5, 3)$ | - | - |
| **DiLoCo** | np.logspace$(-5, -1, 5)$ | $\{1, 0.7, 0.5, 10^{-1}, 10^{-2}\}$ | - | $\{10^{-7}, 10^{-5}, 10^{-3}\}$ |

Table 5: Best hyperparameter selection over a sweep of various parameter grids. 'ilr' = inner optimizer learning rate, 'olr' = outer optimizer learning rate, 'ieps' = inner optimizer $\varepsilon$, 'oeps' = outer optimizer $\varepsilon$, 'o_u', 'o_d' = outer optimizer $u$, $d$, 'i_u', 'i_d' = inner optimizer $u$, $d$. Here, $\varepsilon$ is the adaptivity or $\varepsilon$-smoothing parameter employed in the denominator of adaptive optimizers to enhance stability of learning dynamics.

| Algorithm | Dataset | ilr | olr | ieps | oeps | o_u | o_d | i_u | i_d |
|---|---|---|---|---|---|---|---|---|---|
| **Avg-SGD** | STS-B | 0.019 | - | - | - | - | - | - | - |
| | RTE | 0.095 | - | - | - | - | - | - | - |
| | QNLI | 0.0059 | - | - | - | - | - | - | - |
| | QQP | 0.0074 | - | - | - | - | - | - | - |
| | CoLA | 0.019 | - | - | - | - | - | - | - |
| | SST-2 | 0.0074 | - | - | - | - | - | - | - |
| | MRPC | 0.038 | - | - | - | - | - | - | - |
| | MNLI | 0.0059 | - | - | - | - | - | - | - |
| **Avg-$L_2$Clip** | STS-B | 0.56 | - | - | - | - | - | 1.5 | 0.0 |
| | RTE | 1 | - | - | - | - | - | 0.14 | 0.0 |
| | QNLI | 0.33 | - | - | - | - | - | 0.14 | 0.0 |
| | QQP | 0.44 | - | - | - | - | - | 0.14 | 0.0 |
| | CoLA | 0.33 | - | - | - | - | - | 0.14 | 0.0 |
| | SST-2 | 0.11 | - | - | - | - | - | 0.27 | 0.0 |
| | MRPC | 0.22 | - | - | - | - | - | 0.41 | 0.0 |
| | MNLI | 0.11 | - | - | - | - | - | 0.41 | 0.0 |
| **Avg-$BiClip$** | STS-B | 0.44 | - | - | - | - | - | 0.0001 | 0.0001 |
| | RTE | 1 | - | - | - | - | - | 0.0001 | 6.7e-5 |
| | QNLI | 0.44 | - | - | - | - | - | 0.0001 | 6.7e-5 |
| | QQP | 0.56 | - | - | - | - | - | 0.0001 | 3.3e-5 |
| | CoLA | 0.89 | - | - | - | - | - | 0.0001 | 0.0001 |
| | SST-2 | 0.56 | - | - | - | - | - | 0.0001 | 6.7e-5 |
| | MRPC | 0.89 | - | - | - | - | - | 0.0001 | 6.7e-5 |
| | MNLI | 0.56 | - | - | - | - | - | 0.0001 | 3.3e-5 |
| **Avg-$BiClip$ ($L_2$)** | STS-B | 0.067 | - | - | - | - | - | 0.75 | 0.75 |
| | RTE | 1 | - | - | - | - | - | 0.0001 | 6.7e-5 |
| | QNLI | 0.067 | - | - | - | - | - | 0.75 | 0.75 |
| | QQP | 0.11 | - | - | - | - | - | 0.5 | 0.33 |
| | CoLA | 0.067 | - | - | - | - | - | 0.75 | 0.75 |
| | SST-2 | 0.1 | - | - | - | - | - | 0.75 | 0.38 |
| | MRPC | 0.11 | - | - | - | - | - | 1 | 1 |
| | MNLI | 0.033 | - | - | - | - | - | 1.5 | 1.5 |
| $Bi^2Clip$ | STS-B | 0.5 | 0.5 | - | - | 0.0001 | 0.0001 | 0.0001 | 1e-7 |
| | RTE | 1 | 1 | - | - | 0.0001 | 0.0001 | 0.001 | 5e-5 |
| | QNLI | 0.5 | 1 | - | - | 0.0001 | 0.0001 | 0.0001 | 5e-5 |
| | QQP | 0.5 | 1 | - | - | 1.5 | 1e-7 | 0.0001 | 5e-5 |
| | CoLA | 0.5 | 1 | - | - | 0.0001 | 0.0001 | 0.0001 | 0.0001 |
| | SST-2 | 0.5 | 1 | - | - | 0.75 | 1e-7 | 0.0001 | 1e-7 |
| | MRPC | 1 | 1 | - | - | 0.0001 | 0.0001 | 0.0001 | 1e-7 |
| | MNLI | 0.5 | 1 | - | - | 0.75 | 1e-7 | 0.0001 | 1e-7 |
| $Bi^2Clip$ ($L_2$) | STS-B | 0.56 | 3.2 | - | - | 0.1 | 0.05 | 0.1 | 0.05 |
| | RTE | 0.1 | 0.56 | - | - | 0.1 | 0.1 | 0.56 | 0.56 |
| | QNLI | 0.1 | 0.1 | - | - | 3.2 | 3.2 | 0.56 | 1e-7 |
| | QQP | 0.1 | 3.2 | - | - | 0.56 | 1e-7 | 0.56 | 0.56 |
| | CoLA | 0.1 | 3.2 | - | - | 0.1 | 0.05 | 0.56 | 1e-7 |
| | SST-2 | 0.56 | 0.1 | - | - | 3.2 | 3.2 | 0.1 | 1e-7 |
| | MRPC | 0.56 | 0.1 | - | - | 0.56 | 0.56 | 0.1 | 0.1 |
| | MNLI | 0.1 | 0.56 | - | - | 3.2 | 1.6 | 0.56 | 1e-7 |

Table 6: Best hyperparameter selection over a sweep of various parameter grids. 'ilr' = inner optimizer learning rate, 'olr' = outer optimizer learning rate, 'ieps' = inner optimizer $\varepsilon$, 'oeps' = outer optimizer $\varepsilon$, 'o_u', 'o_d' = outer optimizer $u$, $d$, 'i_u', 'i_d' = inner optimizer $u$, $d$. Here, $\varepsilon$ is the adaptivity or $\varepsilon$-smoothing parameter employed in the denominator of adaptive optimizers to enhance stability of learning dynamics.

| Algorithm | Dataset | ilr | olr | ieps | oeps | o_u | o_d | i_u | i_d |
|---|---|---|---|---|---|---|---|---|---|
| **Adam-SGD** | STS-B | 0.017 | 4.6e-5 | - | 1e-7 | - | - | - | - |
| | RTE | 0.033 | 4.6e-5 | - | 1e-7 | - | - | - | - |
| | QNLI | 0.017 | 2.2e-4 | - | 1e-7 | - | - | - | - |
| | QQP | 0.017 | 2.2e-4 | - | 1e-7 | - | - | - | - |
| | CoLA | 0.033 | 0.001 | - | 1e-5 | - | - | - | - |
| | SST-2 | 0.017 | 4.6e-5 | - | 1e-7 | - | - | - | - |
| | MRPC | 0.017 | 4.6e-5 | - | 1e-7 | - | - | - | - |
| | MNLI | 0.017 | 2.2e-4 | - | 1e-7 | - | - | - | - |
| **Adam-**$L_2Clip$ | STS-B | 0.067 | 0.033 | - | 0.001 | - | - | 0.75 | 0.0 |
| | RTE | 0.033 | 1e-5 | - | 1e-7 | - | - | 1.5 | 0.0 |
| | QNLI | 0.067 | 0.067 | - | 0.001 | - | - | 0.75 | 0.0 |
| | QQP | 0.067 | 0.033 | - | 0.001 | - | - | 1.5 | 0.0 |
| | CoLA | 0.1 | 0.033 | - | 0.001 | - | - | 0.75 | 0.0 |
| | SST-2 | 0.1 | 0.033 | - | 0.001 | - | - | 1.5 | 0.0 |
| | MRPC | 0.033 | 0.033 | - | 0.001 | - | - | 0.75 | 0.0 |
| | MNLI | 0.067 | 0.033 | - | 0.001 | - | - | 0.75 | 0.0 |
| **Adam-**$BiClip$ | STS-B | 0.0056 | 3.2e-4 | - | 1e-5 | - | - | 0.01 | 0.0067 |
| | RTE | 3.2e-4 | 1.8e-5 | - | 1e-7 | - | - | 0.01 | 0.0067 |
| | QNLI | 0.0056 | 3.2e-4 | - | 1e-7 | - | - | 0.01 | 0.0067 |
| | QQP | 0.0056 | 0.00032 | - | 1e-7 | - | - | 0.01 | 0.0033 |
| | CoLA | 0.0056 | 1.8e-5 | - | 1e-7 | - | - | 0.01 | 0.01 |
| | SST-2 | 0.0056 | 1.8e-5 | - | 1e-7 | - | - | 0.01 | 0.0067 |
| | MRPC | 0.0056 | 0.0056 | - | 0.001 | - | - | 0.056 | 0.019 |
| | MNLI | 0.0056 | 3.2e-4 | - | 1e-5 | - | - | 0.01 | 0.0033 |
| **Adam-**$BiClip$ ($L_2$) | STS-B | 0.033 | 0.033 | - | 0.001 | - | - | 1.5 | 0.75 |
| | RTE | 0.033 | 0.067 | - | 0.001 | - | - | 0.75 | 0.38 |
| | QNLI | 0.033 | 0.067 | - | 0.001 | - | - | 1.5 | 0.75 |
| | QQP | 0.067 | 0.033 | - | 0.0001 | - | - | 0.75 | 0.38 |
| | CoLA | 0.033 | 0.033 | - | 0.001 | - | - | 1.5 | 0.75 |
| | SST-2 | 0.067 | 0.033 | - | 0.001 | - | - | 1.5 | 1e-7 |
| | MRPC | 0.033 | 0.033 | - | 0.001 | - | - | 1.5 | 1e-7 |
| | MNLI | 0.067 | 0.033 | - | 0.001 | - | - | 1.5 | 0.75 |
| $Adam^2$ | STS-B | 1.8e-5 | 1.8e-5 | 1e-5 | 1e-7 | - | - | - | - |
| | RTE | 1.8e-5 | 1.8e-5 | 1e-5 | 1e-7 | - | - | - | - |
| | QNLI | 1.8e-5 | 3.2e-4 | 1e-5 | 1e-5 | - | - | - | - |
| | QQP | 1.8e-5 | 3.2e-4 | 1e-5 | 1e-7 | - | - | - | - |
| | CoLA | 1.8e-5 | 0.0056 | 1e-5 | 0.001 | - | - | - | - |
| | SST-2 | 1.8e-5 | 1.8e-5 | 0.001 | 1e-7 | - | - | - | - |
| | MRPC | 1.8e-5 | 1.8e-5 | 1e-5 | 1e-7 | - | - | - | - |
| | MNLI | 1.8e-5 | 3.2e-4 | 1e-5 | 1e-7 | - | - | - | - |

Table 7: The notational setup is analogous to Table 6. For DiLoCo*, we provide the Nesterov learning rate and weight decay parameter in the i_u, i_d entries, respectively.

| Algorithm | Dataset | ilr | olr | ieps | oeps | o_u | o_d | i_u | i_d |
|---|---|---|---|---|---|---|---|---|---|
| **Adagrad-SGD** | STS-B | 0.017 | 0.0046 | - | 0.001 | - | - | - | - |
| | RTE | 0.033 | 0.001 | - | 1e-5 | - | - | - | - |
| | QNLI | 0.017 | 0.001 | - | 1e-5 | - | - | - | - |
| | QQP | 0.017 | 0.0001 | - | 1e-5 | - | - | - | - |
| | CoLA | 0.017 | 2.2e-4 | - | 1e-7 | - | - | - | - |
| | SST-2 | 0.017 | 2.2e-4 | - | 1e-5 | - | - | - | - |
| | MRPC | 0.017 | 2.2e-4 | - | 1e-7 | - | - | - | - |
| | MNLI | 0.017 | 0.0001 | - | 1e-7 | - | - | - | - |
| **RMSProp-SGD** | STS-B | 0.017 | 1e-5 | - | 1e-7 | - | - | - | - |
| | RTE | 0.017 | 1e-5 | - | 1e-7 | - | - | - | - |
| | QNLI | 0.033 | 0.001 | - | 1e-5 | - | - | - | - |
| | QQP | 0.017 | 1e-5 | - | 1e-7 | - | - | - | - |
| | CoLA | 0.017 | 1e-5 | - | 1e-7 | - | - | - | - |
| | SST-2 | 0.017 | 1e-5 | - | 1e-7 | - | - | - | - |
| | MRPC | 0.033 | 1e-5 | - | 1e-7 | - | - | - | - |
| | MNLI | 0.017 | 1e-5 | - | 1e-7 | - | - | - | - |
| **Adagrad-**$BiClip$ | STS-B | 1e-5 | 2.2e-4 | - | 1e-7 | - | - | 1.5 | 1.5 |
| | RTE | 0.033 | 2.2e-4 | - | 1e-7 | - | - | 1.5 | 1e-7 |
| | QNLI | 1e-5 | 0.0046 | - | 0.001 | - | - | 1.5 | 1.5 |
| | QQP | 1e-5 | 0.0046 | - | 0.0001 | - | - | 1.5 | 1.5 |
| | CoLA | 0.1 | 2.2e-4 | - | 1e-7 | - | - | 0.0001 | 5e-5 |
| | SST-2 | 1e-5 | 0.0046 | - | 0.001 | - | - | 1.5 | 1.5 |
| | MRPC | 1e-5 | 2.2e-4 | - | 1e-7 | - | - | 1.5 | 0.75 |
| | MNLI | 1e-5 | 0.0046 | - | 0.001 | - | - | 1.5 | 1.5 |
| **RMSProp-**$BiClip$ | STS-B | 1e-5 | 1e-5 | - | 1e-7 | - | - | 1.5 | 1.5 |
| | RTE | 0.067 | 1e-5 | - | 1e-7 | - | - | 0.0001 | 5e-5 |
| | QNLI | 0.1 | 1e-5 | - | 1e-7 | - | - | 0.0001 | 0.0001 |
| | QQP | 0.1 | 0.0046 | - | 1e-7 | - | - | 0.0001 | 5e-5 |
| | CoLA | 0.1 | 0.0046 | - | 0.001 | - | - | 0.0001 | 1e-7 |
| | SST-2 | 0.1 | 1e-5 | - | 1e-7 | - | - | 0.0001 | 0.0001 |
| | MRPC | 1e-5 | 0.0046 | - | 0.001 | - | - | 0.75 | 0.75 |
| | MNLI | 0.1 | 0.0046 | - | 0.001 | - | - | 0.0001 | 0.0001 |
| **DiLoCo*** | STS-B | 1.8e-5 | 0.7 | 1e-5 | - | - | - | 0.9 | 0.1 |
| | RTE | 1.8e-5 | 1 | 1e-5 | - | - | - | 0.95 | 0.0001 |
| | QNLI | 1.8e-5 | 1 | 1e-5 | - | - | - | 0.9 | 0.0001 |
| | QQP | 1.8e-5 | 1 | 1e-5 | - | - | - | 0.95 | 0.0001 |
| | CoLA | 1.8e-5 | 1 | 1e-5 | - | - | - | 0.95 | 0.1 |
| | SST-2 | 1.8e-5 | 0.1 | 1e-5 | - | - | - | 0.9 | 0.0001 |
| | MRPC | 1.8e-5 | 0.7 | 1e-5 | - | - | - | 0.9 | 0.1 |
| | MNLI | 1.8e-5 | 1 | 1e-5 | - | - | - | 0.9 | 0.1 |

Table 8: Best hyperparameter selection over a sweep of various parameter grids for GLUE tasks. The notation is analogous to Table 6.

| Algorithm | Dataset | ilr | olr | ieps | oeps | o_u | o_d | i_u | i_d |
|-----------|---------|-----|-----|------|------|-----|-----|-----|-----|
| **Avg-Adagrad** | STS-B | 3e-5 | - | 1e-8 | - | - | - | - | - |
| | RTE | 1.5e-4 | - | 1e-6 | - | - | - | - | - |
| | QNLI | 3.3e-4 | - | 0.001 | - | - | - | - | - |
| | QQP | 3.3e-4 | - | 0.001 | - | - | - | - | - |
| | CoLA | 6.7e-5 | - | 1e-6 | - | - | - | - | - |
| | SST-2 | 3.3e-4 | - | 0.001 | - | - | - | - | - |
| | MRPC | 1.5e-4 | - | 1e-6 | - | - | - | - | - |
| | MNLI | 3.3e-4 | - | 0.001 | - | - | - | - | - |
| **Avg-Adam** | STS-B | 1.4e-5 | - | 1e-6 | - | - | - | - | - |
| | RTE | 3e-5 | - | 1e-8 | - | - | - | - | - |
| | QNLI | 6.2e-6 | - | 1e-8 | - | - | - | - | - |
| | QQP | 1.4e-5 | - | 1e-8 | - | - | - | - | - |
| | CoLA | 6.2e-6 | - | 1e-8 | - | - | - | - | - |
| | SST-2 | 6.2e-6 | - | 1e-8 | - | - | - | - | - |
| | MRPC | 3e-5 | - | 1e-8 | - | - | - | - | - |
| | MNLI | 3e-5 | - | 0.0001 | - | - | - | - | - |

Table 9: Best hyperparameter selection over a sweep of various parameter grids for WMT. The conventions are identical with Tables 6-8.

| Algorithm | Dataset | ilr | olr | ieps | oeps | o_u | o_d | i_u | i_d |
|-----------|---------|-----|-----|------|------|-----|-----|-----|-----|
| **Avg-SGD** | TED-T (en-de) | 0.03 | - | - | - | - | - | - | - |
| | TED-T (en-fr) | 0.015 | - | - | - | - | - | - | - |
| | NewsComm (en-fr) | 0.015 | - | - | - | - | - | - | - |
| **Avg-$L_2Clip$** | TED-T (en-de) | 0.89 | - | - | - | - | - | 1.4 | 0.0 |
| | TED-T (en-fr) | 0.89 | - | - | - | - | - | 0.55 | 0.0 |
| | NewsComm (en-fr) | 0.78 | - | - | - | - | - | 0.41 | 0.0 |
| $Bi^2Clip$ | TED-T (en-de) | 1 | 1 | - | - | 0.0001 | 0.0001 | 0.75 | 1e-7 |
| | TED-T (en-fr) | 1 | 1 | - | - | 0.0001 | 0.0001 | 0.75 | 1e-7 |
| | NewsComm (en-fr) | 0.5 | 1 | - | - | 1.5 | 1e-7 | 0.0001 | 5e-5 |
| $Adam^2$ | TED-T (en-de) | 3.2e-4 | 0.0056 | 1e-7 | 0.001 | - | - | - | - |
| | TED-T (en-fr) | 1.8e-5 | 1.8e-5 | 1e-5 | 1e-7 | - | - | - | - |
| | NewsComm (en-fr) | 3.2e-4 | 0.0056 | 1e-5 | 0.001 | - | - | - | - |

# F  ADDITIONAL EXPERIMENTS

$BiClip$ is inspired by the principles of adaptivity, particularly the selection of coordinate-wise learning rates based on historical gradient statistics in adaptive optimizers. It leverages this intuition by efficiently amplifying smaller gradient values while tempering larger gradients. This selective adjustment enables $BiClip$ to maintain computational efficiency while achieving highly competitive performance, as demonstrated in Tables 1 and 2, where it rivals more resource-intensive optimizers such as Adam.

However, Figure 4 highlights how gradient distributions can be distinctly altered by adaptive or clipping operations, which is reflected in their respective optimal learning rates. We note that $L_2$ clipping primarily affects gradients at the extremes—those whose $L_2$-norms exceed a predefined threshold—while leaving the broader gradient distribution largely unchanged during the optimization process. This limited modification contrasts with the more nuanced adjustments achieved by $BiClip$ or Adam.

## F.1  EXPANDED ALGORITHM PERFORMANCE EVALUATION (GLUE)

Table 10: Evaluation results on GLUE Benchmark datasets during test time. Metrics: CoLA (Matthews Correlation Coefficient, MCC), SST-2 (Accuracy), MRPC (Accuracy/F1), STS-B (Spearman/Pearson), QQP (Accuracy/F1), MNLI (Accuracy), QNLI (Accuracy), RTE (Accuracy). Entries marked with 0.0 indicate the actual metric value (averaged across the granularity of each datapoint in the baseline dataset), which implies random guessing or failure to learn. Top **first**, **second**, and **third** best-performing algorithms are highlighted. We note that nested optimization algorithms utilizing adaptivity or coordinate-wise $BiClip$ on both inner and outer optimizers generally achieve greater than 80% averaged performance (out of 100%). For $Adam^2$, preconditioners are transmitted between the inner and outer optimizers, whereas DiLoCo requires maintaining preconditioners on the inner optimizers, both of which incur significant communication or memory overhead.

| Algorithm | MNLI | QNLI | QQP (Acc/F1) | RTE | SST-2 | MRPC (Acc/F1) | CoLA | STS-B (S/P) | Average |
|---|---|---|---|---|---|---|---|---|---|
| **Avg-SGD** McMahan et al. (2017) | 81.13 | 83.21 | 78.71/78.69 | 57.40 | 90.94 | 67.30/80.52 | 0.0 | 26.76/28.20 | 61.17 |
| **Avg-**$L_2Clip$ Yang et al. (2022) | 81.82 | 85.68 | 80.00/79.82 | 54.51 | 91.97 | 68.38/81.22 | 0.0 | 41.27/40.96 | 64.15 |
| **Avg-**$BiClip$ ($L_2$) | 81.95 | 86.16 | 84.62/79.89 | 55.59 | 92.31 | 68.38/81.23 | 0.0 | 36.93/37.22 | 64.03 |
| **Avg-Adagrad** | 84.70 | 88.79 | 87.09/83.34 | 64.26 | 93.34 | 71.56/82.63 | 27.72 | 81.93/81.26 | 76.97 |
| **Avg-Adam** | 84.97 | 89.47 | 87.66/84.09 | 64.62 | **93.80** | 81.86/87.74 | 41.41 | **86.21/86.55** | 80.76 |
| **Avg-**$BiClip$ | 85.08 | 89.45 | 87.83/84.12 | 66.06 | **94.03** | 71.32/82.45 | 41.40 | 84.08/84.48 | 79.12 |
| $Bi^2Clip$ ($L_2$) | 84.31 | 89.20 | 86.36/82.60 | 72.20 | 93.34 | 86.52/90.23 | **60.02** | 82.41/83.00 | 82.74 |
| **Adagrad-SGD** Reddi et al. (2021) | 82.40 | 86.61 | 82.51/77.68 | 71.48 | 92.08 | 85.53/89.52 | 47.80 | 40.37/42.24 | 72.69 |
| **RMSProp-SGD** Reddi et al. (2021) | 84.20 | 88.46 | 87.12/83.30 | **72.56** | 91.85 | 85.50/89.17 | 52.39 | 45.72/41.80 | 74.73 |
| **Adam-SGD** Reddi et al. (2021) | 82.93 | 86.98 | 85.99/80.87 | 66.78 | 90.71 | 87.01/90.09 | 49.93 | 44.48/41.26 | 73.37 |
| **Adam-**$L_2Clip$ | 82.54 | 86.69 | 85.88/80.72 | 59.92 | 89.67 | 85.29/89.90 | 48.54 | 69.19/67.16 | 76.86 |
| **Adagrad-**$BiClip$ | **85.54** | **90.02** | 88.60/**85.05** | **73.36** | 93.23 | 85.78/89.86 | 48.87 | 84.03/85.90 | 82.75 |
| **RMSProp-**$BiClip$ | **85.56** | 89.82 | 88.50/84.44 | 70.75 | **93.69** | 84.80/88.92 | 50.99 | **87.65/87.79** | **82.99** |
| **Adam-**$BiClip$ | 84.26 | 89.20 | **88.64**/84.74 | 69.67 | 92.43 | 86.52/90.09 | **56.12** | 82.83/79.71 | 82.20 |
| **Adam-**$BiClip$ ($L_2$) | 83.18 | 86.47 | 85.63/80.27 | 67.50 | 89.56 | 86.02/89.65 | 53.17 | 74.73/73.48 | 79.06 |
| $Adam^2$ Wang et al. (2021b) | 85.11 | 88.87 | **89.04/85.51** | 71.48 | 92.66 | **87.50/91.03** | 52.70 | 84.47/83.82 | 82.93 |
| **DiLoCo** Douillard et al. (2024) | **85.68** | **89.87** | **88.78/85.19** | 67.87 | 91.89 | **87.99/91.20** | 54.77 | 85.93/84.76 | **83.08** |
| $Bi^2Clip$ | 85.06 | 89.73 | 84.93/83.97 | **76.53** | **93.80** | **89.21/92.44** | **60.08** | **87.07/86.89** | **84.52** |

## F.2  PERFORMANCE UNDER NON-IID DATA

### F.2.1  CUSTOM SHAKESPEARE DATASET

Though not the main focus of this work, in this section, we aim to briefly evaluate the performance of TailOPT and baselines under non-datacenter, distributed environments. We utilized the LEAF repository Caldas et al. (2018), originally a benchmark suite for federated learning, which provides datasets, tools, and baselines to evaluate algorithms under real-world conditions. LEAF emphasizes non-IID data distributions, enabling the study of federated systems where data is naturally heterogeneous across smaller compute nodes. Among the datasets in LEAF, we modified the Shakespeare dataset, originally designed for next-character prediction, where each user now represented a character from Shakespeare's works. After preprocessing, the dataset contained 1144 inner compute nodes, each corresponding to a character's dialogue, with substantial variations in sample sizes, vocabulary, and syntax across compute nodes. This structure mirrors the imbalanced, domain-specific data distributions often encountered in federated learning.

To better align with common NLP tasks, we further modified the Shakespeare dataset by redefining the prediction task from (LSTM) next-character prediction to (transformer) next-token prediction.

More specifically, the text was tokenized into sequences of words rather than characters, making the task more semantically meaningful while retaining the dataset's inherent non-IID nature.

Table 11: Perplexity scores on the Federated Shakespeare Next Word Prediction Task at a 0.1% participation rate, for distillGPT-2 architecture fine-tuning after 3 communication rounds.

| Algorithm | Avg-SGD | Avg-$L_2Clip$ | Avg-$BiClip$ | RMSProp-$BiClip$ | $Bi^2Clip$ | $Adam^2$ |
|---|---|---|---|---|---|---|
| Perplexity Score | 1.9813 | 2.0126 | **1.7827** | 2.0054 | 1.9112 | 1.9445 |

### F.2.2 CUSTOM PHILOSOPHER DATASET

To mitigate potential data leakage, we constructed a custom dataset, termed the Philosopher Dataset, to evaluate the non-IID setting and facilitate training from scratch. The Philosopher Dataset was synthesized by allocating each literary work to one of eight compute nodes, followed by an 80-20 train-test split. These texts were open sourced from Project Gutenberg[3], an extensive online repository offering over 75,000 classic or traditional books while strictly adhering to copyright protections.

Table 12: Composition of the Philosopher Dataset.

| Title | Author | Translator |
|---|---|---|
| The Critique of Pure Reason | Immanuel Kant | J. Meiklejohn |
| The Collected Works of William Hazlitt, Volume One | William Hazlitt | - |
| The Works of Jane Austen | Jane Austen | - |
| The Republic | Plato | Benjamin Jowett |
| War and Peace | Leo Tolstoy | - |
| The Federalist Papers | Alexander Hamilton, John Jay, James Madison | - |
| The Count of Monte Cristo | Alexandre Dumas | - |
| The Brothers Karamazov | Fyodor Dostoevsky | Constance Garnett |

We instantiated a shallower GPT-2 architecture comprising 2 layers, 256 embedding dimensions, and 4 attention heads. This model was trained from scratch on the Philosopher Dataset. The training results are summarized in Table 13.

Table 13: Perplexity scores on the Philosopher Next Word Prediction Task at a 100% participation rate for the compressed GPT-2 architecture after 3 communication rounds.

| Algorithm | Avg-SGD | Avg-$L_2Clip$ | Avg-$BiClip$ | RMSProp-$BiClip$ | $Bi^2Clip$ | $Adam^2$ |
|---|---|---|---|---|---|---|
| Perplexity Score | 2.6361 | 2.1183 | **1.6266** | 1.7983 | 2.3488 | 2.5861 |

**Discussion.** In the synthesized non-IID setting, we observe that algorithmic instantiations employing joint adaptivity or adaptive approximations–i.e., incorporating adaptivity or its efficient approximations at both the inner and outer optimizers–tend to underperform slightly. This aligns with the theoretical intuition that highly sensitive, rapidly adapting optimizers are more susceptible to unmitigated client drift, effectively overfitting to the biases of local data shards at the inner optimizers. However, Avg-$BiClip$, which integrates a clipping mechanism to regulate noise variance and stabilize optimization dynamics, exhibits notably robust performance. In particular, Avg-$BiClip$ achieves the strongest results in settings with high data heterogeneity across compute nodes, suggesting that $BiClip$ mitigates not only noise variance but also client drift. We further compare these findings to results on the synthetic dataset (Appendix E.1) where noise-injected data were distributed IID across nodes, contrasting with the Shakespeare and Philosopher datasets, which are explicitly designed to be non-IID.

We note that the perplexities obtained are lower compared to those achieved on larger text datasets, such as WikiText-103 or large-scale Common Crawl subsets (e.g., distillGPT reportedly achieves a perplexity of around 16 on the WikiText-103 benchmark, a long-term dependency language modeling dataset)[4]. This arises from the smaller size of the Shakespeare and Philosopher datasets in comparison

---

[3] https://www.gutenberg.org/
[4] https://github.com/huggingface/transformers/tree/main/examples/research_projects/distillation

to larger benchmarks. Finally, we provide the optimal hyperparameters for the non-IID experiments in Table 14.

Table 14: Best hyperparameter selection over a sweep of various parameter grids. The conventions are identical with Tables 6-9.

| Algorithm | Dataset | ilr | olr | ieps | oeps | o_u | o_d | i_u | i_d |
|---|---|---|---|---|---|---|---|---|---|
| **Avg-SGD** | Shakespeare | 0.012 | - | - | - | - | - | - | - |
| | Philosopher | 0.15 | - | - | - | - | - | - | - |
| **Avg-**$L_2Clip$ | Shakespeare | 0.56 | - | - | - | - | - | 0.55 | 0 |
| | Philosopher | 1 | - | - | - | - | - | 0.41 | 0 |
| **Avg-**$BiClip$ | Shakespeare | 1 | - | - | - | - | - | 0.0001 | 3.3e-5 |
| | Philosopher | 1 | - | - | - | - | - | 0.0001 | 3.3e-5 |
| **RMSProp-**$BiClip$ | Shakespeare | 0.067 | 2.2e-4 | - | 1e-5 | - | - | 0.75 | 1e-7 |
| | Philosopher | 0.067 | 0.0046 | - | 0.001 | - | - | 0.75 | 1e-7 |
| $Bi^2Clip$ | Shakespeare | 1 | 1 | - | - | 1.5 | 1e-7 | 0.0001 | 0.0001 |
| | Philosopher | 1 | 1 | - | - | 1.5 | 1e-7 | 0.0001 | 5e-5 |
| $Adam^2$ | Shakespeare | 1.8e-5 | 0.0056 | 1e-7 | 0.001 | - | - | - | - |
| | Philosopher | 1.8e-5 | 0.0056 | 1e-5 | 1e-5 | - | - | - | - |

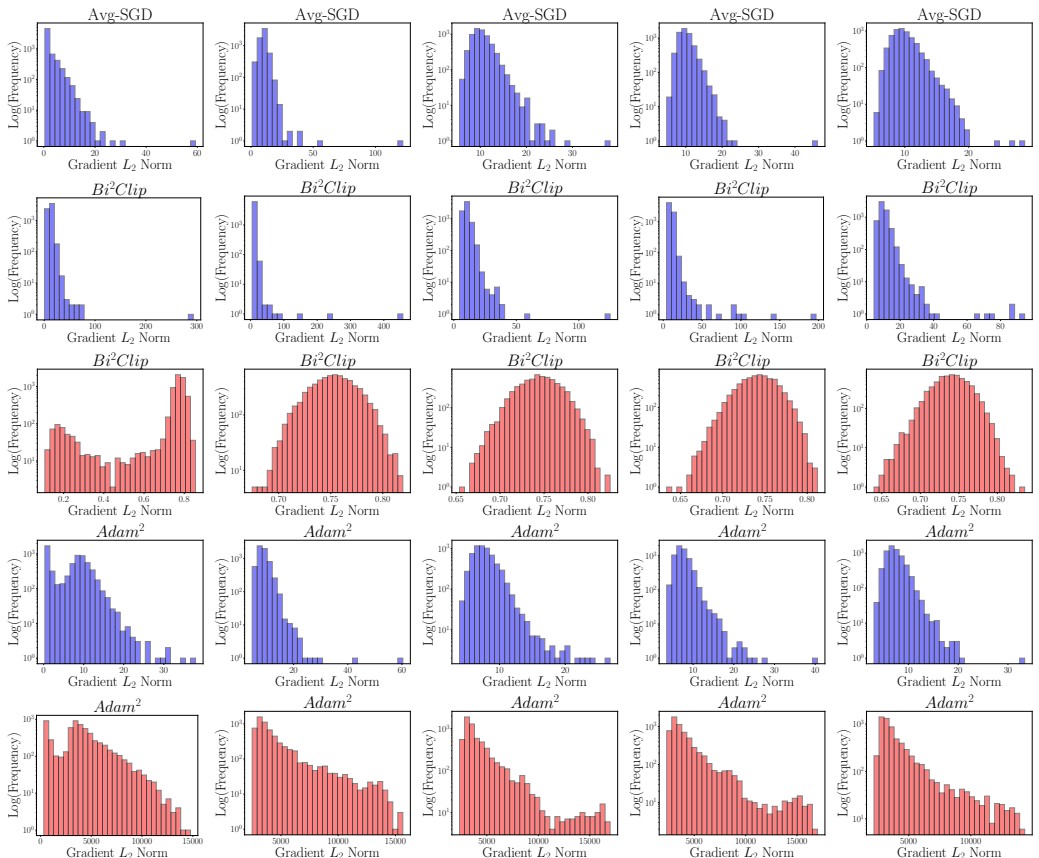

Figure 4: Gradient statistics for MNLI in the GLUE Benchmark across different algorithms for the first 5 communication rounds, where rounds increase from left to right. (Top) We visualize local minibatch stochastic gradient (used as model updates in Avg-SGD) distributions, where the outliers can dominate model updates upon outer pseudogradient aggregation. The $BiClip$ and Adam optimizers mitigate this phenomenon in different ways. (Middle) Row 2 displays the local gradients accumulated from all inner optimizers during $Bi^2Clip$ prior to clipping, which uncovers the presence of outliers akin to those visible in Avg-SGD. In Row 3, the identical gradients are plotted after the coordinate-wise $BiClip$ operation is applied. It is observed that $BiClip$ stabilizes updates by thresholding large and small gradient coordinates, constraining model update lengths within a defined range. The distribution of gradient lengths have changed significantly, with outliers autonomously being mollified. (Bottom) Similar to above, row 4 shows the accumulated gradient lengths across all inner optimizers while training via $Adam^2$. In row 5, it is observed that Adam amplifies gradients across a larger scale, with optimal hyperparameters accordingly downscaling model updates by utilizing smaller learning rates at both inner and outer optimizers. Optimal inner optimizer learning rates are 0.0059, 0.5, and 1.8e-5 for Avg-SGD, $Bi^2Clip$, and $Adam^2$, respectively, with corresponding outer optimizer learning rates of 1 and 3.2e-4 for the latter two algorithms. Test-time results show that $Bi^2Clip$ outperforms $Adam^2$, which in turn outperforms Avg-SGD (Table 1). Finally, we note that upon centering, the aggregate update gradient histograms in red depict the stochastic gradient noise distributions upon application of the optimizer strategy. $BiClip$ attenuates the pure gradient noise (in blue) by projecting the noise distribution to an almost bell-shaped curve (in red), while Adam implicitly samples gradient noise from a left-leaning, skewed distribution.

