# OpenReview forum: "Efficient Distributed Optimization under Heavy-Tailed Noise"
_ICLR.cc/2025/Workshop/MCDC — MCDC @ ICLR 2025_

### Official Review · Reviewer_YyBz · 2025-02-23

**Rating:** 7
**Confidence:** 3
**Fit:** 5

**Summary:**

The authors present TailOPT, a framework for handling heavy-tailed gradient noise in distributed training. The main technical contribution is BiClip, which does coordinate-wise clipping to mimic adaptive optimizers without their memory overhead. While the theoretical analysis aims to prove convergence under heavy-tailed noise with potentially unbounded variance, there are some gaps in the proofs that need addressing.

**Reason For Giving A Higher Score:**

This is tackling a real problem in distributed training and the solution is both theoretically grounded and practically useful. BiClip is a neat trick for getting adaptive behavior without the memory hit. The GLUE results are impressive and the analysis, while not perfect, establishes meaningful guarantees.

**Reason For Giving A Lower Score:**

The math needs tightening up, especially in the convergence proofs. Experiments are too NLP-focused to really show broad applicability. Missing practical details about implementation and parameter tuning that would be crucial for adoption.

**Strengths And Weaknesses:**

The key innovation here is BiClip's approach to amplifying small gradients while tempering large ones at a coordinate level. This is clever and the empirical results on GLUE and WMT benchmarks look solid. The implementation is also memory-efficient compared to methods like Adam, which will matter for large-scale training.

That said, I'm concerned about several aspects:
1. The asymptotic approximation in eq. 14-15 requires more rigorous justification, particularly in the treatment of boundary terms when integrating by parts and in proving that the claimed limit holds uniformly. The current analysis does not adequately address potential singularities when \nu <= 0 or justify the exchange of limits, which is crucial for establishing the tightness of the convergence bounds
2. All experiments are on NLP tasks; I'd like to see empirical analysis on other domains like CV or RL
3. No real discussion of how sensitive the method is to the choice of clipping thresholds

**Suggestions:**

See about in the weaknesses section.

---

### Official Review · Reviewer_3djD · 2025-02-27

**Rating:** 9
**Confidence:** 4
**Fit:** 4

**Summary:**

The paper introduces TailOPT as an efficient framework designed to address heavy-tailed noise. TailOPT utilizes BiClip mechanism that clips the coordinate-wise values from both and below . Theoretical guarantees and empricial evidence are given to show the effectiveness of the proposed method.

**Reason For Giving A Higher Score:**

1. Rigorous theoris are provided to show theconvergence of BiClip under heavy-tailed noise.
2. Abundant ablation studies give good insights on how and when BiClip methods outperform traditional adaptive methods.

**Reason For Giving A Lower Score:**

N/A

**Strengths And Weaknesses:**

Strength:
1. The paper is well-written, with the movtivation clearly described and excellent expositions.
2. Theoretical guarantees of TailOPT are provided.
3. Experimental configuration is extensive. The abundant ablation studies greatly help in demonstrating the effectiveness of the proposed method.

Weakness/Minor Comments:
1. Most of the existing works [1] show a one-sided (upper side) clipping is sufficient for convergence under heavy-tailed noises. The authors claim BiClip takes advantage of the adaptivity by incorporating a two-sided clipping mechanism. It might be better to make it clear on how/whether the lower side clipping (d) has an effect on the heavy-tailed noise.
2. It's a little surprising to see BiClip can outperform Adam in most cases, even without leveraging momentum. Does BiClip method natively support momentum? If so, how momentum mechanism can be applied to the client updates? Does applying momentum to BiClip improve the performance (at some additional cost)?

[1] Zhang, J., Karimireddy, S.P., Veit, A., Kim, S., Reddi, S., Kumar, S. and Sra, S., 2020. Why are adaptive methods good for attention models?. Advances in Neural Information Processing Systems, 33, pp.15383-15393.

**Suggestions:**

1. A proof sketch demonstrating the key challenges in the theories will be informative and helpful.

---

### Decision · Program_Chairs · 2025-03-06

**Decision:**

Accept

**Comment:**

This distributed optimization work proposes an adaptive inner clipping that enables good performance with highly noisy gradients with less memory overhead than existing counterparts. Distributed optimization is a relevant topic of this workshop, and the reviewers all recommend acceptance Therefore, we're happy to accept this work to the workshop.